# Towards a conceptualization of the hydrological processes behind changes of young water fraction with elevation: a focus on mountainous alpine catchments

Alessio Gentile[1], Davide Canone[1], Natalie Ceperley[2,3], Davide Gisolo[1], Maurizio Previati[1], Giulia Zuecco[4,5], Bettina Schaefli[2,3], and Stefano Ferraris[1]

[1]Interuniversity Department of Regional and Urban Studies and Planning (DIST),
Politecnico and Università degli Studi di Torino, Torino, Italy
[2]Institute of Earth Surface Dynamic (IDYST), Faculty of Geosciences and Environment (FGSE),
University of Lausanne, Lausanne, Switzerland
[3]Institute of Geography (GIUB) and Oeschger Centre for Climate Change Research (OCCR),
University of Bern, Bern, Switzerland
[4]Department of Land, Environment, Agriculture and Forestry (TESAF), University of Padova, Legnaro, Italy
[5]Department of Chemical Sciences (DiSC), University of Padova, Padua, Italy

**Correspondence:** Bettina Schaefli (bettina.schaefli@giub.unibe.ch)

**Abstract.** The young water fraction ($F_{yw}^*$), defined as the fraction of catchment outflow with transit times of less than 2–3 months, is increasingly used in hydrological studies that exploit the potential of isotope tracers. The use of this new metric in catchment intercomparison studies is helpful to understand and conceptualize the relevant processes controlling catchment functioning. Previous studies have shown surprising evidence that mountainous catchments worldwide yield low $F_{yw}^*$. These low values have been partially explained by isolated hydrological processes, including deep vertical infiltration and long groundwater flow paths. However, a thorough framework illustrating the relevant mechanisms leading to a low $F_{yw}^*$ in mountainous catchments is missing.

The main aim of this paper is to give an overview of what drives $F_{yw}^*$ variations according to elevation, thus clarifying why it generally decreases at high elevation. For this purpose, we assembled a data set of 27 study catchments, located in both Switzerland and Italy, for which we calculate $F_{yw}^*$. We assume that this decrease can be explained by the groundwater storage potential, quantified by the areal extent of Quaternary deposits over a catchment ($F_{qd}$), and the low-flow duration (LFD) throughout the period of isotope sampling (PoS). In snow-dominated systems, LFD is strictly related to the snowpack persistence, quantified through the mean fractional snow cover area ($F_{SCA}$). The drivers are related to the catchment storage contribution to the stream that we quantify by applying a cutting-edge baseflow separation method to the discharge time series of the study sites and by estimating the mean baseflow fraction ($F_{bf}$) over the PoS.

Our results suggest that Quaternary deposits could play a role in modulating $F_{yw}^*$ elevation gradients via their capacity to store groundwater, but subsequent confirmation with further, more detailed geological information is necessary. LFD indicates the proportion of PoS in which the stream is sustained and dominated by stored water coming from the catchment storage. Accordingly, our results reveal that the increase of LFD at high elevations, to a large extent driven by the persistence of winter snowpacks and the simultaneous lack of a liquid water input to the catchments, results in lower $F_{yw}^*$. In our data set, $F_{bf}$ reveals a strong complementarity with $F_{yw}^*$, suggesting that the latter could be estimated as $F_{yw}^* \simeq 1 - F_{bf}$ for catchments without stable water isotope measurements.

As a conclusion, we develop a perceptual model that integrates all the results of our analysis into a framework for how hydrological processes control $F_{yw}^*$ according to eleva-

tion. This lays the foundations for an improvement of the theory-driven models.

## 1 Introduction

Mountainous alpine catchments are often assumed to generate high shares of rapid surface or subsurface runoff due to the presence of exposed bedrock and steep landscapes. Consequently, the role of groundwater storage in high-elevation catchments has been often neglected (Hayashi, 2020). On the contrary, multiple worldwide studies quantified a considerable groundwater input to streamflow in high mountain catchments using tracer or water balance methods (Somers and McKenzie, 2020). Several studies from the Rocky Mountains and Andes show that, on average, about 47 % of groundwater annually sustains the streamflow (Saberi et al., 2019; Somers et al., 2019; Carroll et al., 2018; Harrington et al., 2018; Cowie et al., 2017; Baraer et al., 2009, 2015; Gordon et al., 2015; Frisbee et al., 2011; Liu et al., 2004; Clow et al., 2003). Similar percentages, 49 % and 48 %, are also found in the Himalayas and the Alps, respectively (Chen et al., 2018; Engel et al., 2016; Käser and Hunkeler, 2016; Williams et al., 2016; Wilson et al., 2016; Andermann et al., 2012). It is well known that the water is stored longer than a year or a few years and that stored water plays a key role in streamflow generation processes (McDonnell, 2017; Jasechko, 2019). The study of water age has implications for predicting the timing of nutrient cycles and pollutant transport, since water age and solute dynamics are closely coupled (Li et al., 2021). Nevertheless, water age quantification is not straightforward.

Kirchner (2016a, b) proposed a new metric to quantify the share of catchment outflow with transit times lower than roughly 0.2 years or 2–3 months: the young water fraction. This metric can be conveniently inferred from the dampening effect that a catchment has on the seasonal cycle of stable water isotopes in precipitation, i.e., by estimating the ratio of the amplitudes of the seasonal cycles of stable water isotopes in streamflow and in precipitation (Kirchner, 2016a). In this method, the seasonal cycle of stream water isotope measurements is modeled using a sine wave that can be flow weighted or not, using the discharge measured at the moment of sampling as a weight (von Freyberg et al., 2018). Isotopes measured in precipitation can be modeled with a sine function weighted according to the volume of precipitation to reduce the influence of low-precipitation periods and to account for temporally aggregated rainfall samples (von Freyberg et al., 2018). Flow-weighted fits to the seasonal tracer cycles predict the flow-weighted average young water fraction ($F_{yw}^*$) in streamflow, while unweighted fits to the seasonal tracer cycles predict the unweighted one ($F_{yw}$) (Kirchner, 2016b). Gallart et al. (2020a) recently highlighted the advantages of the flow-weighted analysis to compensate for subsampled high-flow periods, thus reducing the under-

estimation of the young water fraction. Hereafter, we will use the symbol "*" for referring to a flow-weighted variable, in order to be consistent with previous studies (von Freyberg et al., 2018; Gallart et al., 2020a).

$F_{yw}^*$ is increasingly used in hydrological studies because it has the advantage of being free from the aggregation errors inherent to mean transit time (MTT) estimates obtained through the classical convolution approach (Kirchner, 2016a). Even more so, $F_{yw}^*$ is an informative descriptor of catchment hydrological functions, of nutrients cycles and of pollutant transport (Stockinger et al., 2019; Benettin et al., 2017; Jasechko et al., 2016; Xia et al., 2023). For these reasons, this new metric is useful for catchment intercomparison studies to find what are the main hydroclimatic and landscape characteristics that drive the transit times of water lower than a threshold age, which varies from about 2 to 3 months. Indeed, previous work has tried to study the relationship between $F_{yw}^*$ and catchment characteristics. von Freyberg et al. (2018) found that young water fractions of 22 Swiss catchments are significantly positively correlated with selected hydroclimatic indices and with the fraction of saturated area, suggesting that $F_{yw}^*$ depends on catchment wetness, which promotes rapid flow paths. Interestingly, von Freyberg et al. (2018) found a statistically significant positive correlation with elevation after removing the five snow-dominated catchments, which expressed the smallest $F_{yw}^*$ (von Freyberg et al., 2018). Likewise, Lutz et al. (2018) estimated $F_{yw}^*$ for 24 catchments in Germany and found the smallest values for higher-elevation sites. These results are partially consistent with those of Jasechko et al. (2016), who based on the analysis of 254 watersheds worldwide discovered a reduction of $F_{yw}^*$ in mountainous, steeper terrains. This could be related to deep vertical infiltration caused by fractures generated by high rock stress in complex terrain morphologies or by freely draining soils (i.e., cambisols and luvisols), both associated with high-elevation environments (Lutz et al., 2018; Jasechko et al., 2016; Gleeson et al., 2014). In addition, topographic roughness increases flow path and, correspondingly, transit time (Gleeson and Manning, 2008; Frisbee et al., 2011; Jasechko et al., 2016). Despite these studies, there is still a lack of a unified framework of how the variation among mountainous catchments results in less young water at high elevation.

An early example from the Swiss Alps showed that high celerity originates from massive meltwater infiltration that pushes out groundwater reserves: streamflow following snowmelt is older than meltwater infiltrated in the current year (Martinec, 1975). The resulting effect on water partitioning between the surface and the subsurface should be analyzed considering the temporal concentration of water input during the snowmelt period, but this remains largely unexplored (Rey et al., 2021). Despite this lack of studies on water partitioning during snowmelt, several studies have demonstrated the pivotal role of snowmelt in recharging groundwater during summer in high-elevation environments (Hayashi,

2020; Cochand et al., 2019; Du et al., 2019; Flerchinger et al., 1992).

From a water modeling perspective and thus from a water age perspective, snowpack storage and groundwater storage can be considered a single entity: they both constitute catchment storage. Therefore, the analytical estimation of $F_{yw}^*$ must reflect this "conceptual" decision of whether to consider the snowpack storage to be part of the catchment storage. This point has been previously addressed by von Freyberg et al. (2018). If total precipitation is considered as catchment input (direct input case), the snowpack is implicitly considered to be part of the catchment storage, and $F_{yw}^*$ results from the combination of snowpack and subsurface storage. In this direct input case, $F_{yw}^*$ is computed from the amplitudes of the seasonal cycles of stable isotopes of water in precipitation ($A_P$) and streamflow ($A_s^*$). If total *liquid* water input (composed of rainfall and snowmelt before sometimes called equivalent precipitation) is considered to be catchment input, $F_{yw}^*$ is computed based on the amplitudes of the cycles in equivalent precipitation ($A_{Peq}$) and in streamflow ($A_s^*$). This $F_{yw}^*$ value then results from subsurface storage alone, since snowpack storage is excluded from the catchment storage (von Freyberg et al., 2018). If $F_{yw}^*$ is estimated using a direct input setting (i.e., total precipitation directly), $F_{yw}^*$ is expected to be smaller, since the catchment storage is larger (von Freyberg et al., 2018). Also, Ceperley et al. (2020) investigated the role of water input from snow in $F_{yw}^*$ estimation, concluding that the low values in high alpine snow-dominated catchments result from a combination of snow cover effects and the storage in the subsurface. In the present work, the main aim is not to address how the snowpack affects $F_{yw}^*$ estimation in a single catchment, as this was treated previously (von Freyberg et al., 2018; Ceperley et al., 2020), but to investigate the hydrological processes (also related to the snowpack storage) that lead to variations in $F_{yw}^*$ between catchments located at different elevations with a focus on high-elevation alpine catchments.

Some authors have revealed the possibility of Quaternary deposits (e.g., talus, moraine and alluvium) to store groundwater in high-elevation alpine catchments (Arnoux et al., 2021; Hayashi, 2020; Christensen et al., 2020). The stored water in these deposits can in fact sustain streamflow during the low-flow period (Hayashi, 2020; Arnoux et al., 2021), as supported by the strong positive correlation found by Arnoux et al. (2021) between the fraction of Quaternary deposits and the winter flow index (a low-flow indicator reflecting the groundwater contribution to the stream) for 13 alpine catchments. During winter, the period without liquid water input can last 6 months or more in high-elevation catchments. The occurrence of such long periods of low flows hints towards important amounts of stored water (or old water) that are well connected to the stream network, and it thereby remains accessible throughout the low-flow period (Somers et al., 2019).

To further discuss the role of low flow in $F_{yw}^*$ estimation, let us first consider that $F_{yw}^*$ can be theoretically estimated based on the flow-weighted average of young water fractions (Kirchner, 2016b):

$$F_{yw}^* = \frac{A_S^*}{A_P} \simeq \frac{\sum\limits_{i=1}^{n} Q(t_i) F_{yw}(t_i)}{\sum\limits_{i=1}^{n} Q(t_i)}, \tag{1}$$

where $n$ is the number of time steps (e.g., days) in the period of isotope sampling, PoS, $Q(t_i)$ is the discharge at the time $t_i$ (e.g., daily discharge) and $F_{yw}(t_i)$ is the young water fraction at the time $t_i$ (e.g., daily young water fraction). As is clear from Eq. (1), $F_{yw}^*$ becomes low if either $F_{yw}(t_i)$ is low for high flows or if $F_{yw}(t_i)$ is very low for many time steps or both. The low-flow periods correspond to the recession periods in which there is no new rainfall or meltwater input in the catchments. Thus, during these periods, the catchment storage releases stored water (or old water) to the stream sustaining the streamflow (Hayashi, 2020). Thus, we can anticipate that low $Q(t_i)$ values imply low $F_{yw}(t_i)$ values. Accordingly, the proportion of the low-flow period during a specified time window should reduce the amount of young water reaching the stream during that time window. Nevertheless, the $F_{yw}(t_i)$ is higher during high-flow (wet) periods (von Freyberg et al., 2018; Wilusz et al., 2017; Gallart et al., 2020b). Thus, the overall effect of the proportion of low-flow and high-flow periods upon $F_{yw}^*$ remains a priori unclear. It is however tempting to think that the duration of low-flow period or the share of baseflow could explain $F_{yw}^*$ variations at different elevations (since both low-flow duration and the share of baseflow change with elevation). In addition, in high-elevation, snow-dominated catchments, the persistence of the snowpack is the main driver of the long low-flow duration, since the low-flow period at high elevation corresponds to the presence of the seasonal snowpack (corresponding to an absence of *liquid* water input), while the high-flow period is generally snowmelt driven. Such snowmelt generally occurs in late spring or summer, and it is likely to be older than 2–3 months (because the peak snow fall occurred 3 months earlier). As a result, summer discharge mainly consists of old water, either of current snowmelt that reaches the stream via faster surface or subsurface flow paths, or old snowmelt (main component of groundwater storage) pushed out in the stream by infiltrated rainwater or meltwater. Part of the snowpack can release young water, but this is a minor component in catchments with a seasonal snowpack. In contrast, in catchments with an ephemeral snowpack, it is common to observe intermittent winter snowmelt that is likely younger than 2–3 months: snowmelt is temporally close to snowfall. In this case, streamflow receives relatively more young water from short-lived snowpacks. However, it is still unclear if seasonal or ephemeral snow cover dynamics can affect the $F_{yw}^*$ (Ceperley et al., 2020).

An innovative focus of our work is on variables that were not previously considered for explaining elevation gradients of young water fractions. We specifically exclude catchment size, annual precipitation, bedrock porosity, pasture cover and open water cover that have been discussed and shown to have little correlation in the work of Jasechko et al. (2016).

A special case in terms of explanatory variables is mean annual precipitation: Jasechko et al. (2016) did not observe any significant correlation between the $F_{yw}^*$ and annual precipitation in their worldwide study. Lutz et al. (2018) found that $F_{yw}^*$ decreases with increasing mean annual precipitation, based on 24 catchments in Germany. In contrast, in the relatively wet rainfall-dominated and hybrid catchments studied by von Freyberg et al. (2018), $F_{yw}^*$ was shown to increase with mean monthly precipitation and correspondingly also with elevation. In their study, discharge (unsurprisingly correlated with precipitation) was considered as a proxy of catchment wetness, which favors rapid flow paths and thereby increases $F_{yw}^*$ (von Freyberg et al., 2018). In snow-dominated systems, the use of mean annual precipitation as a proxy for catchment wetness could be misleading because the seasonal snowpack leads to a very dry period of the year despite the high *solid* water input. In other words, the temporal concentration of the liquid water input is the relevant variable. Indeed, the saturation of the system (i.e., high wetness conditions) can be observed also when the annual precipitation is low if a large volume of water (stored in the snowpack) is released in a relatively concentrated time interval. Indeed, despite the fact that precipitation and, correspondingly, discharge are higher in snow-dominated than in rainfall-dominated catchments, $F_{yw}^*$ is generally lower in snow-dominated systems that are potentially wetter than rainfall-dominated ones. This suggests that the precipitation can only partially explain the variations of $F_{yw}^*$ and that other variables should be put under observation.

Accordingly, we omit here total annual precipitation as an explanatory variable of low $F_{yw}^*$ in snow-dominated catchments (but we consider precipitation for rainfall-dominated and hybrid catchments) and study a new set of hydrological variables to gain new insights into $F_{yw}^*$ along elevation gradients: the fraction of Quaternary deposits ($F_{qd}$), the mean fraction of baseflow ($F_{bf}$), the low-flow duration (LFD) and the mean fractional snow cover area ($F_{SCA}$), defined in detail in Sect. 3.2 and 3.3. We first describe the data set (Sect. 2). Then, we present the $F_{yw}^*$ estimation method (Sect. 3.1) followed by the correlation analysis of the selected hydrological variables with the estimated $F_{yw}^*$, and we bring these results back into the ongoing scientific discussion of $F_{yw}^*$ (Sect. 4.2–4.6).

## 2   Study sites

We analyze 27 study catchments located both in Switzerland and Italy integrating observations from multiple pub-

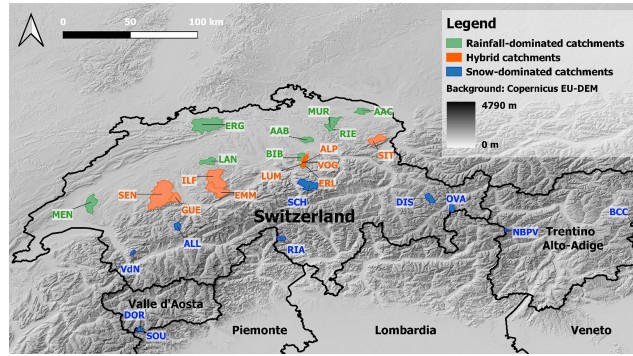

**Figure 1.** Location of the 27 study catchments with indication of the hydroclimatic regime.

lished data sets (25 catchments) with new additional observations (2 catchments) (Fig. 1). Geomorphological and hydroclimatic characteristics of the study sites are reported in Table 1.

Specifically, we assembled the 22 Swiss catchments studied by von Freyberg et al. (2018) with the three alpine catchments investigated by Ceperley et al. (2020) (Vallon de Nant, Noce Bianco at Pian Venezia and Bridge Creek catchment) into a single data set. Hereafter, we refer to these catchments with the ID reported in the above-mentioned published papers (Table 1). We also consider two additional high-elevation catchments located near the Nivolet Pass (Valsavaranche, Aosta Valley, Italy) (Gisolo et al., 2022). In this alpine environment, we monitor the mainstream, called "Dora del Nivolet", and a secondary river called "Source". Hereafter we refer to these catchments with the IDs DOR and SOU, respectively. A detailed description of the DOR and SOU catchments is reported in the Supplement.

The von Freyberg et al. (2018) data set includes catchments with areas between 0.7 and 351 km$^2$ and mean elevations between 472 and 2369 m a.s.l. With the five catchments added here, the complete data set includes catchment areas between 0.14 and 359 km$^2$ and spans mean elevation between 472 and 3049 m a.s.l. The mean monthly precipitation ranges between 61.3 and 168.7 mm per month, while mean discharge ranges between 28.6 and 138.9 mm per month. The mean slope ranges from 4 to 34°, and our study sites reveal an increase of steepness with elevation (Fig. 2a). Precipitation increases with elevation until 1500 m a.s.l., above which it decreases (Fig. 2c), CE2 highlighting a change of precipitation regime as described by previous studies (Santos et al., 2018). The five catchments added to the initial data set of von Freyberg et al. (2018) allow the analysis to explore the high-elevation regions (mean elevation > 1500 m a.s.l.) that were previously poorly represented. Most of the study catchments reveal a sedimentary, bedrock but dolomitic and metamorphic bedrocks, characteristic of high-elevation sites, are also included in our data set. Moreover, the presence of unconsolidated Quaternary deposits is widespread among our

**Table 1.** Catchment geomorphological and hydroclimatic characteristics. The catchment area and mean slope are directly calculated in Google Earth Engine. For the slope calculation, we use the Shuttle Radar Topography Mission (SRTM) DEM (Farr et al., 2007). We obtained mean elevation and precipitation information of the existing data set directly from published papers (von Freyberg et al., 2018; Ceperley et al., 2020). Discharge ($Q$), precipitation ($P$) and isotopic composition ($\delta^{18}O$) data are all referred to CE1 the period of sampling (PoS) indicated in this table. The letter in brackets in the first column indicates the hydroclimatic regime: (R) is rainfall dominated, (H) is hybrid and (S) is snow dominated.

| ID (Regime) | Area (km$^2$) | Mean elevation (m a.s.l.) | Elevation range (min–max) | Mean slope (°) | Dominant geology | Monthly $P$ (mm per month) | Monthly $Q$ (mm per month) | PoS $\delta^{18}O$, $Q$, $P$ |
|---|---|---|---|---|---|---|---|---|
| AAB (R) | 46.07 | 635 | 519–1092 | 5.73 | Sedimentary rock | 106.1 | 56.48 | Sep 2010–Feb 2013 |
| AAC (R) | 47.25 | 472 | 408–560 | 4.02 | Unconsolidated sediments | 85.1 | 35.73 | Jul 2010–Dec 2011 |
| ALL (S) | 28.71 | 1852 | 1293–2742 | 25.48 | Sedimentary rock and unconsolidated sediments | 99.4 | 118.04 | Sep 2010–May 2015 |
| ALP (H) | 46.59 | 1154 | 845–1894 | 16.50 | Sedimentary rock and unconsolidated sediments | 158.2 | 123.52 | May 2010–Dec 2015 |
| BCC (S) | 0.14 | 2121 | 1932–2515 | 22.88 | Dolomite | 100.3 | 111.83 | Mar 2010–Oct 2017 |
| BIB (R) | 31.83 | 999 | 827–1495 | 12.43 | Sedimentary rock and unconsolidated sediments | 150.2 | 94.78 | May 2010–Nov 2015 |
| DIS (S) | 42.75 | 2369 | 1663–3139 | 26.28 | Metamorphic rock | 76.4 | 108.11 | Nov 2010–May 2015 |
| DOR (S) | 16.99 | 2711 | 2390–3430 | 19.37 | Metamorphic rock | 147.4 | 107.79 | Nov 2017–Jan 2022 |
| EMM (H) | 124.03 | 1285 | 743–2216 | 19.71 | Sedimentary rock | 116.6 | 91.99 | Jun 2010–Nov 2013 |
| ERG (R) | 260.47 | 584 | 305–1165 | 13.86 | Sedimentary rock | 87.7 | 37.88 | Jun 2010–Nov 2015 |
| ERL (H) | 0.74 | 1359 | 1117–1650 | 13.53 | Sedimentary rock | 162.4 | 138.04 | Jul 2010–May 2015 |
| GUE (H) | 55.23 | 1037 | 556–2152 | 16.84 | Sedimentary rock and unconsolidated sediments | 94.9 | 77.69 | Jul 2010–Dec 2012 |
| ILF (H) | 186.94 | 1037 | 681–2087 | 19.36 | Sedimentary rock | 127.5 | 81.09 | Jul 2010–May 2015 |
| LAN (R) | 59.76 | 760 | 598–1100 | 10.08 | Sedimentary rock | 118.2 | 54.78 | Jul 2010–May 2015 |
| LUM (H) | 1.20 | 1336 | 1092–1508 | 12.49 | Sedimentary rock | 157.1 | 113.63 | Oct 2010–Nov 2015 |
| MEN (R) | 105.02 | 679 | 447–926 | 6.19 | Sedimentary rock and unconsolidated sediments | 89.3 | 28.64 | Jul 2010–Feb 2013 |
| MUR (R) | 79.92 | 648 | 467–1036 | 10.52 | Sedimentary rock and unconsolidated sediments | 116.6 | 60.57 | Jul 2010–Nov 2014 |
| NBPV (S) | 8.39 | 3049 | 2298–3769 | 23.27 | Metamorphic and sedimentary rock | 117.8 | 137.80 | May 2013–Sep 2015 |
| OVA (S) | 26.87 | 2364 | 1519–3160 | 32.73 | Dolomite | 61.3 | 73.21 | Aug 2010–Sep 2013 |
| RIA (S) | 23.85 | 1986 | 881–2908 | 32.93 | Metamorphic rock | 129.3 | 143.49 | Jul 2010–Dec 2012 |
| RIE (R) | 3.18 | 794 | 671–938 | 13.23 | Sedimentary rock and unconsolidated sediments | 121.1 | 85.58 | Jul 2010–Feb 2013 |

| ID (Regime) | Area (km$^2$) | Mean elevation (m a.s.l.) | Elevation range (min–max) | Mean slope (°) | Dominant geology | Monthly $P$ (mm per month) | Monthly $Q$ (mm per month) | PoS $\delta^{18}$O, $Q$, $P$ |
|---|---|---|---|---|---|---|---|---|
| SCH (S) | 107.61 | 1719 | 487-3260 | 28.78 | Sedimentary rock and unconsolidated sediments | 140.0 | 138.93 | Apr 2011–May 2015 |
| SEN (H) | 350.24 | 1068 | 554–2184 | 15.35 | Sedimentary rock | 95.2 | 53.66 | Oct 2010–Mar 2013 |
| SIT (H) | 74.23 | 1301 | 768–2500 | 22.15 | Sedimentary rock | 168.7 | 115.47 | Nov 2010–May 2015 |
| SOU (S) | 0.16 | 2636 | 2390–2790 | 25.74 | Metamorphic rock | 147.4 | 64.47 | Nov 2017–Jan 2022 |
| VdN (S) | 13.55 | 1966 | 1189–3051 | 34.00 | Sedimentary rock | 132.6 | 99.14 | Nov 2015–Dec 2018 |
| VOG (H) | 1.57 | 1335 | 1038–1540 | 18.42 | Sedimentary rock | 162.2 | 120.24 | Jun 2010–Nov 2015 |

study catchments: only two catchments (BCC and SOU) do not reveal this type of geology. The complete data set now explores case studies from the Swiss plateau and pre-alpine area; from the Jura; and from five different alpine regions, including the northern part of the Swiss Alps, the southern Swiss Alps (Alpi Ticinesi), the western Italian Alps (Alpi Graie), the Rätische Alps and the Dolomites. Overall, this represents a good range of geologies as well as of climatic conditions.

In order to be consistent with previous studies (von Freyberg et al., 2018; Staudinger et al., 2017), we classify the 23 Swiss catchments according to the hydroclimatic regimes proposed by Staudinger et al. (2017) which group the regimes defined by Weingartner and Aschwanden (1992) in three categories: rainfall dominated (R), hybrid (H) and snow dominated (S). For the four Italian catchments, where the aforementioned classification schemes cannot be rigorously applied, we use that proposed by Stoelzle et al. (2020). This classification scheme is based on mean and maximum catchment elevation, periods of typical low flow, snow onset and beginning of snowmelt and was already used by Stoelzle et al. (2020) to classify catchments outside the Swiss borders (e.g., German catchments). According to this classification scheme, the four Italian catchments (DOR, SOU, BCC and NBPV) are all categorized as snow dominated (S). The classification of BCC is also consistent with the one given in a previous study without considering the application of a formal classification scheme (Penna et al., 2016). Across the three considered streamflow regimes, a shift of the monthly hydrograph peak (computed using discharge data in the PoS) from winter to summer months is observed (Fig. 3): this flow peak shifting is a clear sign of the increasing predominance of snowmelt in the streamflow generation processes. Our data set includes NBPV, whose area is 42 % glacier covered and consequently exhibits a characteristic glacier-dominated streamflow regime with a monthly peak in late summer (Zuecco et al., 2019; Carturan, 2016).

NBPV has been classified as snow dominated following the Stoelzle et al. (2020) classification scheme. Nevertheless, its characteristics suggest it may belong to a fourth category of glacier-dominated catchments. Unfortunately, this category has not been considered by the aforementioned classification scheme, and the definition of the classifiers for a new category is outside the scope of this work. In this catchment, the effect of glacier melt on $F_{\text{yw}}^*$ cannot be neglected, and this was partially discussed by Ceperley et al. (2020). In our data set, also the Dischmabach (DIS) and the Vallon de Nant (VdN) catchments are 2 % and 3 % glacier covered, but we assume that the effect on $F_{\text{yw}}^*$ is negligible when compared with that of the seasonal snowpack.

## 3  Material and methods

### 3.1  Young water fraction estimation from seasonal cycles of stable water isotopes in precipitation and stream water: the "direct" input

Kirchner (2016a) designed the young water fraction as the proportion of the transit time distribution younger than a threshold age ($\tau_{\text{yw}}$). By assuming that the transit time distribution mathematical form is the regularized lower incomplete gamma function for all the study catchments, the theoretical young water fraction ($F_{\text{yw}}^{\text{T}}$) can be expressed as

$$F_{\text{yw}}^{\text{T}} = P\left(\tau < \tau_{\text{yw}}\right) = \int_{0}^{\tau_{\text{yw}}} \frac{\tau^{\alpha-1}}{\beta^{\alpha}\Gamma(\alpha)} e^{-\frac{\tau}{\beta}} d\tau, \tag{2}$$

where $\alpha$ and $\beta$ are the shape and scale factor, respectively.

By using thought experiments, Kirchner (2016a) has demonstrated that for a given shape factor $\alpha$ (ranging from 0.2 to 2) and across a wide range of scale factors $\beta$, the theoretical young water fraction can be accurately predicted by the amplitude ratios of seasonal sine curves fitted

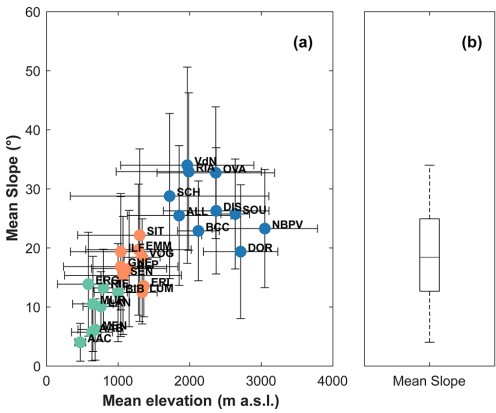
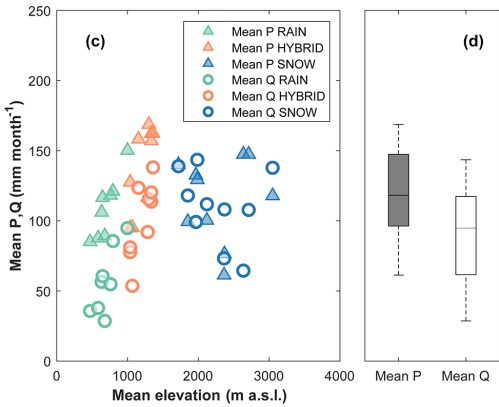

**Figure 2. (a)** Mean slope against mean elevation. Vertical bars represent the mean slope standard deviation, horizontal bars represent min–max elevation range, **(b)** boxplot of the mean slope values, **(c)** mean precipitation and discharge against elevation, and **(d)** boxplots of the mean precipitation and discharge values. Here and later: the boxplots show the median and the interquartile range (IQR), the whiskers are defined as the IQR multiplied by 1.5 and outliers are plotted with red "+" markers.

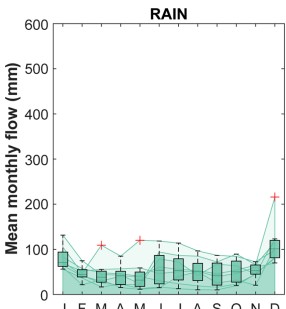
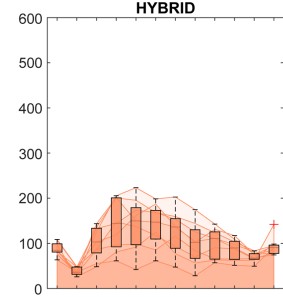
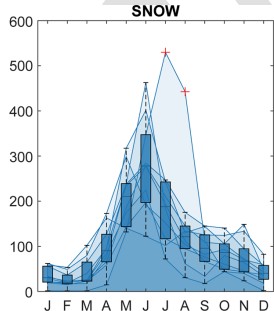

**Figure 3.** Boxplots of mean monthly flow for all the study catchments grouped according to their flow regime (rainfall dominated, hybrid, snow dominated). Colored areas represent the monthly flow of each study catchment belonging to the relative regime.

to stream water and precipitation isotope values by considering a $\tau_{yw}$ of 2–3 months. Operatively, we model seasonal isotope (e.g., $\delta^{18}O$) cycles in stream water and precipitation as reported in Eqs. (3a) and (3b):

$$\delta^{18}O_S(t) = A_S \sin(2\pi f t - \varphi_S) + k_S, \tag{3a}$$

$$\delta^{18}O_P(t) = A_P \sin(2\pi f t - \varphi_P) + k_P, \tag{3b}$$

where $\delta^{18}O$ (‰) is the isotopic composition of water sampled at the time $t$ (expressed in decimal years), $A$ (‰) is the amplitude of the seasonal isotope cycle, $\varphi$ (in radians, with $2\pi$ rad = 1 year) is the phase, $f$ (yr$^{-1}$) is the frequency and $k$ (‰) is the vertical offset of the seasonal isotope cycle. The subscript $S$ refers to stream water, while the subscript $P$ refers to precipitation. The sine wave is fitted to the isotopes measured in precipitation weighted according to the volume of precipitation, reducing the influence of low-precipitation periods and accounting for temporally aggregated rainfall samples (von Freyberg et al., 2018); the sine fit of stream water isotope measurements can be discharge-weighted or not, using the discharge measured at the moment of sampling as weights (von Freyberg et al., 2018). The sine

curves of Eqs. (3a) and (3b) are fitted on the isotope measurements using the iteratively re-weighted least squares (IRLS) method (for reducing the influence of outliers), which leads to estimates of $A$, $\varphi$ and $k$ parameters. A function for performing a sine fit using IRLS, based on the IRLS function made available by Kirchner and Knapp (2020), is available in the Supplement.

Accordingly, depending on the unweighted or the flow-weighted fit, an unweighted amplitude ($A_S$) or a flow-weighted amplitude ($A_S^*$) can be obtained, respectively. Such amplitudes can be used to calculate the time-weighted (Eq. 4a) or the flow-weighted (Eq. 4b) young water fractions ($F_{yw}$ or $F_{yw}^*$, respectively) via the "amplitude ratio approach":

$$F_{yw} = \frac{A_S}{A_P}, \tag{4a}$$

$$F_{yw}^* = \frac{A_S^*}{A_P}. \tag{4b}$$

Gallart et al. (2020a) highlighted the advantages of the flow-weighted analysis (generally yielding $A_S^*$ greater than $A_S$) to compensate for subsampled high-flow periods, which would

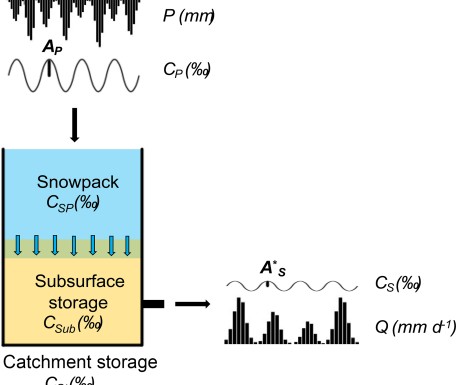

**Figure 4.** Schematic representation of the "direct input" approach for estimating $F_{yw}^*$. Light blue arrows indicate that meltwater coming from the snowpack preferentially infiltrates. The term $C$ refers to the isotopic composition. The subscript $P$ refers to precipitation, $S$ refers to stream, $SP$ refers to snowpack, $Sub$ refers to subsurface storage and $St$ refers to catchment storage.

otherwise lead to a young water fraction underestimate. Accordingly, in this work, we calculate the flow-weighted young water fractions for all the study catchments by applying Eq. (4b). We obtain the standard errors ($SE_{F_{yw}^*}$) of the estimated $F_{yw}^*$ using Eq. (5):

$$SE_{F_{yw}^*} = \frac{1}{A_P} \sqrt{SE_{A_S^*}^2 + \left(\frac{A_S^*}{A_P}\right)^2 \cdot SE_{A_P}^2}, \tag{5}$$

where $SE_{A_S^*}$ and $SE_{A_P}$ are the standard errors of the regression coefficients $A_S^*$ and $A_P$. The analytical choice of using the amplitude ($A_P$) fitted to precipitation isotopes, instead of the amplitude ($A_{Peq}$) fitted to the equivalent precipitation (i.e., rain plus snowmelt) isotopes, for estimating $F_{yw}^*$ implies that the snowpack (and/or the glacier) is considered as part of the catchment storage. Thus, the damped seasonal cycle observed in the stream is given by the mixing of precipitation with snowpack (and/or the glacier) and subsurface storage (the last two considered as a single entity), as illustrated in Fig. 4.

Since we have assumed that the transit time distribution belongs to the family of gamma distributions, we can determine the parameter $\alpha$ of such a distribution by solving the following implicit expression for $\alpha$ (Eq. 6):

$$\varphi_S^* - \varphi_P = \alpha \arctan\left(\sqrt{\left(A_S^*/A_P\right)^{-\frac{2}{\alpha}} - 1}\right). \tag{6}$$

We optimized to find the best solution of Eq. (6) using the best-fitting parameters ($\varphi_S^*$, $\varphi_P$, $A_S^*$, $A_P$) with $\alpha$ spanning a wide interval between 0.01 and 20. For the relevant math, the reader is referred to Kirchner (2016a). We estimate the uncertainty of $\alpha$ assuming all the fitted parameters having a Gaussian distribution with a standard deviation equal to the

regression error (Eqs. 3a and 3b). We generate 1000 random samples of such parameters, and we estimate the optimal $\alpha$ for each parameter set and then compute their standard deviation.

As in past studies (Gallart et al., 2020a; Lutz et al., 2018), we use the parameter $\alpha$ to estimate $\tau_{yw}$ with the following second-order polynomial fit (Kirchner, 2016a):

$$\frac{\tau_{yw}}{T} \approx 0.0949 + 0.1065\alpha - 0.0126\alpha^2, \tag{7}$$

where $T$ is the period of the tracer cycle (for a seasonal cycle, $T = 1$ year). We estimate the uncertainty of $\tau_{yw}$ by calculating the standard deviation of the threshold ages computed using the 1000 optimal $\alpha$ values previously obtained. A code for estimating $\alpha$ and $\tau_{yw}$ with their uncertainties is made available in the Supplement.

The comparison of $F_{yw}^*$ among different catchments is potentially subject to a bias given by possibly different threshold ages ranging between 2 and 3 months. Accordingly, we couple each $F_{yw}^*$ with the corresponding $\tau_{yw}$ to illustrate what the term "young" means for each study catchment.

## 3.2 Snow cover persistence quantified through the mean fractional snow cover area ($F_{SCA}$)

In this paper, we quantify the snowpack persistence by calculating the mean fractional snow cover area ($F_{SCA}$). It is calculated for each catchment over the period 1 October 2017–30 September 2021 (hereafter defined as PoC, i.e., period of calculation) by using the collection of Sentinel-2 L2A satellite images available in Google Earth Engine (Gorelick et al., 2017). Temporally, this relatively recent satellite has increased the visitation frequency to a subweekly temporal resolution and increased the spatial resolution to 20 m for snow cover (Gascoin et al., 2019). High temporal resolution makes Sentinel-2 images preferable to Landsat images, which are available only once every 16 d and whose total number is often further reduced because of cloudiness (Hofmeister et al., 2022). The PoC generally differs from the PoS for the 27 study catchments. This is because Sentinel-2 L2A satellite images are not available before March 2017. For each image available in the PoC, we calculate the normalized difference snow index (NDSI) as suggested in the work of Dozier (1989):

$$NDSI = \frac{r_{green} - r_{SWIR}}{r_{green} + r_{SWIR}}, \tag{8}$$

where $r_{green}$ is the reflectance in the green band (Sentinel-2 band 3) and $r_{SWIR}$ is the shortwave infrared reflectance band (Sentinel-2 band 11). We classify as snowy pixels those with an NDSI value > 0.4 (Dozier, 1989). Based on the pixel-by-pixel snow classification, we compute the snapshot fractional snow cover area ($f_{SCA}$) according to Di Marco et al. (2020) and Hofmeister et al. (2022):

$$f_{SCA} = \frac{N_{snow}}{N_{tot} - N_{clouds}}, \tag{9}$$

where $N_{\text{snow}}$ is the number of snow cover pixels according to the applied NDSI threshold method, $N_{\text{tot}}$ is the total number of pixels within the catchment area and $N_{\text{clouds}}$ is the number of pixels classified as clouds and water bodies (Hofmeister et al., 2022). We identify the cloudy pixels directly using the Sentinel-2 band "Scene Classification Map". We operatively calculate $N_{\text{snow}}$, $N_{\text{tot}}$ and $N_{\text{clouds}}$ using a Google Earth Engine code.

By using this procedure for calculating $f_{\text{SCA}}$, we sometimes obtain $f_{\text{SCA}} > 1$. The NDSI threshold method is generally able to distinguish between snow and no-snow pixels (Aalstad et al., 2020). Accordingly, clouds and snow have similar reflectance in the green band, but clouds highly reflect in the shortwave infrared band, while snow reflectance is low in this band. Thus, the $N_{\text{snow}}$ estimation is generally accurate. On the other hand, it is necessary to disregard the false positive pixels deriving from cloud detection (i.e., snow classified as clouds). If $f_{\text{SCA}} > 1$, we calculate $f_{\text{SCA}}$ as $N_{\text{snow}}/N_{\text{tot}}$, since this is the only heuristic solution that guarantees no overestimation. Moreover, by looking at sample Sentinel-2 images during the summer periods for all the catchments, we impose $f_{\text{SCA}} = 0$ during July and August, since when $f_{\text{SCA}} \neq 0$, this usually results from clouds falsely identified as snow: imposing $f_{\text{SCA}} = 0$ clearly leads to fewer errors (only missing occasional summer snowfall events of very shallow depth) than falsely accounting for (far more) frequent clouds. The NBPV catchment is an exception: we do not impose $f_{\text{SCA}} = 0$ during July and August, since it generally has snow over the glacier also during summer. Finally, we compute the mean fractional snow cover area ($F_{\text{SCA}}$) for each catchment by averaging all $f_{\text{SCA}}$ values available for all snow images in the PoC, without interpolation between the time steps.

## 3.3 Fraction of Quaternary deposits, low-flow duration and the groundwater contribution to the stream

Similarly to Arnoux et al. (2021), we calculate the portion of the catchment area occupied by Quaternary deposits ($A_{\text{qd}}$) (available from government geological data sets) with respect to the total catchment area ($A$). Thus, we calculate the fraction of Quaternary deposits ($F_{\text{qd}}$) as reported by Eq. (10):

$$F_{\text{qd}} = \frac{A_{\text{qd}}}{A}. \tag{10}$$

Additionally, we use the same winter flow index (WFI) as Arnoux et al. (2021), as indicated by Eq. (11):

$$\text{WFI} = \frac{Q_{\text{NM7}}}{Q_{\text{mean}}}, \tag{11}$$

where $Q_{\text{NM7}}$ is the minimum discharge over 7 consecutive days during the winter period (from November to June) and $Q_{\text{mean}}$ is the mean annual discharge. We calculate it for the 27 study catchments during the PoS. To relate WFI to low

flow, we apply the recent baseflow separation technique described by Duncan (2019) to the discharge time series of the 27 study catchments (within the PoS indicated in Table 1). In short, this method comprises a single backward pass through the data to fit an exponential master baseflow recession curve (Eq. 12.1), followed by the single forward pass (Eqs. 12b and 12c) of the Lyne and Hollick (1979) algorithm. This allows the smoothing of CE3 the connection between segments of the master recession by simulating a gradual groundwater recharge during the runoff event (Duncan, 2019):

$$M(t_{i-1}) = \frac{M(t_i) - c}{k} + c \tag{12a}$$

$$Q_{\text{q}}(t_i) = k Q_{\text{q}}(t_{i-1}) + (M(t_i) - M(t_{i-1})) \frac{1+k}{2} \tag{12b}$$

$$Q_{\text{bf}}(t_i) = M(t_i) - Q_{\text{q}}(t_i), \tag{12c}$$

where $M(t_i)$, $Q_{\text{q}}(t_i)$ and $Q_{\text{bf}}(t_i)$ are the master recession value, the quick recession flow and the baseflow at time $t_i$, respectively. In this study, we consider daily time steps (i.e., $t_i - t_{i-1} = 1$ d). This method has two parameters: $k$ is the recession constant, $c$ is a constant flow added to the exponential decay component. We set the recession constant $k = 0.925$ (Nathan and McMahon, 1990): we add no constant flow to the exponential decay (i.e., in terms of the method by Duncan (2019), $c = 0$). A code with the implementation of the Duncan (2019) baseflow filter has been made available in the supplementary material.

To express the catchment storage contribution to streamflow in a form that is directly comparable to the $F_{\text{yw}}^*$, we define the baseflow fraction ($F_{\text{bf}}$) as reported in Eq. (13):

$$F_{\text{bf}} = \frac{1}{n} \sum_{i=1}^{n} \frac{Q_{\text{bf}}(t_i)}{Q(t_i)}, \tag{13}$$

where $Q_{\text{bf}}(t_i)$ is the baseflow (mm d$^{-1}$) at the time $t_i$ (obtained as indicated by Eq. 12c) and $Q(t_i)$ is the discharge (mm d$^{-1}$) at the time $t_i$. We tested the uncertainty of $k$ by drawing random samples (10 000) from a normal distribution spanning Nathan and McMahon's (1990) recommended range for $k$ from 0.90 to 0.95, with a mean of 0.925 and a standard deviation equal to 25 % of the range. Thereby, we obtain 10 000 values of $F_{\text{bf}}$ for each catchment of which we compute the standard deviation.

As introduced in Sect. 1, $F_{\text{yw}}^*$ can be low if the snapshot young water fraction $F_{\text{yw}}(t_i)$ is very low for many time steps. If we consider the discharge ($Q$) as a proxy for the catchment wetness, we can reliably assert that $F_{\text{yw}}(t_i)$ is low for low $Q(t_i)$. Thus, another important variable is the duration of the low-flow period. In this study, we define a low-flow period ($T_{\text{Low}}$) as follows:

$$T_{\text{Low}} = \forall t_i : \frac{Q_{\text{bf}}(t_i)}{Q(t_i)} \geq 0.85. \tag{14}$$

Thus, a low-flow period is defined here as a period when 85 % of the total flow is composed of baseflow (i.e., base-

flow dominated). Accordingly, we define the low-flow duration (LFD) as the proportion of the time steps (e.g., days) in the PoS that can be considered as a low-flow period according to Eq. (14).

## 4   Results and discussion – towards a harmonious and exhaustive framework of the hydrological processes that drive the young water fraction variations with elevation

We present and discuss hereafter the $F_{yw}^*$ and $\tau_{yw}$ estimates (Sect. 4.1) and the identified relations between $F_{yw}^*$ and the studied explanatory variables (Sect. 4.2–4.5), followed by the perceptual model that describes the main processes driving the $F_{yw}^*$ variations with mean catchment elevation and that harmonizes our results with previous studies (Sect. 4.6).

### 4.1   Young water fractions ($F_{yw}^*$) and corresponding threshold ages

Assembling $F_{yw}^*$ values determined by different authors who very likely used different source codes could possibly result in a bias. Indeed, differences in $F_{yw}^*$ among catchments could be driven by the different methods rather than the physical factors. Therefore, the same approach must be applied to all the study catchments to remove the bias introduced by the estimation method of $F_{yw}^*$. For all the study catchments, sinusoidal cycles were fitted to both precipitation and stream water $\delta^{18}$O data by using the IRLS regression (results for six representative study catchments in Fig. 5; complete results in Fig. S2). We estimate $F_{yw}^*$ via Eq. (4b) by using the best-fitting amplitudes of seasonal cycles. The best-fit amplitudes ($A_P$, $A_S^*$), phases ($\varphi_P$, $\varphi_S^*$) and corresponding standard errors are reported in Table 2.

The $F_{yw}^*$ values achieved in this study are consistent with published $F_{yw}^*$ values of Ceperley et al. (2020) and with direct-input $F_{yw}^*$ of von Freyberg et al. (2018). Accordingly, a two-sample Kolmogorov–Smirnov test accepts the null hypothesis that the new and past $F_{yw}^*$ estimates are from the same continuous distribution at the 5 % significance level. The $F_{yw}^*$ estimates are reported in Fig. 6a against the mean catchment elevation and are also listed in Table 2. $F_{yw}^*$ increases with mean catchment elevation until 1500 m a.s.l., which corresponds to the elevation above which all catchments are snow dominated (with NBPV detected as an outlier as will be discussed in Sect. 4.5). This pattern is also reflected by the median $F_{yw}^*$ within each flow regime: the median $F_{yw}^*$ is 0.13 for rainfall-dominated catchments, 0.29 for hybrid catchments and 0.12 for snow-dominated catchments. Such results are consistent with previous studies that have shown the tendency toward low $F_{yw}^*$ in mountainous catchments (Ceperley et al., 2020; von Freyberg et al., 2018; Lutz et al., 2018; Jasechko et al., 2016).

Even though we remove the bias introduced by the estimation method of $F_{yw}^*$, the application of Eq. (4b) implicitly introduces another bias if we want to use $F_{yw}^*$ for intercomparison purposes (which is the goal of this work). By computing the amplitude ratio, we estimate $F_{yw}^T$ (Eq. 2), without defining a corresponding $\tau_{yw}$ (which can be estimated via Eq. 7). Therefore, part of the scatter of $F_{yw}^*$ between catchments might be because of different $\tau_{yw}$ rather than physical factors, also if the $\tau_{yw}$ is expected to vary modestly between 2 and 3 months (Kirchner, 2016a). Accordingly, past studies estimated $F_{yw}^*$ using the amplitude ratio approach without information about the corresponding $\tau_{yw}$ (Stockinger et al., 2019; von Freyberg et al., 2018; Jasechko et al., 2016). Nevertheless, we estimate $\tau_{yw}$ and $\alpha$ for each study catchment: the resulting estimates are reported in Fig. 6b and are listed in Table 2. Our estimates of the $\alpha$ parameter ($0.19 \leq \alpha \leq 2.1$) are consistent with the shape factor range ($0.2 \leq \alpha \leq 2$) investigated by Kirchner (2016a). Consequently, the $\tau_{yw}$ obtained by applying Eq. (7) falls between 1.38 and 3.16 months. As expected, $\tau_{yw}$ varies in a narrow range consistent with the explanation provided by Kirchner (2016a). However, in order to have a fully coherent metric for all the catchments, the optimal procedure would be to set $\tau_{yw}$ and to calculate the young water fraction corresponding to this $\tau_{yw}$. Nevertheless, establishing a constant $\tau_{yw}$ for all the catchments could be a tricky choice. Indeed, by setting a $\tau_{yw}$ higher than that obtained via Eq. (7), we are improperly using the TTD to estimate the young water fraction. From this reasoning, the only solution is to set $\tau_{yw}$ equal to the overall minimum $\tau_{yw}$. This choice ensures that the TTD is used properly to estimate the young water fraction in all the sites. What we could expect is that all the young water fractions would be lower with respect to those obtained with the amplitude ratio approach. However, changes in young water fraction depending on changes in $\tau_{yw}$ are unintuitive, since they vary with the TTD shape. Accordingly, a constant reduction of $\tau_{yw}$ would change the area under the transit time pdf differently based on the $\alpha$ value. Thus, the overall effect of the reduced $\tau_{yw}$ upon the young water fraction remains a priori unknown.

Since in this study we are using the amplitude ratio approach, our $F_{yw}^*$ estimates refer to the proportion of runoff younger than an inconsistent threshold age. This variation by catchment (albeit limited) is the main limitation of this work.

### 4.2   The role of Quaternary deposits

In line with the results of Arnoux et al. (2021), we find a negative statistically significant correlation between $F_{yw}^*$ and WFI ($\rho_{Spearman} = -0.5$, $p$ value $< 0.01$; see Fig. S6), suggesting (unsurprisingly) that more groundwater contribution to streamflow increases the water age. WFI and $F_{qd}$ values for all the study catchments are reported in Table 2. To analyze the relationship of $F_{yw}^*$ with Quaternary deposits, we exclude the SOU and BCC catchments, since they show $F_{qd} = 0$ (see Table 2). The inclusion of catchments with

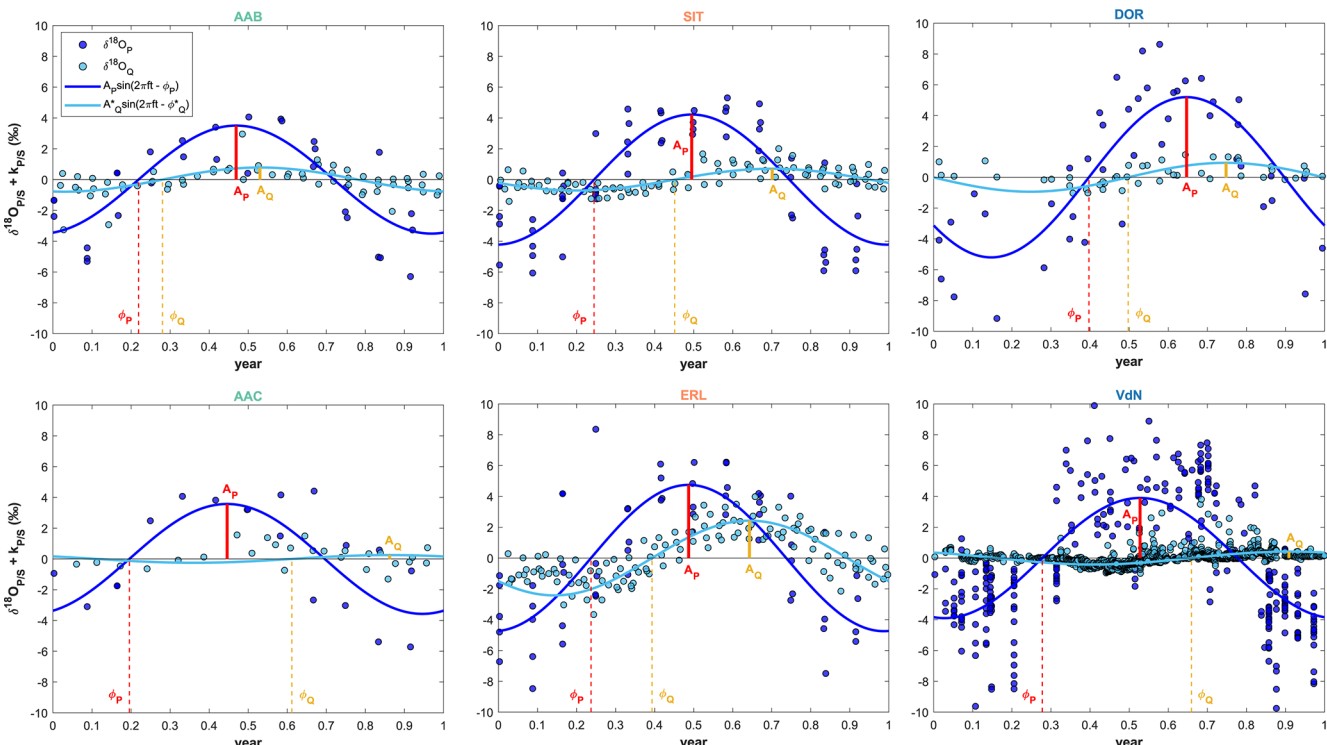

**Figure 5.** Sinusoidal cycles of both precipitation and streamflow fitted to the $\delta^{18}$O data (using the IRLS method) for six representative study catchments. Amplitudes (‰) and phases (year) are indicated in the figure. Please note that both $\delta^{18}$O data and sinusoidal cycles of precipitation and stream water are vertically shifted of $k_P$ and $k_S$, respectively.

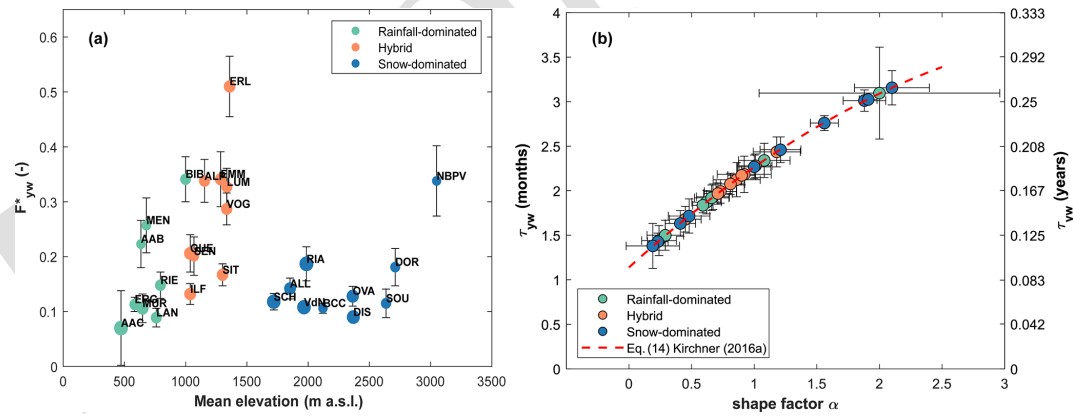

**Figure 6. (a)** $F_{yw}^*$ as function of mean catchment elevation. Points dimension is proportional to CE4 $\tau_{yw}$ **(b)**, obtained with Eq. (14) of Kirchner (2016a), as function of the shape factor $\alpha$.

$F_{qd} = 0$ would bias the analysis, since an absent parameter cannot modulate the share of groundwater and thus the young water fraction in the stream.

By focusing first only on the snow-dominated catchments, a linear fit on the data returns a negative slope of $-0.36$ ($R^2 = 0.52$), indicating a reduction of $F_{yw}^*$ with increasing $F_{qd}$ (Fig. 7a). Moreover, we find a Spearman rank correlation coefficient of $-0.6$ with a $p$ value of 0.13, meaning a negative but not statistically significant correlation between $F_{yw}^*$

and $F_{qd}$. This result can be explained by considering several factors. First, water storage in Quaternary deposits is not the only groundwater storage contribution to the stream in such environments: additional storage is provided by the bedrock fractures (Gleeson et al., 2014; Jasechko et al., 2016; Martin et al., 2021), possibly caused by rock stress and high erosion rates and by the bedrock geology, which has influence on groundwater retention capacity (Hayashi, 2020). Second, the area covered by Quaternary deposits could be an insuffi-

**Table 2.** Summary table with all the relevant quantities estimated for the 27 study catchments. `TS1`

| ID (Reg.) | $A_S^* \pm SE$ (‰) | $A_P \pm SE$ (‰) | $\phi_S^* \pm SE$ (rad) | $\phi_P \pm SE$ (rad) | $F_{yw}^* \pm SE$ (–) | $\alpha \pm SD$ (–) | $\tau_{yw} \pm SD$ (yr) | LFD (d d$^{-1}$) | $F_{bf}$ (–) | $F_{SCA}$ (–) | WFI (–) |
|---|---|---|---|---|---|---|---|---|---|---|---|
| AAB (R) | $0.78 \pm 0.12$ | $3.51 \pm 0.44$ | $1.76 \pm 0.17$ | $1.38 \pm 0.13$ | $0.22 \pm 0.04$ | $0.24 \pm 0.14$ | $0.12 \pm 0.014$ | 0.44 | 0.71 | 0.14 | 0.14 |
| AAC (R) | $0.25 \pm 0.24$ | $3.58 \pm 0.61$ | $3.84 \pm 0.74$ | $1.23 \pm 0.18$ | $0.07 \pm 0.07$ | $2.00 \pm 0.96$ | $0.258 \pm 0.043$ | 0.5 | 0.73 | 0.11 | 0.15 |
| ALL (S) | $0.82 \pm 0.09$ | $5.77 \pm 0.43$ | $3.21 \pm 0.13$ | $1.78 \pm 0.08$ | $0.14 \pm 0.02$ | $1.00 \pm 0.13$ | $0.189 \pm 0.011$ | 0.62 | 0.84 | 0.45 | 0.24 |
| ALP (H) | $1.25 \pm 0.09$ | $3.7 \pm 0.35$ | $2.56 \pm 0.07$ | $1.58 \pm 0.1$ | $0.34 \pm 0.04$ | $0.73 \pm 0.13$ | $0.166 \pm 0.011$ | 0.41 | 0.69 | 0.28 | 0.06 |
| BCC (S) | $0.51 \pm 0.03$ | $4.84 \pm 0.25$ | $2.64 \pm 0.11$ | $2.28 \pm 0.07$ | $0.11 \pm 0.01$ | $0.23 \pm 0.08$ | $0.119 \pm 0.008$ | 0.71 | 0.87 | 0.42 | 0.23 |
| BIB (R) | $1.27 \pm 0.09$ | $3.73 \pm 0.35$ | $2.45 \pm 0.07$ | $1.58 \pm 0.1$ | $0.34 \pm 0.04$ | $0.62 \pm 0.12$ | $0.156 \pm 0.011$ | 0.39 | 0.65 | 0.23 | 0.06 |
| DIS (S) | $0.62 \pm 0.04$ | $6.86 \pm 0.43$ | $3.89 \pm 0.09$ | $1.77 \pm 0.07$ | $0.09 \pm 0.01$ | $1.56 \pm 0.11$ | $0.23 \pm 0.007$ | 0.74 | 0.89 | 0.58 | 0.19 |
| DOR (S) | $0.94 \pm 0.15$ | $5.2 \pm 0.53$ | $3.13 \pm 0.15$ | $2.5 \pm 0.1$ | $0.18 \pm 0.03$ | $0.41 \pm 0.12$ | $0.136 \pm 0.012$ | 0.65 | 0.85 | 0.61 | 0.06 |
| EMM (H) | $1.2 \pm 0.12$ | $3.53 \pm 0.38$ | $2.74 \pm 0.1$ | $1.64 \pm 0.12$ | $0.34 \pm 0.05$ | $0.86 \pm 0.19$ | $0.177 \pm 0.016$ | 0.3 | 0.6 | 0.28 | 0.01 |
| ERG (R) | $0.42 \pm 0.03$ | $3.71 \pm 0.29$ | $2.44 \pm 0.07$ | $1.53 \pm 0.09$ | $0.11 \pm 0.01$ | $0.59 \pm 0.08$ | $0.153 \pm 0.007$ | 0.51 | 0.75 | 0.07 | 0.05 |
| ERL (H) | $2.42 \pm 0.11$ | $4.75 \pm 0.46$ | $2.47 \pm 0.05$ | $1.49 \pm 0.1$ | $0.51 \pm 0.05$ | $0.92 \pm 0.22$ | $0.182 \pm 0.017$ | 0.21 | 0.5 | 0.3 | 0.01 |
| GUE (H) | $0.72 \pm 0.09$ | $3.51 \pm 0.4$ | $2.98 \pm 0.1$ | $1.44 \pm 0.13$ | $0.21 \pm 0.03$ | $1.18 \pm 0.19$ | $0.203 \pm 0.014$ | 0.44 | 0.71 | 0.23 | 0.07 |
| ILF (H) | $0.58 \pm 0.07$ | $4.36 \pm 0.36$ | $3 \pm 0.12$ | $1.55 \pm 0.09$ | $0.13 \pm 0.02$ | $1.01 \pm 0.13$ | $0.19 \pm 0.011$ | 0.53 | 0.77 | 0.22 | 0.12 |
| LAN (R) | $0.34 \pm 0.06$ | $3.79 \pm 0.36$ | $2.6 \pm 0.16$ | $1.57 \pm 0.1$ | $0.09 \pm 0.02$ | $0.67 \pm 0.13$ | $0.161 \pm 0.012$ | 0.69 | 0.87 | 0.1 | 0.4 |
| LUM (H) | $1.62 \pm 0.09$ | $4.97 \pm 0.44$ | $2.48 \pm 0.06$ | $1.52 \pm 0.09$ | $0.33 \pm 0.03$ | $0.71 \pm 0.11$ | $0.164 \pm 0.01$ | 0.37 | 0.66 | 0.33 | 0.08 |
| MEN (R) | $0.77 \pm 0.12$ | $3 \pm 0.36$ | $1.84 \pm 0.17$ | $1.39 \pm 0.14$ | $0.26 \pm 0.05$ | $0.29 \pm 0.15$ | $0.125 \pm 0.014$ | 0.59 | 0.79 | 0.09 | 0.18 |
| MUR (R) | $0.37 \pm 0.08$ | $3.51 \pm 0.36$ | $2.96 \pm 0.21$ | $1.4 \pm 0.11$ | $0.11 \pm 0.03$ | $1.08 \pm 0.21$ | $0.195 \pm 0.016$ | 0.53 | 0.77 | 0.14 | 0.19 |
| NBPV (S) | $1.52 \pm 0.09$ | $4.48 \pm 0.81$ | $3.1 \pm 0.13$ | $2.81 \pm 0.32$ | $0.34 \pm 0.06$ | $0.19 \pm 0.21$ | $0.115 \pm 0.021$ | 0.35 | 0.7 | 0.76 | 0 |
| OVA (S) | $0.9 \pm 0.10$ | $7 \pm 0.61$ | $3.37 \pm 0.14$ | $1.68 \pm 0.09$ | $0.13 \pm 0.02$ | $1.21 \pm 0.16$ | $0.205 \pm 0.012$ | 0.66 | 0.85 | 0.54 | 0.21 |
| RIA (S) | $0.91 \pm 0.12$ | $4.86 \pm 0.52$ | $3.54 \pm 0.14$ | $1.22 \pm 0.12$ | $0.19 \pm 0.03$ | $2.10 \pm 0.3$ | $0.263 \pm 0.016$ | 0.63 | 0.84 | 0.47 | 0.11 |
| RIE (R) | $0.59 \pm 0.06$ | $3.96 \pm 0.48$ | $2.34 \pm 0.12$ | $1.34 \pm 0.14$ | $0.15 \pm 0.02$ | $0.66 \pm 0.14$ | $0.16 \pm 0.012$ | 0.43 | 0.68 | 0.18 | 0.07 |
| SCH (S) | $0.6 \pm 0.06$ | $5.13 \pm 0.4$ | $4.08 \pm 0.11$ | $1.74 \pm 0.09$ | $0.12 \pm 0.02$ | $1.88 \pm 0.17$ | $0.251 \pm 0.01$ | 0.61 | 0.85 | 0.47 | 0.2 |
| SEN (H) | $0.83 \pm 0.12$ | $4.1 \pm 0.41$ | $2.23 \pm 0.14$ | $1.55 \pm 0.12$ | $0.2 \pm 0.04$ | $0.45 \pm 0.13$ | $0.14 \pm 0.012$ | 0.49 | 0.75 | 0.22 | 0.24 |
| SIT (H) | $0.71 \pm 0.07$ | $4.22 \pm 0.31$ | $2.83 \pm 0.1$ | $1.54 \pm 0.08$ | $0.17 \pm 0.02$ | $0.9 \pm 0.11$ | $0.181 \pm 0.009$ | 0.35 | 0.68 | 0.28 | 0.07 |
| SOU (S) | $0.6 \pm 0.12$ | $5.2 \pm 0.53$ | $3.24 \pm 0.23$ | $2.5 \pm 0.1$ | $0.11 \pm 0.03$ | $0.48 \pm 0.17$ | $0.143 \pm 0.016$ | 0.56 | 0.82 | 0.53 | 0.01 |
| VdN (S) | $0.42 \pm 0.01$ | $3.89 \pm 0.19$ | $4.14 \pm 0.03$ | $1.74 \pm 0.05$ | $0.11 \pm 0.01$ | $1.91 \pm 0.07$ | $0.252 \pm 0.004$ | 0.62 | 0.84 | 0.49 | 0.04 |
| VOG (H) | $1.38 \pm 0.08$ | $4.81 \pm 0.42$ | $2.6 \pm 0.06$ | $1.5 \pm 0.09$ | $0.29 \pm 0.03$ | $0.81 \pm 0.11$ | $0.173 \pm 0.01$ | 0.39 | 0.66 | 0.27 | 0.07 |

cient proxy of the groundwater storage potential: the knowledge of the thickness of these deposits (i.e., their volume) and the bedrock topography are crucial factors for controlling groundwater storage (Arnoux et al., 2021; Hayashi, 2020), but corresponding data are not available to date.

$F_{yw}^*$ values of the hybrid catchments reveal a weak positive correlation with Quaternary deposits ($\rho_{Spearman} = 0.13$, $p$ value $= 0.74$), while for rainfall-dominated catchments they show a negative correlation ($\rho_{Spearman} = -0.52$, $p$ value $= 0.2$); however, both correlations are not statistically significant. These weak correlations suggest that $F_{qd}$ represents only a limited part of the catchment geology responsible for groundwater flow and that it can only be considered as a first-order measure of geological groundwater storage.

We furthermore observe that $F_{qd}$ decreases with mean catchment elevation in our data set (Fig. 7b), revealing a negative statistically significant correlation ($\rho_{Spearman} = -0.5$, $p$ value $< 0.01$). This negative correlation reflects the fact that $F_{qd}$ decreases when the mean slope increases (Arnoux et al., 2021).

To conclude, we stress that more catchments and more geological information would be required to statistically validate these observations about the role of the groundwater storage potential for explaining young water fraction variations.

## 4.3 The role of groundwater flow (baseflow) in $F_{yw}^*$

### 4.3.1 Baseflow under different hydroclimatic regimes

The baseflow time series resulting from the baseflow separation of Duncan (2019) for six representative study catchments (two of each regime) are reported in Fig. 8 (complete results in Fig. S3). This figure shows the effect of groundwater recharge from rain and snowmelt through the "smoothed" baseflow proposed by Duncan (2019). This "smoothing" simulates a delayed storage contribution to the stream following the recharge phase during an input event. This recharge phase promotes the system wetness, thus favoring increasing quick flow. The increasing quick flow during events also leads to an increase of $F_{yw}(t_i)$, as found previously (von Freyberg et al., 2018). However, the relative amount of baseflow remains high during events: the mean baseflow fraction during the high-flow period is 0.49 and 0.52 for hybrid and rainfall-dominated catchments, while it is 0.63 for snow-dominated catchments. In agreement with worldwide stable-isotope-based hydrograph separation results (Jasechko, 2019), this outcome underlines the mobilization of stored water (i.e., old water) during rainfall and snowmelt events, and this process seems to be particularly relevant in high-elevation catchments.

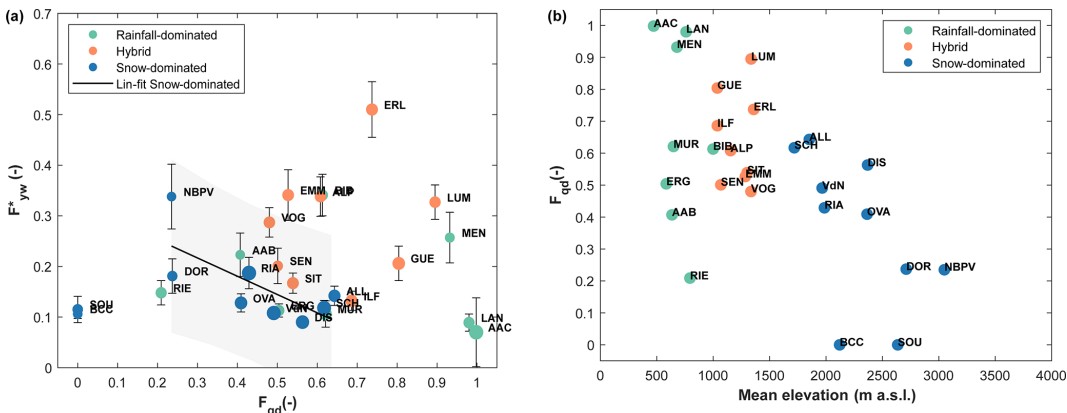

**Figure 7. (a)** Young water fraction against fraction of Quaternary deposits. Points dimension is proportional to $\tau_{yw}$. **(b)** Fraction of Quaternary deposits against mean catchments elevation.

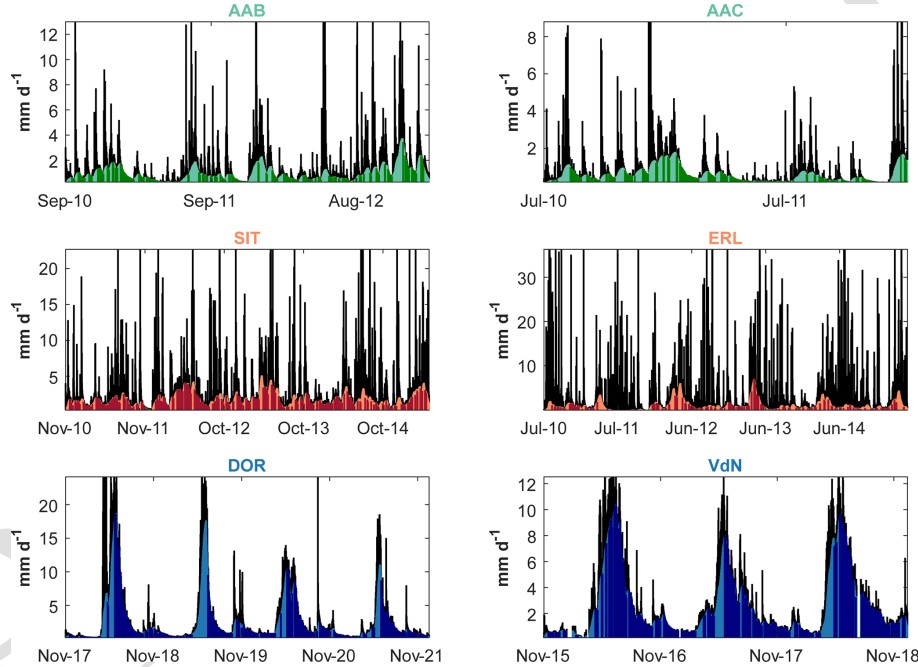

**Figure 8.** Baseflow separation for six representative study catchments using the Duncan (2019) filter. The black area represents the daily discharge, while the colored area represents the estimated daily baseflow. The darker color represents a time step in which at least 85 % of the daily discharge is composed by baseflow.

Accordingly, in snow-dominated systems, the snowmelt largely transits through the groundwater store (Hayashi, 2020; Cochand et al., 2019; Du et al., 2019; Flerchinger et al., 1992; Martinec, 1975), as schematized in Fig. 4, and the very high baseflow in high mountain catchments during summer is a direct sign of meltwater infiltration and percolation to groundwater that pushes old snowmelt (the main groundwater storage component) out to the stream network, as also found by Martinec (1975). This is also supported by the fact that groundwater, in such catchments, often has the isotopic

signature of snowmelt (Michelon et al., 2023; Pavlovskii et al., 2018).

When examining the overall flow (and not only at the high-flow periods), $F_{bf}$ is generally lower for hybrid catchments (mean of $F_{bf} = 0.67$) than for rainfall-dominated (mean of $F_{bf} = 0.74$) and snow-dominated catchments (mean of $F_{bf} = 0.83$). The values of $F_{bf}$ for all the study sites are reported in Table 2. In the BCC catchment, the $F_{bf}$ (0.87) is consistent with the previous findings of Penna et al. (2016) who found, using stable water isotopes, that on average between 80 % and 98 % of the discharge in BCC is composed of pre-event

water (assumed to represent groundwater). On average, the $F_{bf}$ computed over the entire PoS is higher than that computed during the high-flow periods. This result suggests, unsurprisingly, that the largest percentage of base flow is released during low-flow periods. Accordingly, the variations of $F_{bf}$ with elevation among different catchments (Fig. 9b) can be explained considering the changes in low-flow duration (LFD) with elevation, as will be discussed in Sect. 4.4.

Baseflow filters were already applied in previous studies, and their results were correlated with $F_{yw}^*$. For example, von Freyberg et al. (2018) found a strong positive correlation ($\rho_{Spearman} = 0.73$, $p$ value $< 0.001$) between $F_{yw}^*$ and the quick-flow index (QFI), calculated as the mean ratio between $(Q - Q_{bf})$ and $Q$, where $Q$ is the daily discharge and $Q_{bf}$ is the daily baseflow calculated in their paper with the Lyne and Hollick (1979) baseflow filter. By relating the $F_{bf}$ to $F_{yw}^*$, we have found a strong negative correlation ($\rho_{Spearman} = -0.73$, $p$ value $< 0.001$), as shown in Fig. 9a, consistent with the results of von Freyberg et al. (2018).

In snow-free systems, $F_{yw}^*$ is by definition related to $F_{bf}$: baseflow is composed of groundwater and groundwater is the dominant source of old water in such systems (in absence of large lakes). In snow-influenced systems, through the "direct input" approach for estimating $F_{yw}^*$, we consider the snowpack (i.e., a temporarily old water storage) as part of the catchment storage. However, the share of snowmelt (with age $> 3$ months) that flows off quickly as surface or fast subsurface runoff will not show up in $F_{bf}$. In other words, $F_{bf}$ is not able to take into account all the snowmelt but only the part of meltwater that infiltrates and recharges the groundwater storage, which is a large portion of the overall snowmelt.

### 4.3.2 The complementarity between the fraction of baseflow ($F_{bf}$) and the young water fraction ($F_{yw}^*$)

A by-product of this work is that the $F_{bf}$, estimated with the Duncan (2019) baseflow filter, is roughly the complementary term of $F_{yw}^*$ (Fig. 9a and b), which is an important result for catchments where isotope measurements are missing. In such catchments, $F_{yw}^*$ could potentially be estimated without the application of the amplitude ratio approach as follows: CE5

$$F_{yw}^* \simeq 1 - F_{bf}. \tag{15}$$

Some of our case studies show considerable "residuals" of $1 - (F_{bf} + F_{yw}^*)$ (Fig. 9b). This is partially due to the uncertainty of the parameters used for estimating $F_{bf}$. In this regard, Duncan (2019) suggests some calibration guidelines to obtain optimal parameters set for baseflow estimation per catchment. In this work, we did not use the calibration guidelines, but we simply used the recession parameter proposed by Nathan and McMahon (1990) in order to achieve factual and reproducible results. In addition, the estimation of baseflow during an event is generally less rigorous than during the recession phase (Duncan, 2019), affecting the $F_{bf}$ estimation. Moreover, $F_{yw}^*$ values are influenced by the sampling

rate: the higher the frequency of sampling is, the higher the young water fraction is (Gallart et al., 2020a; Stockinger et al., 2016). Thus, the young water fraction calculated with the amplitude ratio approach generally underestimates the "theoretical" young water fraction, and we simply compensate by computing the flow-weighted young water fraction ($F_{yw}^*$). In hybrid and snow-dominated catchments, these "residuals" can also be explained by considering that the $F_{bf}$ does not include surface runoff or fast lateral subsurface flow of meltwater, likely older than the estimated threshold ages, following a snowmelt event. On the other hand, these residuals might also be related to the non-linear recession behavior of catchments, which was shown by Santos et al. (2018) to dominate Swiss low-elevation (i.e., rain dominated) catchments, when the exponential recession assumption of the baseflow filter necessarily leads to less reliable results (Duncan, 2019).

### 4.4 Low-flow duration (LFD) and $F_{yw}^*$

The values of LFD for all the study sites are reported in Table 2. Specifically, LFD is lower for hybrid catchments (median of LFD $= 0.39$), and it is increasingly higher for rainfall (median of LFD $= 0.50$) and snow-dominated catchments (median of LFD $= 0.62$). In hybrid catchments, the presence of rain and snowmelt events spanning large parts of the year and the relatively low evapotranspiration (compared to rainfall-dominated catchments) due to reduced temperatures (Goulden et al., 2012) dramatically reduces the duration of low-flow periods, and this is also visible from the recurring discharge peaks (Fig. 8). In low-lying, rain-dominated catchments, evapotranspiration and precipitation are respectively higher and lower than in hybrid catchments, leading to longer low-flow periods (usually during summer and autumn). Under current climate and according to our data set, in snow-dominated catchments, we observe longer winter low-flow periods (streamflow decreasing below 0.5 to $1\,\mathrm{mm\,d^{-1}}$ for the highest locations; see Fig. S7) on an annual scale than in hybrid catchments. To gain additional insights into the high LFD in snow-dominated catchments and the low LFD in hybrid catchments, it is necessary to further consider the role of snowpack persistence, discussed in the following section. The variations of LFD with elevation are shown in Fig. 11b.

Low-flow periods are typically baseflow dominated (or old water dominated). Accordingly, as anticipated in Sect. 4.3, the variation of $F_{bf}$ between catchments reflects the proportion of the low-flow duration during the PoS. We observe that the higher the LFD is, the higher the $F_{bf}$ is: in fact, they are strongly positively correlated ($\rho_{Spearman} = 0.97$, $p$ value $< 0.001$) as shown in Fig. 10. The negative correlation between LFD and $F_{yw}^*$ is lower ($\rho_{Spearman} = -0.74$, $p$ value $< 0.001$; Fig. 11a) but nevertheless suggests that LFD is an important predictor for $F_{yw}^*$.

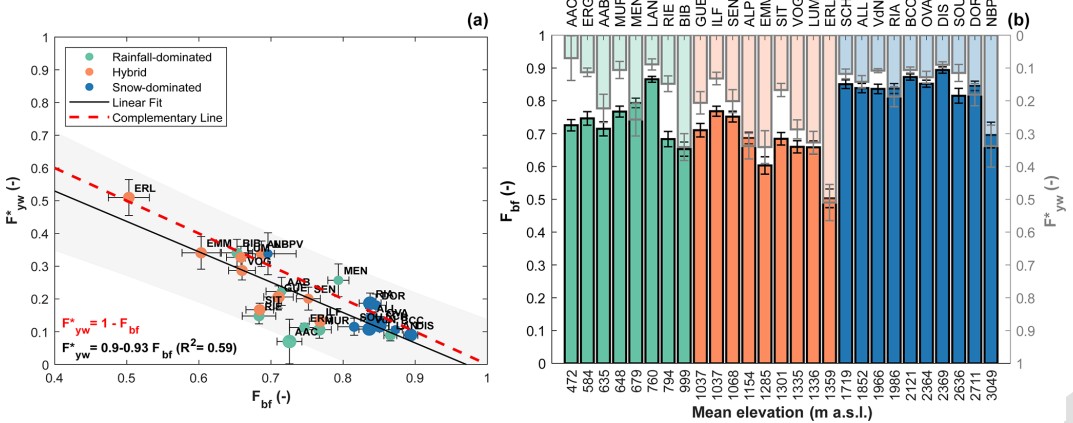

**Figure 9. (a)** Young water fraction plotted against fraction of baseflow: vertical and horizontal bars represent ± standard deviation. Gray area represents the 95 % prediction bounds of a linear regression of $F_{yw}^*$ on $F_{bf}$. Points dimension is proportional to $\tau_{yw}$. **(b)** Fraction of baseflow and young water fraction against mean elevation. Bars with black edge indicate $F_{bf}$ (left axis), while bars with gray edge indicate $F_{yw}^*$ (right axis). Vertical bars represent ± standard deviation.

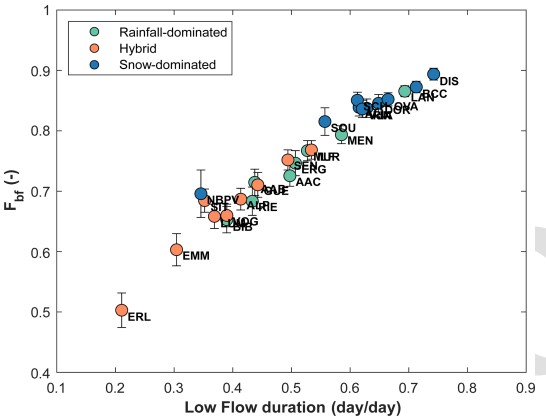

**Figure 10.** $F_{bf}$ against the low-flow duration (LFD).

## 4.5 The role of snowpack persistence

We explore next the presence of an ephemeral or seasonal snowpack as a relevant factor for the time concentration of liquid water input and for LFD. We consider the $F_{SCA}$, calculated as reported in Sect. 3.2, as a proxy of the snowpack persistence. The $F_{SCA}$ values for all the study sites are reported in Table 2. The underlying $f_{SCA}$ time series for six representative study catchments (two for each hydroclimatic regime) are reported in Fig. 12 (for complete results, see Fig. S4). All the catchments characterized by a seasonal snow cover (i.e., snow dominated) reveal a high $F_{SCA}$ (> 0.40, median of $F_{SCA} = 0.51$). Gradually smaller $F_{SCA}$ values correspond to increasingly more ephemeral snowpacks with intermittent snowmelt events during the winter season (Petersky and Harpold, 2018), as reflected by the spiky $f_{SCA}$ time series of hybrid and rainfall-dominated catchments (Fig. 12).

Our results exhibit a bell-shaped behavior of $F_{yw}^*$ with varying $F_{SCA}$ (Fig. 13a). Specifically, we observe a general increase of $F_{yw}^*$ for $F_{SCA}$ values roughly below 0.3. This result can be explained considering that especially in hybrid catchments (median of $F_{SCA} = 0.28$), but partially also in rain-dominated catchments (median of $F_{SCA} = 0.13$), streamflow receives relatively more young water from ephemeral snowpacks. These short-lived snowpacks melt during the winter season resulting in only a short delay between precipitation input and melt (i.e., no water aging in the snowpack), and correspondingly meltwater flows off quickly into the stream (reducing LFD, Fig. 14), e.g., in presence of a frozen surface soil layer. In fact, ephemeral and slightly thick snowpacks do not protect the underlying soil from freezing (Harrison et al., 2021; Rey et al., 2021). Even for low-elevation locations ($< \approx 1500$ m a.s.l.), freezing conditions are regularly observed during winter (Keller et al., 2017).

For $F_{SCA}$ values roughly higher than 0.3, we observe a decrease of $F_{yw}^*$ with $F_{SCA}$; here all the catchments of our data set are snow dominated. The mechanisms at play here are as follows: (i) in catchments with seasonal snowpacks, streamflow receives snowmelt in spring and summer that is at least partly older than 2–3 months (because part of the snow fell more than 3 months before the melt occurs); and (ii) the building up of a persistent, deep snowpack can promote deep vertical infiltration during the main melt period, either by insulating the soil and thereby preventing/reducing freezing (Harrison et al., 2021; Rey et al., 2021; Jasechko et al., 2016) or by gradual soil thawing during the melt period (Rey et al., 2021; Scherler et al., 2010). The temporal dynamic of snow accumulation and melt supports the pivotal role of snowmelt in recharging groundwater during summer in high-elevation environments (Cochand et al., 2019; Du et al., 2019; Flerchinger et al., 1992). A similar result

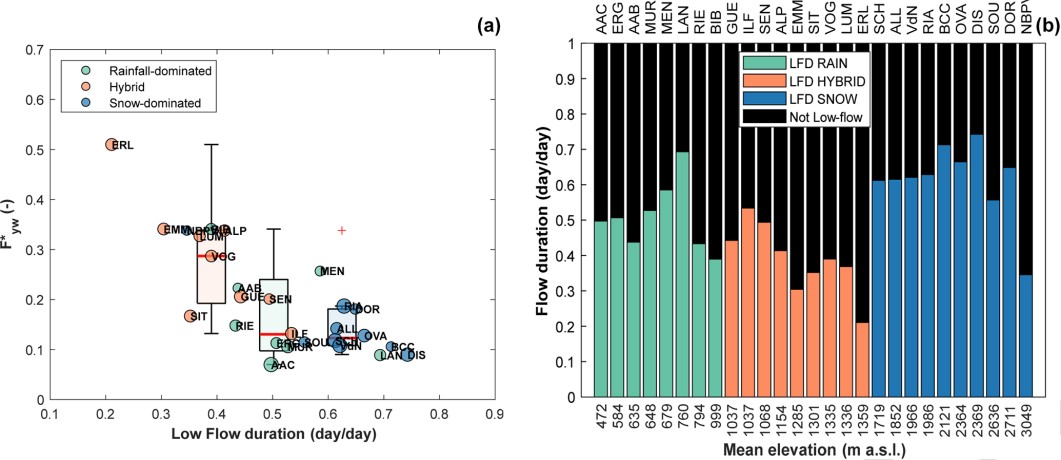

**Figure 11. (a)** $F_{\mathrm{yw}}^*$ against the low-flow duration, LFD. Boxplots of $F_{\mathrm{yw}}^*$ for catchments belonging to the same regime are plotted in correspondence to the median LFD. Points dimension is proportional to $\tau_{\mathrm{yw}}$. **(b)** LFD against mean elevation.

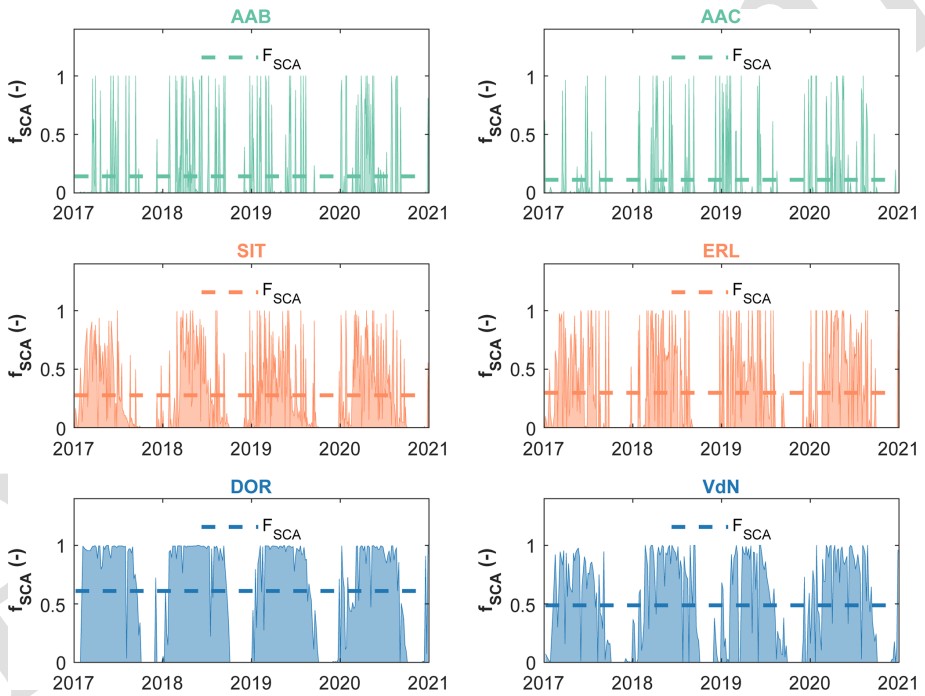

**Figure 12.** Time series of $f_{\mathrm{SCA}}$ for six representative study catchments (two for each hydroclimatic regime), illustrating the gradual increase of the $F_{\mathrm{SCA}}$ passing from rainfall-dominated to snow-dominated catchments.

was also found for dolomitic catchments (such as BCC and OVA) by Lucianetti et al. (2020), who discovered that different proportions of rain and snow contribute to the recharge of springs in the Dolomites, with a gradually higher meltwater contribution in springs with increasing elevation. This role of snowmelt supports our analytical choice of computing $F_{\mathrm{yw}}^*$ through the "direct input" approach, thus considering the snowpack as part of the catchment storage. In addition, the potentially large shares of meltwater that recharge groundwater via deep vertical infiltration also result in old water

sustaining winter baseflow (Fig. S5): the persistent snowpack and the absence of a *liquid* water input favor a groundwater storage release that creates a longer winter low-flow period that increases LFD (Fig. 14), thus reducing $F_{\mathrm{yw}}^*$, as discussed in Sect. 4.4.

$F_{\mathrm{SCA}}$ is strongly correlated with the mean catchment elevation in our data set ($\rho_{\mathrm{Spearman}} = 0.97$, $p$ value $< 0.01$; Fig. 13b). A posteriori, we could have considered mean elevation instead of $F_{\mathrm{SCA}}$ as a proxy for snowpack persistence. However, a priori, it could be approximative to describe the

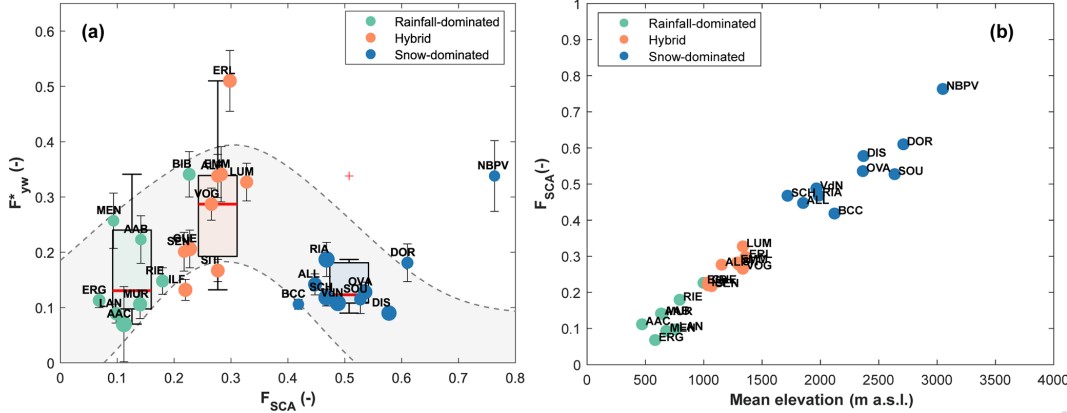

**Figure 13. (a)** Young water fraction against $F_{SCA}$. The gray area represents the perceptual bell-shaped behavior of $F_{yw}^*$ with increasing $F_{SCA}$. Points dimension is proportional to $\tau_{yw}$. **(b)** $F_{SCA}$ against mean elevation.

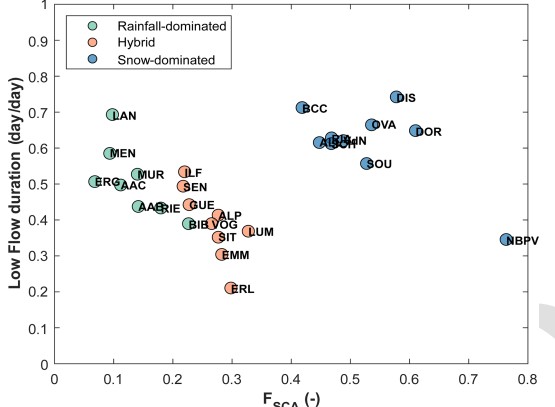

**Figure 14.** Low-flow duration (LFD) against $F_{SCA}$.

snow cover persistence only with the increasing elevation: the persistence of snow in a catchment also depends on overall topographic and climatic characteristics, specifically relating to snow and aspect (Painter et al., 2023). In fact, catchments with very different characteristics (e.g., different elevation ranges and different areas) can reveal a similar mean elevation, but the snowpack persistence could considerably change. This is the reason why we focused on $F_{SCA}$ that integrates these physical factors.

The above mechanisms are unable to explain the hydrological function of the glacier-dominated NBPV catchment, which has a very high $F_{yw}^*$ and is an outlier among the snow-dominated catchments (Fig. 13a). The high $F_{yw}^*$ of the high-elevation glacier-covered (42 %) catchment can be explained considering that the glacier melt produces high amounts of streamflow that transit the glacier system very quickly during the summer, given generally fast englacial and subglacial flow paths and the often limited water storage capacity in the glacier forefield (Müller et al., 2022; Saberi et al., 2019; Jansson et al., 2003). Schmieder et al. (2019) also found a high

young water fraction in an Austrian glacier-covered (35 %) catchment, leading them to the conclusion that the basin behaves locally like a "Teflon basin" with quickly transmitted ice melt.

## 4.6 Process interplay along elevation: perceptual model

The identified key drivers of young water fractions for rainfall-dominated, hybrid and snow-dominated catchments can conveniently be summarized into a perceptual model of the involved hydrological processes and their seasonal interplay (Fig. 15).

High-elevation catchments are characterized by long winter low-flow periods, resulting from the build-up of a seasonal snowpack, and are sustained by the emptying of groundwater (or old water) storage. Accordingly, such storage releases stored water, mainly old meltwater, for prolonged periods where the snowpack can last for several months (typically from December to early April) before releasing water during the melting season. Such seasonal snowpack can protect the underlying soils from freezing, thus promoting meltwater infiltration and groundwater recharge. From this viewpoint, snowpack is considered as part of the catchment storage, and there is a thin line between groundwater and meltwater in snow-dominated catchments. Snowmelt or rain events push out old meltwater to the stream during summer, suggested by the relatively high amount of daily baseflow during the melting season. During this period, the high catchment wetness might even lead to saturation and thereby favor fast flow paths of meltwater or rainwater, which in turn can temporarily increase the young water fraction. Despite this increase during high-flow periods, the prevailing winter low-flow periods in such systems lead to a reduction of the average annual young water fraction.

In catchments with an ephemeral snowpack, at lower elevations, snowmelt events occur regularly during winter such that water released from the corresponding short-lived snow-

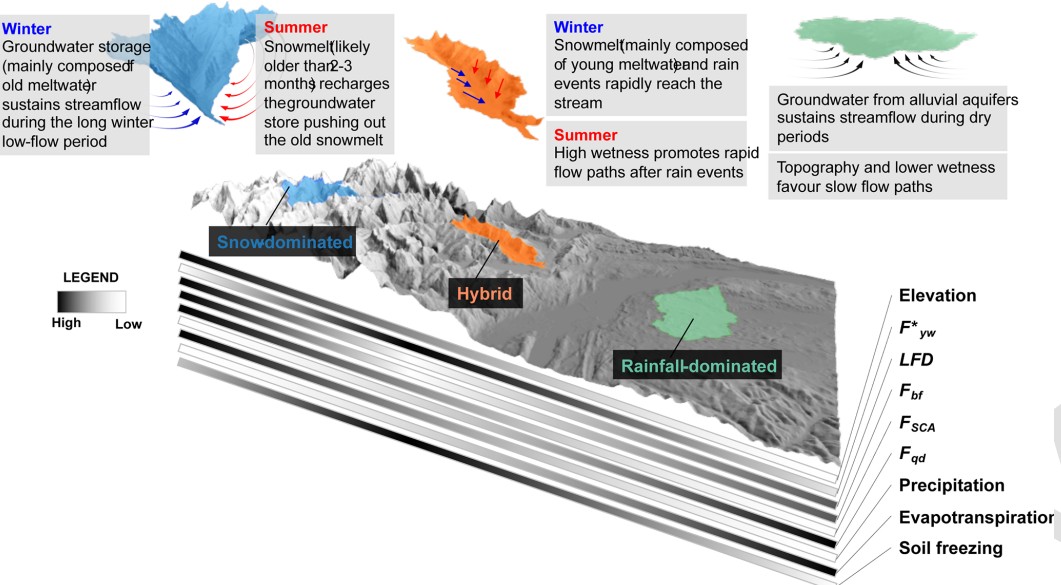

**Figure 15.** Perceptual model of the hydrological processes that drive the young water fraction variations with elevation. This model emerges from our analysis and harmonizes these results with those of previous studies. For snow-dominated and hybrid catchments, we indicate the dominant processes, occurring during summer and during winter, that lead to low and high $F_{yw}^*$, respectively. TS2

pack is likely younger than 3 months. Moreover, ephemeral snowpacks do not protect the underlying soils from freezing, and rapid flow paths can emerge during episodic or long-term soil surface freezing, by increasing the young water fraction. The high $F_{yw}^*$ of such systems is also explained by the simultaneity of snowmelt and rain events during extended parts of the year (leading to large volume of annual precipitation) and the relatively low (compared to rainfall-dominated catchments) evapotranspiration. Both processes increase the catchment's wetness and reduce the low-flow period's length.

Finally, at the lowest elevations, lower amounts of precipitation and higher evapotranspiration favor longer low-flow periods, mainly sustained by old groundwater from alluvial aquifers, which lead to both a $F_{yw}^*$ and a catchment wetness reduction. Further, the relatively flat topographies at the lowest elevations favor slow flow paths increasing the transit times of water.

How well current hydrological models can represent the interplay of these processes along elevation gradients is left for future research, but our perceptual model builds a solid basis for an improvement of theory-driven models (Clark et al., 2016).

## 5 Conclusion

This study proposes a conceptualization of the processes behind changes in young water fraction ($F_{yw}^*$) with elevation, defined here following Kirchner (2016a) as stream water that is younger than a threshold age of about 2–3 months. The analysis is focused on amplitude-ratio-based young water fractions for a set of 27 study catchments located in Switzerland and Italy, which span a wide range of geological and hydroclimatic conditions. The young water fraction estimates ($F_{yw}^*$) obtained from the phase and amplitude information of seasonal isotope cycles correspond to different young water threshold ages for the different catchments, which is a limitation of this work. However, the threshold ages vary only modestly from about 1.5 to 3 months.

Our analysis focuses on mountainous catchments to fill the knowledge gap, referring to the surprisingly low young water fractions at high elevations (> 1500 m a.s.l.), but we have also considered catchments at lower elevations to obtain a complete picture of the dominant hydrological processes at different elevations.

We have focused on variables and processes that were not previously considered for explaining elevation gradients of young water fraction. We have investigated the role of (i) groundwater storage potential, (ii) catchment storage contribution to the stream, (iii) low-flow duration and (iv) snowpack persistence. Our results suggest that (ii), (iii) and (iv) are interconnected: low-flow periods are generally sustained by old water deriving from the catchment storage, and the length of such periods is driven by the snowpack persistence at high elevations. The proportion of low-flow periods during the period of isotope sampling strongly influences the amount of old water contributing to the stream, thus reducing the estimated $F_{yw}^*$. Consequently, the low-flow duration, which varies with elevation, can be retained as a driver of the $F_{yw}^*$ changes with elevation. Given the importance of low-flow periods, we have also investigated the role of ground-

water storage potential, represented here by the portion of catchment area covered by Quaternary deposits. Our results suggest that an exhaustive description of the groundwater storage potential should be completed with more detailed geological information, e.g., the geology and topography of bedrock, the fraction of fractured bedrock and the deposits' thickness, which is challenging to retrieve from a geological data set. We have brought together the results of this analysis in a perceptual model that describes a framework for how hydrological processes control the $F_{yw}^*$ according to elevation, laying the foundations for an improvement of theory-driven models.

The strong complementarity between $F_{yw}^*$ and the mean fraction of baseflow obtained for our data set suggests that $F_{yw}^*$ could be estimated starting from automated baseflow separation techniques for catchments in which stable water isotope measurements are not available. This complementarity should however be validated in future work, by considering, e.g., alternative baseflow separation techniques and different hydroclimatic conditions.

Finally, the conceptualization of the hydrological processes described in this paper do not fit the high young water fraction of the single glacier-dominated catchment of our data set. In conclusion, we encourage future studies to compare and to collect isotopic data from glacier-dominated catchments to better understand the processes in such systems that, under glacier retreat due to climate change, will see a gradual transition to purely snow-dominated systems.

*Data availability.* Time series of both $\delta^2$H and $\delta^{18}$O in streamflow and precipitation, complemented with MeteoSwiss daily precipitation data, for the 22 Swiss catchments investigated by von Freyberg et al. (2018), are available in the data repository Zenodo at https://doi.org/10.5281/zenodo.4057967 `TS3` (Staudinger et al., 2020). Meteorological, hydrological and isotope data of VdN, BCC and NBPV catchments are available at https://onlinelibrary.wiley.com/action/downloadSupplement?doi=10.1002/hyp.13937&file=hyp13937-sup-0009-Supinfo2.zip (Ceperley et al., 2020).

Daily discharge data for the ERL, LÜM and VOG catchments are provided by the Swiss Federal Institute for Forest, Snow and Landscape research (WSL), Birmensdorf, Switzerland. Streamflow data for the AAB and GUE catchments are provided by the Office for Waste, Water, Energy and Air (WWEA) of the Canton of Zurich and by the Office for Water and Waste of the Canton of Bern, respectively. Daily discharge data of the remaining 17 Swiss catchments studied by von Freyberg et al. (2018) are provided by the Swiss Federal Office for the Environment (FOEN).

The .shp of the AAB, GUE, ERL, LÜM and VOG catchment boundaries are available from the data repository Zenodo at https://doi.org/10.5281/zenodo.4057967 `TS4` (Staudinger et al., 2020). The .shp of NBPV, BCC and VdN catchments are provided by Giulia Zuecco and Anthony Michelon (University of Lausanne, Switzerland) as personal communication. The DOR and SOU catchment boundaries are delineated in a GIS environment using the 10 m resolution digital elevation model (DEM) available from the Aosta Valley Regional Geoportal. Finally, the catchment boundaries of the remaining 17 Swiss catchments investigated by von Freyberg et al. (2018) are directly obtained from the Swiss Federal Office for the Environment (FOEN).

Quaternary cover for all Swiss catchments has been calculated using the Geological Atlas of Switzerland (GeoCover data set; 1 : 25 000 scale) available from the Federal Office of Topography swisstopo. For the DOR and SOU catchments, the vectorized Valsavaranche geological map (1 : 100 000 scale) is provided by the Cartography Office of SCT Geoportal. For the NBPV and BCC catchments, the .shp of unconsolidated sediments is provided by Giulia Zuecco.

DOR and SOU data are available from Alessio Gentile upon reasonable request.

*Code availability.* A GEE code for calculating snow cover area and cloud cover area time series over a region of interest has been made available at https://code.earthengine.google.com/8239cfe7aab498180e5c42475023cb80?noload=true (Gentile, 2023). A Matlab © code with the implementation of the Duncan (2019) baseflow filter and a Matlab © code for performing IRLS and for calculating $F_{yw}^*$, $\alpha$ and $\tau_{yw}$ are both available with the Supplement of this article.

*Supplement.* The supplement related to this article is available online at: https://doi.org/10.5194/hess-27-1-2023-supplement.

*Author contributions.* AG, NC, BS and SF identified the research gap, defined the methodology, developed the perceptual model and prepared the paper. AG analyzed the data set. DG, DC, MP and SF collected the water samples for the DOR and SOU catchments. GZ analyzed spatial data related to NBPV and BCC catchments. All authors revised the paper and gave final approval to the submitted version.

*Competing interests.* The contact author has declared that none of the authors has any competing interests.

*Acknowledgements.* This publication is part of the project NODES, which has received funding from the MUR-M4C2 1.5 of PNRR with grant agreement no. ECS00000036. We warmly thank the COST Action CA19120 – WATSON (WATer isotopeS in the critical zONe) for the acceptance of the application procedure for one virtual mobility (VM) and one short-term scientific mission (STSM). Both activities allowed to speed up the planning and conceptualization of this work as well as to stimulate the collaboration, the sharing of data and ideas. We acknowledge the support of the Valsavarenche Municipality and the Gran Paradiso National Park. The

authors thank Chiara Marchina (University of Padova, Italy) for the isotopic analyses of DOR and SOU samples. Finally, we thank Jana von Freyberg and the anonymous referee for their comments that helped to improve the paper significantly.

*Financial support.* This research has been supported by the Schweizerischer Nationalfonds zur Förderung der Wissenschaftlichen Forschung (grant no. PP00P2_157611), by the PRIN MIUR 2017SL7ABC_005 WATZON Project and by the MIUR – Excellence Department: National funds CE6 allocated to the DIST department.

*Review statement.* This paper was edited by Rohini Kumar and reviewed by Jana von Freyberg and one anonymous referee.

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

**Remarks from the language copy-editor**

**Remarks from the typesetter**