# Peer review of "Towards a conceptualization of the hydrological processes behind changes of young water fraction with elevation: a focus on mountainous alpine catchments"

_EGUsphere, 2022_

## Referee Comment (RC1)

The work of Gentile et al. investigated the causes for young water fraction ($F_{yw}$) variations with elevation ($F_{yw}$ is low at high altitudes) in Alpine catchments. The study areas are 27 catchments in Switzerland and Italy. The authors proposed new criteria for catchment classification into different hydro-climatic regimes. To gain insight into the reason for $F_{yw}$ variations with elevation, this author used a new set of hydrological variables, namely the fractional snow cover area ($F_{SCA}$), the fraction of quaternary deposits ($F_{qd}$), and the fraction of baseflow ($F_{bf}$). In general, the idea of this paper about what drives $F_{yw}$ variations with elevations is novel and of interest for understanding the functioning of catchments in Alpine regions as well as for understanding flow and transport in this region and potentially in other areas. However, the methodology and results do not fully support this idea. The text was not well written. Please find my main comments and line-by-line comments below.

**Main comments**

- Why did the authors need to propose a new criterion for catchment classification? The authors used two variables: (1) streamflow ratio between different months and (2) snow cover fraction for the proposed catchment classification, but later they adjusted the threshold of these two variables to have consistent results with Staudinger et al. (2017). Why didn't they just use the method of Staudinger et al. (2017)?

- The objective is to investigate what drives $F_{yw}$ variation with elevation. The authors proposed using a new set of hydrological variables, but what are the relations between these variables with elevation? For example, what are the relations between $F_{SCA}$, $F_{qd}$, $F_{bf}$ with elevation? With $F_{SCA}$, I can infer from the text, but it was not explained in the text until the last sections (Section 5.2) of the manuscript. $F_{SCA}$ cannot be directly related to elevation, instead, it needs to be related to the catchment classification then from catchment classification to mean elevation. However, in other areas, can we still relate $F_{SCA}$ to elevation? With the other variables ($F_{qd}$ and $F_{bf}$), it is unclear to me what are their relations to elevations. In addition, $F_{qd}$ does not seems to be a good variable because there is no significant relation between $F_{yw}$ and $F_{qd}$.

- The manuscript needs to be restructured and revised. There is a lack of clarification in the text. More description of the study area characteristics is needed. Much of the information provided in *Study Sites*, and *Material and Methods* is not relevant (e.g., shape file, detailed source of data, etc.). Instead, citing the sources of the various data (both from individuals and organizations) can be moved to either the *Authors' Contributions* or *Acknowledgements, or in the supporting information* Sections or to a table rather than describe them within the text of the article, making it very difficult to read such detailed information. If possible, I would also suggest the authors publish their data in an open repository.

**Minor comments**

Title: "$F_{yw}$" could be changed to "young water fraction" for general readers.

L14: "The young water fraction ($F_{yw}$),..., is increasingly used in hydrological studies, replacing the widely used Mean Transit Time, which is subject to aggregation error." This sentence provides misleading information. I think $F_{yw}$ cannot replace Mean Transit Times (MTT) since the two characterize different aspects of the transit times, e.g., $F_{yw}$ contains information about the younger part of the TT distribution

(how much water in outflow is younger than 0.2 years) while MTT contains information about the whole TT domain. "aggregation error" could be changed to "aggregation bias".

L33-34: The sentence "..$F_{bf}$, considered…complement of $F_{yw}$" does not clearly show the relation you found between $F_{yw}$ and the baseflow fraction. Please be clearer about what you mean by explicitly saying that $F_{bf}$ is a good proxy for $F_{yw}$ as the higher $F_{yw}$ is, the lower $F_{bf}$.

L44: "the streamflow is older than the annual snowmelt" is not clear to me, what is the age of streamflow and the age of snowmelt water in this case?

L46: why "even"? I would expect exactly that during the absence of rainfall and snowmelt the streamflow is mainly sustained by groundwater.

L46-50: The two sentences here do not seem to be connected, one about residence time and the next one about transit times.

L53: "Kirchner (2016a, b) proposed a new metric to quantify water age at the catchment scale". I think you are mentioning the $F_{yw}$, I don't think this is "the water age at the catchment scale" but the amount of water with age < 0.2 years. How can we know the "water age at the catchment scale "only based on the amount of water in outflow (discharge) that is < 0.2 years?  For example, if  $F_{yw} = 0.2$, what is the "water age at the catchment scale"

L55-58: please revised the sentence structure

L58-59: please see my comments on line 14

L70: "In line with these findings" can be removed because Lutz et al. (2008) did not state that Fyw above 1500 m decreases

L82-83: "…more efficient groundwater recharge, consequently reducing or increasing the young streamflow…" It is not clear to me, should it be "reducing" only instead of "reducing or increasing"?

L88: "…remarkable fraction of groundwater…" it is a bit vague, could you please be more precise?

L91-92: "…a dynamic storage contribution to streamflow…" Please clarify this term.

L99: Why don't the authors use a new set of hydrological variables (Fsca, Fqd, Fbf, WFI) in combination with traditional variables to gain new insights into the Fyw along elevation gradients?

L104: "…into three hydro-climatic regimes proposing a new criterion of classification…" Why? I think a brief explanation is needed.

Sections 2 and 3.1: Both sections about the data (e.g., Section 2: existing data, additional dataset, complete data, and Section 3: discharge data and catchment boundary), why do the authors need two different sections? The data description section (entire section 2) needs to be restructured and revised to make it more concise and clearer. I think this can be done using a table. In the text, the authors could summarize and report key information, so the reader does not have to search through the many sources you have cited. The authors can here focus more on catchment attributes, such as climate (e.g., average annual precipitation and discharge), land use cover, geology, and discharge.

"*Furthermore, 21 out of the 22 … (Staudinger et al., 2017)*". This part is not relevant in my opinion.

"*Two high-elevation catchments … Arnoux et al., 2021)*". This part is not relevant in my opinion.

L147: In my opinion, the "Complete Dataset" subsection is not necessary. It is sufficient to illustrate the existing data in subsection 2.1 and conclude the section with 2.2.

L156-160: Like von Freyberg (2018) … are reported in Table 1. If subsection 2.3 is deleted, move it to 2.2 as the final sentence.

Figure 1: the background cannot be easily seen, I think you could replace with a DEM map. In addition, I cannot differentiate between Quaternary deposits and hybrid catchments visually.

Table 1: I am curious to see the relation between average elevations and average slopes for the 27 catchments, is there a positive correlation? (also for average elevation with annual precipitation)

Section 3.1: Here, I would expect more description of the discharge dynamics (e.g., giving an order of magnitude to these data by telling what is the annual discharge, whether the runoff is seasonal, etc). I would suggest moving the description of how discharge was measured and derived to the appendix. The source of data could be combined into the same table suggested for section 2 (or move to the appendix or data availability section).

Line 190: I suggest mentioning the study period for the isotope data and Fyw for the different study catchments since it is different.

Figure 2: In summer there is a higher average monthly flow from snow-dominated catchments than from rain-dominated ones (due to increased snowmelt, I suppose), and in winter it is the other way around. Please explain this better in the text because it is not clear. In addition, it is not easy to differentiate between the three boxplots, I would suggest having three separated boxplot figures with the same y axis limit. This figure should be described in the text (there is no description of this figure, it was only cited in line 243)

L197: no comma after "where"

L221: As I understood from the text (before and after this line), there is indeed a "formal" classification method

Section 3.3: After reading the entire manuscript up to section 3.3. I am not clear why the authors need to classify streamflow into three regimes and why the classifier should be based on snow-related characteristics (e.g., snow cover area).

L240: should it be "it is expressed in mm per unit area and time step"?

L251: "…more than weekly…" do you mean biweekly?

Eq: (5) the denominator ($N_{tot} - N_{clouds}$): This could result in an overestimation of $f_{SCA}$. What is the maximum fraction of cloud cover in these images?

L276-279: The error in fSCA is still there with the "moving window" approach, it is just smoothed. Anyways, at the end, you calculated the average fSCA over the whole period so applying "moving average on a window" does not have any effect?

L282-289: "*Some authors have revealed … Fyw in Alpine catchments*". This is more suitable for Introduction than Methodology. In addition, what is "key possibility"? Does it mean "high possibility"

L292: … *23 Swiss catchments* … Is Fqd calculated only for 23 sub-catchments, while WFI and Fbf for all 27? Why? How does it affect the interpretation of the results? Be clearer about which indices are available for each study site.

L299-301: *For the DOR and SOU … provided by Dr. Giulia Zuecco*. This part is not relevant here, should be moved to the data section.

L315-318: "*For VdN, NBPV and BCC catchments we consider the time windows … we consider discharge measurements in the period November 2017 - January 2022*". I think you should indicate at the beginning the different study periods, because it is confusing to read a lot of data (e.g., stable isotopes of water, Fyw, streamflow...) and indices (e.g., Fqd, WFI, Fbf) for your methodology and find out that your study areas were analyzed in different periods. You should say this explicitly each time you mention a new data item or index or create a table in which you explain it.

Section 4.1. I think this can be moved to the data section or supporting information, as this is only for 2 catchments.

L334-335: "these have the same names as the ones proposed by Staudinger et al. (2017 but the classification is not based on the same criteria" why? I think should be explained earlier in the methodology section.

L336-337: "In order to achieve a classification as consistent as possible with that of Staudinger et al. (2017), but based on these two variables, we propose the thresholds presented in Table 2:" I cannot understand why. If the authors want to have consistent results with Staudinger et al. (2017), why did not they use the method proposed by Staudinger et al. (2017)?

L345-346: "*Following our classification scheme, … and 9 snow-dominated catchments*". How do you compensate for the fact that the catchments data belong to different periods?

L353: "snow-regime" should be explained here

L354: "for the first order estimate of the second classifier" what does it mean?

"Section 4.3: New explanatory variables for the Fyw elevation gradients" I would expect all subsections in section 4.3 will use variables that are related to elevation to explain the relation between Fyw with elevation. However, I cannot see what is the relation between the variables in the section title (e.g., Section 4.3.1. Fractional Snow Cover Area (fSCA) and Fyw) and elevation (Please also see my main comments)

L361-368: part of this information was already described in the introduction, can be removed here or merged into the introduction.

L389-391: *Our results … for hybrid catchments (median Fyw of 0.32) …* Why are there these differences? I suggest arguing and explaining them.

Figure 4a can be moved to the data section, figure caption: "the horizontal bars correspond to +/- standard deviation" of slope or elevation?

L367: "Despite this" why should an increase in slope with elevation result in a correlation between Fwy and slope?

L393: "lowering" could be changed to "decreasing of Fyw with increasing FSCA"

L408: Why were the two catchments with Fqd = 0 is excluded? why do the group need to have features as close as possible to those used by Arnoux et al. (2001)

Section 5.1. I would expect here a discussion about the advantages of the new classification compared to other approaches (e.g., Staudinger et al., 2017), especially when the focus of the study is to understand what drives $F_{yw}$ variations with elevations. The text written in this section does not seem to be relevant to this study.

L473: How does your work harmonize with previous results? I suggest expanding this point by making it clearer and highlighting the novelty of your work compared to the previous studies.

L477: "increase of precipitation and slope with elevation (Fig.12a, Fig. 4a)" I cannot see this in these figures

L483: *higher up* sounds odd. Simply say upstream.

L484-486: *Therefore, it is more likely that ... possibly ephemeral, snowpack*. I do not see a connection between these two statements. If you are saying that lower-order (i.e., more downstream) channels release greater amounts of old water than higher-order (i.e., more upstream) channels, why do you say that water age decreases with elevation? Please clarify this point.

L493: "a persistent, deep snowpack can promote deep vertical infiltration by insulating the soil and thereby preventing freezing" do you mean this happens in winter? If in winter, there might be only snow, how can it be melted and promote deep vertical infiltration? Where is the source of water for vertical infiltration?

L495: what's a temporal concentration? Make it clearer.

L499-501: This is for the karst area, how relevant is it for your area?

L518: I suppose the fast flow paths are due to the fact that the glacier acts as an impermeable layer and thus promotes rapid overland flow? Please explain what you mean.

Figure 13: Which subfigure is for lower altitudes (< 1500m) and which one is for higher altitudes? Figure caption: the word "panels" can be removed because I thought a panel always consists of two subfigures (e.g., the lower panel contains two subfigures c,d)

L531: "unconsolidated sediments are not the only…" could be changed to "water storage in unconsolidated seidments are not the only …"

---

## Referee Comment (RC2)

Review
Title: What drives Fyw variations with elevation in Alpine catchments?
Author(s): Alessio Gentile et al.
MS No.: egusphere-2022-921
MS type: Research article

**General comments**

Gallart et al. address the scientific questions of "… *what drives $F_{yw}$ variations with elevation in Alpine catchments clarifying why $F_{yw}$ is low at high altitudes»* (L20). For this, the authors combine existing and new $F_{yw}$ values from Switzerland and Italy and compare them with several other variables that describe snow cover, baseflow conditions, and geology. From these comparisons the authors develop a perceptual model, suggesting that a longer persistence of the seasonal snowpack results in deeper groundwater flow paths and thus smaller $F_{yw}$ values, in contrast to hybrid catchments with ephemeral snow packs. The authors also present a new classification scheme to identify a catchment's hydro-climatic regime. The analysis of the used data is thorough and most figures are clear and informative. The analysis of satellite images to explore the linkages between snow cover duration and $F_{yw}$ are certainly interesting. However, I would like to encourage the authors to highlight more the novelty of their findings and the scientific contribution of their work, considering that they cite several papers in which comparable analyses have been carried out and similar conclusions (with respect to flow and storage processes) have been reached.

I think that the **research objectives** (or research questions) should be formulated more explicitly in the Introduction in order to guide the following analysis. It is not clear whether the authors attempt to explain the scatter in the $F_{yw}$-gradient relationship (L76), the low $F_{yw}$ values in steep and/or high-elevation catchments (L79), or both.

L156 "*we classify the catchments in the three hydro-climatic regimes (snow-dominated, hybrid and rainfall-dominated) proposed by Staudinger et al. (2017), but we introduce a new formal criterion of classification*": Why is a **new definition of the catchments' hydro-climatic regimes** needed? As far as I can tell, only two catchments, BIB and GUE, were newly classified. The new sites outside of Switzerland could have easily been categorized as hybrid or snow-dominated based on their streamflow and topographical data. Furthermore, the discussion of this new classification scheme (Sect. 4.2 and 5.1) somewhat distracts from the main topic of the paper, which is the investigation of small $F_{yw}$ in high-elevation catchments.

I was surprised to see that the authors did not include annual or seasonal **precipitation** in their analysis. This variable should be tightly related to $F_{bf}$ and $F_{SCA}$. Annual precipitation is also very low at some Swiss high-elevation sites, which would also explain why $F_{yw}$ is low there. What is the reason for not considering precipitation at all?

The important aspect of **snow pack storage** in high-elevation, snow-dominated catchments, which the authors only touch on in the Conclusions section, should instead be brought up much earlier in the manuscript. In fact, it has been discussed already in another paper: «*Another analytical decision that affects the interpretation of $F_{yw}^{*}$ and $F_{yw}$ relates to whether snowpack storage is considered to be part of catchment storage, or not. If one measures precipitation to the snow surface as the catchment input, then snowpack accumulation and melt are implicitly included in catchment storage (e.g. Staudinger et al., 2017). In this case, comparisons of seasonal cycles in precipitation and streamflow should reflect the young water fraction resulting from the combination of snowpack and subsurface storage. Alternatively, if one uses precipitation and snowmelt arriving at the soil surface as the catchment input (for example, with melt pan lysimeters, or modelled snowpack out- flows), then snowpack accumulation and melt are implicitly excluded from catchment storage. In this case, comparisons of seasonal cycles in streamflow and sub-snowpack catchment input should reflect the young water fraction resulting from subsurface storage alone. Because the total catchment storage in the first case (including snowpack storage) is larger than the subsurface storage alone, the resulting young water fractions are expected to be smaller.*» (von Freyberg et al., 2018). In addition, in high-elevation catchments with perennial snow packs, snowmelt in spring and

summer is likely to be older than 2-3 months (because the snow fell more than 3 months before the melt occurs). As a result, although summer discharge might be high it will consist mainly of old snowmelt and groundwater rather than recent rainfall (i.e., $F_{yw}$ is small). In hybrid and rain-dominated catchments, streamflow receives relatively more young water from young snow packs and recent rainfall events, respectively.

The authors seem to overlook this storage aspect of the snowpack and instead focus mainly on the groundwater contribution to streamflow (L82). A main finding of the paper is a strong negative correlation between the baseflow fraction $F_{bf}$ and $F_{yw}$ (Sect. 4.3.3, Fig. 10) from which the authors derive several statements which I'd like to comment on (Sect. 5.4):

L553: "*We find the highest $F_{bf}$ for snow-dominated catchments confirming the presence of high subsurface storage, contributing to streams, in high-elevation catchments*". I would include the snowpack as part of the storage here because winter precipitation is stored in the snowpack until summer when it recharges aquifers or runs off into the stream.

L554.: "*Moreover, the annual baseflow is strongly positively correlated with the $F_{SCA}$ ($\rho Spearman= 0.81$ p-value < 0.01) suggesting a major groundwater contribution with increasing snow cover persistency (Fig. S6)*". This depends strongly on your baseflow estimation method. Further, increasing baseflow and snow cover persistency are both results of increasing catchment elevation and/or annual precipitation. Thus, baseflow cannot simply be linked to snow cover persistency.

L558: "*The hydro-climatic regime is generally a good indicator of the proportion of young water that contributes to streamflow...*" What does this mean exactly? If the authors refer to Tab. 3, there is quite some overlap between the rainfall- and snow-dominated regimes with respect to $F_{yw}$, and thus $F_{yw}$ cannot be estimated from the regime types alone.

L570 (&L37): "*Therefore, we can conclude that the contribution of groundwater storage to streamflow, which is driven by snowpack duration, can be considered as the best explanatory variable of the $F_{yw}$ elevation gradients.*" Again, I would rather argue that not snowpack duration but rather storage capacity (both in the subsurface and the snowpack) together with the hydro-climatic conditions (P-ET) and catchment properties affect the contribution of old water (not necessarily only groundwater) to streamflow, and thus $F_{yw}$. In high-elevation catchments, the snowpack can function like a subsurface water storage that releases (>3 months) old water during the melting season. This old water is meltwater, not groundwater and I suspect that the baseflow separation method used in this paper is not able to differentiate between the two.

Based on the analysis of **slope data** the authors conclude that (L370) "*... that there is an increasing rate of infiltration when the hydro-climatic regime transitions from hybrid to snow-dominated.*". I don't think that this statement is well supported by using slope data in Fig. 4 (no data on infiltration is provided). Instead, the only conclusion that can be drawn from the data presented in this manuscript is that the hybrid catchments receive more precipitation than the rain-dominated catchments (L478), resulting in more recent precipitation becoming streamflow, i.e. higher $F_{yw}$ values. This is analogous to earlier findings in von Freyberg et al. (2018): "*... young water fractions tend to be highest in humid catchments where prompt runoff response is facilitated by fast flow paths and/or high-intensity precipitation events.*"

One outcome is a "*perceptual model of how snow persistency explains $F_{yw}$ during winter and summer along topographic gradients*". This **model**, presented in Fig. 13, tries to summarize the combined effects of catchment properties (steepness, elevation) with processes (ET, wetness, snowmelt). The resulting figure is very complex and difficult to understand. For instance, if a reader seeks to understand the figure without reading the entire paper, is not clear as to what "*increases/decreases with elevation*" means. Does this refer to increases/decreases of $F_{yw}$ within a single catchment or between different (high- to low-elevation) catchments?

**Specific comments**

The **title** of the manuscript does not well reflect the content of the paper. It rather gives the impression that $F_{yw}$ was studied along elevation gradients within (individual) catchments. In addition, the term "Alpine" suggests that solely mountainous catchments within the Alps mountain range were considered, however, catchments such as ERG, AAB and MEN are located in the Jura Mountains and Swiss Plateau, respectively. It would be nice to define early on what is meant here by Alpine, given that the Introduction starts with the general statement (L41) "*Alpine catchments are assumed to generate a high share of surface runoff ...*"

Ideally, the **time periods** that were used to calculate the various metrics should be the same as those of the isotope data used to calculate $F_{yw}$. As far as I can tell, this has been considered only for $F_{bf}$, whereas $F_{SCA}$ was determined based on satellite data from 2017-2021. For WFI and $Q_{June}/Q_{DJF}$, no information is provided. The $F_{yw}$ values in von Freyberg et al. (2018) only cover the time periods 2010-2015, which is not even overlapping with the satellite images used to determine $F_{SCA}$. I would like to encourage the authors to compare data only from the same time periods, especially when these periods included extremely dry/wet climatic conditions.

The terms **elevation and steepness** should not be used synonymously, as in L361:" *Initial evidence of low $F_{yw}$ in high-elevation catchments is given in the work of Jasechko et al. (2016). Based on the analysis of 254 worldwide watersheds, their work reveals a reduction of $F_{yw}$ in steeper terrains.*" Also, low-elevation (rainfall dominated) catchments can be very steep, and there surely exist high-elevation (snow-dominated) catchments with flat topography.

When I look closer at the **$f_{SCA}$ time series** (Fig. 5), I wonder how it is possible that the AAC catchment at around 500m asl. was almost entirely snow covered in summers of 2018 and 2020 ($f_{SCA}$ around 1)? The same is true for the catchments BIB and ERL where the snow cover usually disappears by June each year. Can it be that $f_{SCA}$ tends to be over-estimated with your approach? I would also expect $F_{SCA}$ to be strongly correlated with (mean) catchment elevation so that elevation instead of $F_{SCA}$ could be used in your analysis. As can be seen in Fig. 12, a similar grouping of catchments emerges.

L275 mentions that "*The Noce Bianco Pian Venezia (NBPV) catchment is an exception since it generally has snow over the glacier also during summer.*". As far as I remember, the catchments VdN, DIS and OVA are also partially glacierized. Should they be considered as exceptions as well?

Fig.10: A very similar result is presented already in von Freyberg et al. (2018) where $F_{yw}$ and the quick-flow index QFI, the inverse of the baseflow index, showed a significant positive correlation (note that the QFI and $1/F_{bf}$ will likely not be exactly the same, although both were calculated through digital filtering of discharge time series).

L484: "*In addition, higher order channels, higher up, are more rarely activated than lower order channels that are more often active*" If the authors refer to Strahler stream orders here, higher elevation streams usually have low Strahler orders (starting with first-order streams). The Strahler stream orders increase downstream.

L560-565: Why was BCC not classified as snow-dominated, based on the evidence from previous research?

L566-569: Is it possible that precipitation isotopes in the NBPV catchment were sampled differently compared to the other catchments in this study, e.g. with a heated precipitation collector? This could result in a larger $A_S$ value. Can the authors confirm that the precipitation isotope sampling in the snow-dominated catchments was comparable across all sites?

L596: "*…leads to high baseflow throughout the year...*". This contradicts the data shown in Fig. 9. I would suggest to replace 'baseflow' with 'baseflow fractions $F_{bf}$'.

**Technical comments**

The language of the manuscript is often not precise and needs to be improved. Some sentences are difficult to understand, e.g.

- (L310) "*Additionally, Duncan (2019) provides a specific technique that allows estimation of separate components with physical relevance in the case that baseflow separation techniques were not applied to describe physical processes.*" This sentence is redundant and not scientifically specific (e.g., what are "separate components with physical relevance"?).

- (L33) "*Finally, our work highlights that Fbf, considered as a proxy for groundwater flow, is roughly the one's complement of $F_{yw}$*". Isn't $F_{bf}$ rather a proxy for the groundwater contribution to streamflow? It does not provide any information about flow processes. What does "*roughly the one's complement of $F_{yw}$*« mean?

- L34 «*...we find high Fbf during all low-flow periods, which underlines that streamflow is mainly sustained by groundwater in such flow conditions.*» That high $F_{bf}$ represents a major contribution of groundwater to streamflow is implicit in the method of Duncan (2019). This is not a new finding.

- L496 "*the temporal dynamic of snow accumulation and melt and its effect on deep infiltration supports the pivotal role of snowmelt in recharging groundwater during summer in high-elevation environments ...*" This sentence is redundant. *Snow melt affects deep infiltration* is equivalent to *it plays a role in recharge*.

Sect. 3.2: To indicate whether a variable was flow weighted, earlier papers have added a "*", and thus I would suggest to write $F_{yw}^{*}$ and $A_{S}^{*}$ here as well.

L328: Please verify whether the flow-weighted young water fraction of SOU is indeed 0.01. If so, the following statement "*while flow-weighted $F_{yw}$ remains unchanged for the very small lateral subcatchment*" is false.

**References**

Duncan, H. P.: Baseflow separation – A practical approach, J. Hydrol., 575, 308–313, https://doi.org/10.1016/j.jhydrol.2019.05.040, 2019.

von Freyberg, J., Allen, S. T., Seeger, S., Weiler, M., and Kirchner, J. W.: Sensitivity of young water fractions to hydro-climatic forcing and landscape properties across 22 Swiss catchments, Hydrol. Earth Syst. Sci., 22, 3841–3861, https://doi.org/10.5194/hess-22-3841-2018, 2018.

---

## Community Comment (CC1)

By Peter Jansson

*This review was prepared as part of graduate program Earth & Environment (course Integrated Topics in Earth & Environment) at Wageningen University, and has been produced under supervision of dr Ryan Teuling. The review has been posted because of its potential usefulness to the authors and editor. Although it has the format of a regular review as was requested by the course, this review was not solicited by the journal, and should be seen as a regular comment. We leave it up to the authors and editor which points will be addressed.*

**Overall Impression**

Gentile et al. investigate possible mechanisms that may cause variations in young water fraction ($F_{yw}$) in streamflow to vary with altitude in elevated alpine catchments. Previous studies have found that $F_{yw}$ increases in catchments with higher elevations, however catchments higher than 1500 m a.s.l. show an inverse relationship ($F_{yw}$ decreases with elevation), the causes of which are poorly understood.

The authors of this paper correlated $F_{yw}$ in 27 alpine catchments with measures for snow storage ($F_{SCA}$ – fractional snow cover area) and groundwater storage ($F_{bf}$ – fractional baseflow, $F_{qd}$ – fractional quaternary deposits). The results led to the authors developing a perceptual model showing that catchments with higher $F_{SCA}$ (prominent in elevated catchments with persistent snowpacks) promote more infiltration and gradual discharge in winter, leading to lower $F_{yw}$. This model was corroborated with the strongly inverse relationship between $F_{yw}$ and $F_{bf}$.

This paper presents a reasoning for 'confounding' results found in existing research that was not previously explained. The inverse relationship between elevation and $F_{yw}$ in elevated catchments can be seen in the work of von Freyenberg et al., (2018) and Ceperley et al., (2020). Gentile et al. consequently investigate whether these factors indeed drive $F_{yw}$ variations. The results obtained show a strong relationship between $F_{yw}$ and $F_{bf}$, and a puzzling "bell-shaped" relationship between $F_{yw}$ and $F_{SCA}$. These results on their own right would provide a limited explanation to the 'confounding results'. However, by tying together the different hypothesised processes indicated by the results in the perceptual model, the authors manage to formulate a coherent and thorough concept of catchment processes that would explain the results. Figure 13 illustrates the different processes in the perceptual model very well, allowing for further discussion and research on the topic.

I feel that the reasons for undertaking this research have been justified. Though it was not linked to a pressing environmental concern, this paper continues the use of a relatively new method ($F_{yw}$ variations) to understand catchment dynamics. The paper directly addresses a knowledge gap found in existing research and sought to explain what appeared to be confounding results.

The method used for determining water age at the catchment scale – $F_{yw}$ – has been developed fairly recently (Kirchner, 2016a), yet appears to be robust, and has been used in earlier papers (i.e. Ceperley et al., 2020; Stockinger et al., 2019; von Freyenberg et al., 2018). Though it appears logical to use the same type of data as the earlier research this paper is based on, the authors should be credited for using such a contemporary empirical method to describe catchment behaviour.

Another useful finding of this paper is the introduction of a new, formal classification of alpine catchments into 3 hydro-climatic regimes. Though similar classification systems have existed, the system proposed in this paper used formal and objective criteria, making it suitable for cross-catchment datasets even beyond the alps. This classification system also includes a new proxy for estimating snow coverage using only discharge data – $Q_{june}/Q_{DJF}$. This can be a very useful tool for future research looking to estimate snow cover requiring the necessary satellite data to estimate $F_{SCA}$ directly.

At first glance, the results and conclusion appear impressive. Interpretations of the results were synthesised to address a relevant knowledge gap regarding the understanding of alpine catchment processes. However, the perceptual model proposed by the authors do not appear to be sufficiently backed up; the explanatory

variables used may not accurately represent the catchment processes as the authors intended. In my view, the authors should adapt their methodology to limit the model shortcomings, while also investigating other explanatory variables which can give more credibility to their conclusions. **Overall, I believe the paper requires significant revisions before it can be considered for publication.**

**General Comments**

**Use of $F_{bf}$ to represent groundwater storage processes**

Arguably a larger weakness of the paper compared to the one above, $F_{bf}$ is by definition related to $F_{yw}$, so the correlation found may not be strong enough evidence to convincingly corroborate the perceptual model. It could already be reasonably assumed that streamflow with higher mean residence times would indicate a slower flow path. Additionally, $F_{bf}$ does not directly infer the type of catchment storage; catchments with high snow storage and low groundwater storage may also yield higher $F_{bf}$. Baseflow could also account for a myriad of other catchment characteristics, such as the mean slope and shape coefficient. Hence, the assumption made by the authors that $F_{bf}$ could be used as a proxy for groundwater storage processes is questionable.

As a result, the perceptual model and conclusions may have been supported with the wrong evidence. In line 549 the authors state that the inverse correlation between $F_{bf}$ and $F_{yw}$ "clearly indicates that the observed patterns of $F_{yw}$ are related to water partitioning between the surface and subsurface". However, $F_{bf}$ may be inferring to other processes, inferring the need for cross-validation. **It is hence highly recommended that the authors run additional correlations to identify how $F_{bf}$ links to groundwater-related processes.** Figure 8 a) shows such an example, where $F_{qd}$ is shown to be positively correlated with WFI. This helps justify the inclusion of $F_{qd}$ and WFI as explanatory variables that act as proxies for groundwater flow. A correlation between WFI and $F_{bf}$ would hence illustrate the strength of $F_{qd}$ as a proxy for groundwater flow and storage. If WFI and $F_{bf}$ are indeed positively correlated, the use of $F_{qd}$ would be further justified. A correlation between $F_{SCA}$ and $F_{bf}$ would show the extent of snow-related processes being incorporated in $F_{bf}$.

**Trends with precipitation not analysed**

Precipitation can have a large effect on $F_{yw}$ variations: catchments with higher rainfall may have faster flow paths and hence higher $F_{yw}$. Figure 12a illustrates differences in precipitation levels, and seems to indicate an increasing trend between $F_{yw}$ and precipitation. That would indicate that precipitation should be included as an explanatory variable. Neglecting to include this variable further weakens the strength of the discussion and conclusion; the authors intended to investigate potential drivers of $F_{yw}$ variations with elevation, without including a variable that is known to affect catchment residence times and to vary with elevation. **I therefore suggest that the authors include correlations between $F_{yw}$ and precipitation.** Doing so would provide a more holistic view of the alpine catchment processes, and further enrich the perceptual model, conclusions, and the scope for further research in this new and exciting field.

**Specific Comments**

**Minor findings may deserve more attention.** From the methodology of the paper, a new hydro-climatic regime classification came about, including a new proxy variable that can be used to estimate $F_{SCA}$ with high accuracy ($R^2=0.99$). Though useful in strengthening the methodology, the authors only dedicate a small section of the discussion and a couple of sentences in the conclusion on this classification scheme. The development of the hydro-climatic regime classification is given a lot of attention to despite not being part of the research objectives. If the authors believe that this new method has potential in future studies, it is recommended that these methods could be mentioned more explicitly: either being incorporated in the abstract and treated as a research objective, or moving elaborations to the appendix or to a separate publication.

**Isotope data timespan and resolution not stated.** This uncertainty propagates to form some of the uncertainty in $F_{yw}$. However, by including the temporal data span/resolution, and/or the goodness of fit of the sine curves, readers could get some idea whether the uncertainty stems from the lack of data or from the dynamic nature of the catchment. This can be done in a separate table in the methodology, or in the supplementary material.

**Implications of problem statement/results could be elaborated.** This paper provides a perceptual model that addresses a clear knowledge gap in existing literature. However, the paper fails to link the implications of the results in the wider context of this field. Sure, the knowledge of high-altitude catchment processes is improved, but what does this mean for the environment? Existing literature (i.e. Hayashi 2019) point out that if high-altitude catchments have a large capacity for groundwater storage, the flow dynamics would respond differently to climate change than previously thought, with consequences to downstream settlements. Reflecting their results in this larger scope may give more importance to the study by putting their results in context.

**Significance of results compared to existing knowledge.** While the perceptual model manages to synthesise the results in an effective way, a lot of the results gained have already been known (or could be inferred) when looking at existing literature. For instance, the strong relationship between $F_{bf}$ and $F_{yw}$ could be inferred by definition, while the fact that a significant portion of precipitation is stored in the seasonal snowpack at high-elevated catchments is also previously known. Hence, the authors should further stress the relevance of this study (e.g. using a new tracer-based empirical method to investigate catchment processes to explain "counterintuitive" evidence found in previous studies) in the aim and conclusions of the paper.

**Limited number and variety of catchments in dataset.** Though I appreciate the practical issues when it comes to gathering this much data, I feel that stronger, more significant conclusions could have been drawn with a larger dataset. As stated in activity 1, only 2 catchments have limestone-dominated bedrock, despite a large portion of the alps having such geology. Only 1 catchment had a significant coverage of a glacier (NBPV), despite them covering a large number of high-altitude alpine catchments. Indeed, NBPV was seen as an outlier in many of the correlations – would this have still been the case if more glacier-dominated catchments were included?

**Use of $F_{qd}$ to represent groundwater storage processes.** The paper uses $F_{qd}$ and $F_{bf}$ to represent the groundwater storage- related processes that may explain the variation of $F_{yw}$ with elevation. However, I find that the use of these variables have inherent shortcomings that limit the strength of the conclusions and the validity of the perceptual model. $F_{qd}$ represents only a limited part of the catchment geology responsible for groundwater flow, as suggested by the weak correlation between $F_{qd}$ and $F_{yw}$. This results clashes with those obtained by Arnoux et al., (2021) who found that quaternary deposits played an important role in groundwater storage in alpine catchments, though that study used a modelling approach instead of an empirical one. Additionally, Arnoux et al., (2020) investigated 13 catchments, 4 of which are included in the dataset of Gentile et al., suggesting that not all alpine catchments have quaternary deposits as a major store of groundwater. Hayashi (2020) hypothesised that groundwater storage in alpine catchments could be controlled by the

amount of quaternary deposits, the type of underlying bedrock or the presence of cracks and fissures in the underlying bedrock. This leads me to believe that the quaternary deposit coverage can only act as a first-order measure of geological groundwater storage in alpine catchments, since there can be large differences in the geological structures between various catchments, and how they function to store groundwater. As a result, the perceptual model lacks information on groundwater storage processes. Though the shortcomings of representing groundwater storage using $F_{qd}$ has been adequately explained in the discussion and conclusions, it is recommended that the authors explore other geological information as potential explanatory variables. For instance, bedrock type could have been added as an additional explanatory variable for a geological form of groundwater storage. Additionally, the depth of the deposits are not considered using this methodology, though that data may be difficult to obtain.

**Minor issues: line-by-line**

- Title – "young water fraction" instead of "$F_{yw}$" to make the paper accessible to readers unacquainted with the topic
- Line 22 – briefly justify the necessity of the proposed formal classification scheme
- Line 36 – "low $F_{yw}$ is found" instead of "this results in low $F_{yw}$"
- Line 70-71 – were these catchments above the $F_{yw}$-reset as well? If so, please specify to stress how the findings are "in line" with the previous one
- Line 82 – what is meant by "efficient groundwater recharge"? It may mean high rates of recharge, or that most precipitation is recharged with minimal runoff, making it ambiguous to readers
- Line 94 – here, "dynamic storage" is mentioned as a knowledge gap, but this concept is not referred to again in the rest of the paper, especially regarding whether this concept connects to how the explanatory variables were chosen. Perhaps mention dynamic storage in the discussion when discussing $F_{qd}$ results
- Line 104-105 – Briefly justify the intention to introduce the new classification system
- Line 129-130 – Why were these catchments included to the dataset? Presumably to better represent high-altitude catchments; justifying this would clarify readers about this section
- Line 154 – Rhaetian Alps (Rätische Alpen/Alpi Retiche)
- Line 155 – would avoid using "a good range" – instead perhaps "allows us to explore $F_{yw}$ variations in large range of geological and climatic conditions…"
- Figure 1 – the shading of the catchments are unclear. Perhaps use a cleaner basemap (black/white) so that each catchment colour stands out better (BCC was almost invisible).
- Table 1 – Column 3 shows average elevation, perhaps explain how the averaging was conducted (i.e. area-weighted)
- Line 171 – Would this sub-section not be better suited under section 2, since data on each study site was already being discussed?
- Line 205 – unsure what "according to volume of precipitation" implies; either elaborate, or state "accounting for volume of precipitation…"
- Line 222-223 – this may be better suited in the introduction to help justify the new classification system
- Line 233 – "ambiguities" instead of "tricky-points"
- Figure 2 – the source of the data (Staudinger et al., 2007) was already included in the caption, no need to repeat it in the legend
- Line 248 – Enjoyed reading the overall description in this section!
- Line 267 – Nice scrutiny of methods
- Line 311 – "allows *for* estimation"
- Line 325 – Section 4.1 appears to contribute little to the discussion/conclusions, perhaps it can be removed?

- Line 345 – "evenly" instead of "equally"
- Line 354-355 – Nice exploration of outliers!
- Line 367 – Perhaps further elaborate in the introduction/methods that slope will also be considered as an explanatory variable
- Line 370-371 – Good preliminary result, well-written to explain the subsequent steps in the data analysis
- Figure 5 – the dotted line appears to be some sort of average, clarify whether this is a mean or median
- Line 419 – nice finding (strong correlation between $F_{qd}$ and WFI), would be good to mention this in the discussion to support the choice of including $F_{qd}$ in the methodology
- Line 472-473 – nice wording to introduce perceptual model
- Line 482 – "…recharge. This…"
- Line 487 – "reduce*s*"
- Line 494 – "However, this effect…"
- Line 500 – "depending on elevation."
- Line 519 – Jansson et al. (2003) was not included in the reference list. Also, please review the paper to confirm whether the fact stated was indeed referred to in the paper
- Line 596 – "few streamflow events *are* characterised…."

---

## Author Comment (AC1)

*The work of Gentile et al. investigated the causes for young water fraction ($F_{yw}$) variations with elevation ($F_{yw}$ is low at high altitudes) in Alpine catchments. The study areas are 27 catchments in Switzerland and Italy. The authors proposed new criteria for catchment classification into different hydro-climatic regimes. To gain insight into the reason for $F_{yw}$ variations with elevation, this author used a new set of hydrological variables, namely the fractional snow cover area ($F_{SCA}$), the fraction of quaternary deposits ($F_{qd}$), and the fraction of baseflow ($F_{bf}$). In general, the idea of this paper about what drives $F_{yw}$ variations with elevations is novel and of interest for understanding the functioning of catchments in Alpine regions as well as for understanding flow and transport in this region and potentially in other areas. However, the methodology and results do not fully support this idea. The text was not well written. Please find my main comments and line-by-line comments below.*

**Dear referee #1,**

**We would like to thank you for the overall positive assessment and the numerous detailed comments, which will contribute to our manuscript's improvement considerably.**

**Please find below a point-by-point response to both your main and minor comments. We will incorporate all your constructive feedback once we receive the editor's response.**

**Sincerely,**

**The Authors**

**Main comments**

- *Why did the authors need to propose a new criterion for catchment classification? The authors used two variables: (1) streamflow ratio between different months and (2) snow cover fraction for the proposed catchment classification, but later they adjusted the threshold of these two variables to have consistent results with Staudinger et al. (2017). Why didn't they just use the method of Staudinger et al. (2017)?*

- **We propose a new criterion for the regime classification because our dataset includes catchments outside the Swiss borders (i.e., the four Italian catchments) for which the Weingartner and Aschwanden (1992) and Staudinger et al. (2017) classification scheme cannot be strictly applied since they were designed for the Swiss hydro-climatic regimes. We "manually calibrate" the thresholds of $F_{SCA}$ and $Q_{June}/Q_{DJF}$ for classifying catchments in "rainfall-dominated", "hybrid" and "snow-dominated" as in the work of Staudinger et al. (2017). In this way, the classification scheme is "calibrated" on the Staudinger et al. (2017) catchments and we can apply it also outside the Swiss borders. According to the referees' comments, we will consider the possibility of modifying the classification scheme to make it more straightforward to link to previous classification (e.g., using streamflow and topographical data), but it will remain transferable to other regions.**

- *The objective is to investigate what drives $F_{yw}$ variation with elevation. The authors proposed using a new set of hydrological variables, but what are the relations between these variables with elevation? For example, what are the relations between $F_{SCA}$, $F_{qd}$, $F_{bf}$ with elevation? With $F_{SCA}$, I can infer from the text, but it was not explained in the text until the last sections (Section 5.2) of the manuscript. $F_{SCA}$ cannot be directly related to elevation, instead, it needs to be related to the catchment classification then from catchment classification to mean elevation. However, in other areas, can we still relate $F_{SCA}$ to elevation? With the other variables ($F_{qd}$ and $F_{bf}$), it is unclear to me what are their relations to elevations. In addition, $F_{qd}$ does not seems to be a good variable because there is no significant relation between $F_{yw}$ and $F_{qd}$.*

- **Thank you for this comment: this is a good point. We will add, for each variable ($F_{SCA}$, $F_{qd}$ and $F_{bf}$), a figure that shows the relation with mean catchments elevation. The three figures are reported here below, and we will include them in the revised manuscript.**

a) **The $F_{SCA}$ increases with the mean catchment elevation in our data set, revealing a positive, statistically significant correlation. This suggests the increasing snow cover persistence at high altitudes.**

b) **$F_{qd}$ decreases with the mean catchment elevation in our data set, revealing a negative, statistically significant correlation. This negative correlation reflects the fact that $F_{qd}$ decreases when the mean slope increases (Arnoux et al., 2021) (mean slope increases with mean elevation for the catchments analyzed in this study, as shown in Fig. 4a of the manuscript). We have decided to use $F_{qd}$ because Arnoux et al. (2021) demonstrated a strong positive correlation between $F_{qd}$ and Winter Flow Index (WFI) highlighting the role of unconsolidated deposits in storing groundwater (in terms of age, old water). The missing information about the portion of fractured bedrocks, the thickness of quaternary deposits and the bedrock topography will demand future attention for a complete picture of the role of geology (potentially resulting in a statistically significant correlation with $F_{yw}$).**

c) **$F_{bf}$ reveals an opposite behavior with respect to $F_{yw}$: it decreases until 1500 m and it increases at higher elevations.**

[Figure]

[Figure]

*Figure 1 a) $F_{SCA}$, b) $F_{qd}$, c) $F_{bf}$ and $F_{yw}$ against mean catchment elevation*

- *The manuscript needs to be restructured and revised. There is a lack of clarification in the text. More description of the study area characteristics is needed. Much of the information provided in Study Sites, and Material and Methods is not relevant (e.g., shape file, detailed source of data, etc.). Instead, citing the sources of the various data (both from individuals and organizations) can be moved to either the Authors' Contributions or Acknowledgements, or in the supporting information Sections or to a table rather than describe them within the text of the article, making it very difficult to read such detailed information. If possible, I would also suggest the authors publish their data in an open repository.*

- **Thank you for these suggestions. We will revise the "Study sites" and "Material and Methods" sections accordingly. We will move all the data sources in the "Data availability" section and remove irrelevant information. We will describe the study sites in a more concise manner using a Table and some descriptive figures: e.g., mean slope against mean elevation, mean annual precipitation and mean annual discharge against mean elevation, variations of mean monthly flow with elevation. These changes should make the text more fluent.**

**Minor comments**

*Title: "$F_{yw}$" could be changed to "young water fraction" for general readers.*

**Thank you for suggesting this. We will change "$F_{yw}$" in the title.**

*L14: "The young water fraction ($F_{yw}$),..., is increasingly used in hydrological studies, replacing the widely used Mean Transit Time, which is subject to aggregation error." This sentence provides misleading information. I think $F_{yw}$ cannot replace Mean Transit Times (MTT) since the two characterize different aspects of the transit times, e.g., $F_{yw}$ contains information about the younger part of the TT distribution (how much water in outflow is younger than 0.2 years) while MTT contains*

*information about the whole TT domain. "Aggregation error" could be changed to "aggregation bias".*

**Thanks. Our statement was indeed not precise. We wanted to say that, before the work of Kirchner (2016 a,b), the Mean Transit Time (MTT), obtained with convolution, was used in catchment intercomparison studies. After that key paper, it is generally replaced with $F_{yw}$ that, of course, does not provide the same information as MTT. To avoid misunderstanding, we will change the sentence simply by writing: "The young water fraction ($F_{yw}$), defined as the fraction of catchment outflow with transit times of less than about 2-3 months, is increasingly used in hydrological studies. The use of this new metric in catchments inter-comparison studies is helpful to understand and conceptualize the relevant processes controlling catchments' hydrological function."**

*L33-34: The sentence "..Fbf, considered…complement of $F_{yw}$" does not clearly show the relation you found between $F_{yw}$ and the baseflow fraction. Please be clearer about what you mean by explicitly saying that $F_{bf}$ is a good proxy for $F_{yw}$ as the higher $F_{yw}$ is, the lower $F_{bf}$.*

**Thank you for this recommendation. We will explicitly say that $F_{bf}$ is a good proxy of $F_{yw}$.**

*L44: "the streamflow is older than the annual snowmelt" is not clear to me, what is the age of streamflow and the age of snowmelt water in this case?*

**We will clarify this sentence: "However, early work in the Swiss Alps showed that high celerity is caused by a massive meltwater infiltration that pushes out groundwater reserves: streamflow following snowmelt is older than meltwater infiltrated in the current year (Martinec, 1975)."**

*L46: why "even"? I would expect exactly that during the absence of rainfall and snowmelt the streamflow is mainly sustained by groundwater.*

**We wanted to underline that the hydrograph separation results show that the hydrograph is generally mainly composed of old water at the peak flow. Of course, during no-rain and no-snowmelt periods we expect that streamflow is mainly sustained by groundwater and this is also confirmed by our results. We will change "even" with "especially" to be clearer.**

*L46-50: The two sentences here do not seem to be connected, one about residence time and the next one about transit times.*

**Of course, the transit time distribution and the residence time distribution are two separate concepts. Nevertheless, the streamwater age is influenced by the storage age depending on how much the storage contributes to the stream. However, we will separate the two sentences to avoid confusion.**

*L53: "Kirchner (2016a, b) proposed a new metric to quantify water age at the catchment scale". I think you are mentioning the $F_{yw}$, I don't think this is "the water age at the catchment scale" but the amount of water with age < 0.2 years. How can we know the "water age at the catchment scale" only based on the amount of water in outflow (discharge) that is < 0.2 years? For example, if $F_{yw} = 0.2$, what is the "water age at the catchment scale"*

**We wanted to say that MTTs, obtained with the classic convolution approach, are no longer used since they are subject to the aggregation bias. However, we can say something reliable about water age using a new metric: the young water fraction, that is calculated at the**

**catchment scale. Of course, $F_{yw}$ is not giving the same information of MTT. We will rephrase the sentence: "Kirchner (2016a, b) proposed a new metric to quantify the share of catchment outflow with transit times lower than 0.2 years: the young water fraction ($F_{yw}$)"**

*L55-58: please revised the sentence structure*

**We will revise the sentence structure: "It can be conveniently inferred from the dampening effect that a catchment has on the seasonal cycle of stable water isotopes in precipitation, i.e. by estimating the amplitude ratio of the seasonal cycle of stable water isotopes in streamflow to that of precipitation (Kirchner, 2016a). The $F_{yw}$ concept is increasingly used in hydrological studies because it has the advantage of being free from the aggregation errors inherent to Mean Transit Time (MTT) obtained through the convolution approach (Kirchner, 2016a)"**

*L58-59: please see my comments on line 14*

**We will adapt following our answer to line 14.**

*L70: "In line with these findings" can be removed because Lutz et al. (2008) did not state that $F_{yw}$ above 1500 m decreases*

**Yes, you are right. Lutz et al. (2018) did not state that $F_{yw}$ decreases above 1500 m a.s.l., but they said "In agreement with the results from the global study of European catchments, there is a slight tendency toward smaller $F_{yw}$ values for the subcatchments in the mountainous region". Therefore, to be more precise, we will substitute "In line with these findings" with "Moreover".**

*L82-83: "…more efficient groundwater recharge, consequently reducing or increasing the young streamflow…" It is not clear to me, should it be "reducing" only instead of "reducing or increasing"?*

**Ceperley et al. (2020) said: "our highest elevation study site (NBPV) deviates from this trend by yielding a higher $F_{yw}$, it is too early to draw the conclusion that low $F_{yw}$ could be due to seasonal versus intermittent snow cover dynamics alone." So, we remain vague in the introduction saying "reducing or increasing". From our results presented at the end of the manuscript, we can say that seasonal snow cover favors the groundwater storage emptying during winter and the groundwater storage recharge (because of meltwater infiltration) during summer, thus reducing the young streamflow reaching the stream.**
**We will put "reducing" partially anticipating the results.**

*L88: "…remarkable fraction of groundwater…" it is a bit vague, could you please be more precise?*

**To be more precise, we will specify the average percentages of groundwater according to the cited works: "Several catchments located in the Rocky Mountains and Andes mountain ranges show that , on average, about 47% of groundwater annually sustains the streamflow (Saberi et al., 2019; Somers et al., 2019; Carroll et al., 2018; Harrington et al., 2018; Cowie et al., 2017; Baraer et al., 2015; Gordon et al., 2015; Frisbee et al., 2011; Liu et al., 2004; Clow et al., 2003; Baraer et al., 2009). Similar result is also found in the Himalayas (49%) and Alps (48%) mountain ranges (Chen et al., 2018; Engel et al., 2016; Käser and Hunkeler, 2016; Williams et al., 2016; Wilson et al., 2016; Andermann et al., 2012)"**

*L91-92: "…a dynamic storage contribution to streamflow…" Please clarify this term.*

**We will rephrase this sentence: "There is however still a lack of understanding regarding the mechanisms that lead to a rapid mobilization of old water during storm events and a variable chemical signature depending on the flow regime (Kirchner, 2003). One key to understanding these mechanisms is the concept of dynamic storage, i.e., the storage that controls the streamflow dynamics."**

*L99: Why don't the authors use a new set of hydrological variables ($F_{SCA}$, $F_{qd}$, $F_{bf}$, WFI) in combination with traditional variables to gain new insights into the $F_{yw}$ along elevation gradients?*

**We do not know what is meant by "traditional" variables. However, we decided to use variables that were not previously considered for explaining $F_{yw}$ elevation gradients. This is also because Jasechko et al. (2016) wrote: "Although topographic gradient provides the strongest correlation with young streamflow fractions in our data set, the fraction of unexplained variance is large, suggesting that other variables also play a significant role. We observe no significant correlation between the young streamflow fraction and catchment size, annual precipitation, bedrock porosity, population density, or the fraction of catchment area comprised of pasture land or open water". We will clarify this in the revised version.**

*L104: "...into three hydro-climatic regimes proposing a new criterion of classification..." Why? I think a brief explanation is needed.*

**Please see our answer to your first main comment. We will add a brief explanation.**

*Sections 2 and 3.1: Both sections about the data (e.g., Section 2: existing data, additional dataset, complete data, and Section 3: discharge data and catchment boundary), why do the authors need two different sections? The data description section (entire section 2) needs to be restructured and revised to make it more concise and clearer. I think this can be done using a table. In the text, the authors could summarize and report key information, so the reader does not have to search through the many sources you have cited. The authors can here focus more on catchment attributes, such as climate (e.g., average annual precipitation and discharge), land use cover, geology, and discharge.*

**Thank you. The structure suggested can improve the paper readability. We will restructure Section 2 according to your comments. We will condense existing data, additional dataset and complete data sections in a single section. Moreover, we will move all the data sources reported in Section 3 to the "Data availability" section. We will improve the study site description summarizing the main topographic and hydro-climatic quantities in a table and some figures (please see our answer to your third main comment). Furthermore, we will summarize the description of this information in a paragraph that will be more fluid for the reader that will find all the relevant information about the study sites in a single section.**

*"Furthermore, 21 out of the 22 … (Staudinger et al., 2017)". This part is not relevant in my opinion.*

**You are right. We will remove this part.**

*"Two high-elevation catchments … Arnoux et al., 2021)". This part is not relevant in my opinion.*

**You are right. We will remove this part.**

*L147: In my opinion, the "Complete Dataset" subsection is not necessary. It is sufficient to illustrate the existing data in subsection 2.1 and conclude the section with 2.2.*

**Good idea. We will do as you suggest.**

*L156-160: Like von Freyberg (2018) … are reported in Table 1. If subsection 2.3 is deleted, move it to 2.2 as the final sentence.*

**Yes. We will do as you suggest.**

*Figure 1: the background cannot be easily seen; I think you could replace with a DEM map. In addition, I cannot differentiate between Quaternary deposits and hybrid catchments visually.*

**We will change Fig.1 according to your suggestions. We will use a DEM map as background, and we will change the quaternary deposits and catchments colors to make them clearly distinguishable. A first attempt of the new Fig.1 is reported here below:**

[Figure]

*Figure 2 First attempt for the new Fig. 1*

*Table 1: I am curious to see the relation between average elevations and average slopes for the 27 catchments, is there a positive correlation? (also for average elevation with annual precipitation)*

**We will insert these two figures in the "Study sites" section and we will comment on them. Average slope ranges from 4° to 34° and our study sites reveal an increase of steepness with elevation showing a positive correlation. Precipitation increases with elevation until 1500 m a.s.l. and it decreases for higher elevations highlighting a change of precipitation regime as described by previous studies (Santos et al., 2018)."**

*Section 3.1: Here, I would expect more description of the discharge dynamics (e.g., giving an order of magnitude to these data by telling what is the annual discharge, whether the runoff is seasonal, etc). I would suggest moving the description of how discharge was measured and derived to the*

*appendix. The source of data could be combined into the same table suggested for section 2 (or move to the appendix or data availability section).*

**Thank you for these suggestions. As said before, we will move all the info about the data sources to the "Data availability" section and we will describe in detail discharge dynamics in the "Study sites" section.**

*Line 190: I suggest mentioning the study period for the isotope data and $F_{yw}$ for the different study catchments since it is different.*

**Thank you for this recommendation. The table reported in the "Study sites" section could be a good place for inserting this info.**

*Figure 2: In summer there is a higher average monthly flow from snow-dominated catchments than from rain-dominated ones (due to increased snowmelt, I suppose), and in winter it is the other way around. Please explain this better in the text because it is not clear. In addition, it is not easy to differentiate between the three boxplots, I would suggest having three separated boxplot figures with the same y axis limit. This figure should be described in the text (there is no description of this figure, it was only cited in line 243)*

**We will subdivide the figure in three subplots and we will explain it better in the text. Specifically, snow-dominated catchments reveal a higher average monthly flow during summer than during winter due to the increased snowmelt. This difference is less marked for hybrid catchments due to a quite homogeneous distribution of flow over the summer and winter seasons: this is because the contribution of both rainfalls and snowmelt events are relevant for streamflow generation processes of these catchments. Finally, rainfall-dominated catchments show a higher average monthly flow during winter than during summer because of the almost total absence of summer (i.e., delayed) snowmelt.**

*L197: no comma after "where"*

**Thank you.**

*L221: As I understood from the text (before and after this line), there is indeed a "formal" classification method*

**Yes, there is a formal classification method proposed by Weingartner and Aschwanden (1992), but it was designed for Switzerland. The regimes defined by Weingartner and Aschwanden (1992) were grouped by Staudinger et al. (2017) in three categories: rainfall-dominated, hybrid and snow-dominated.**

*Section 3.3: After reading the entire manuscript up to section 3.3. I am not clear why the authors need to classify streamflow into three regimes and why the classifier should be based on snow-related characteristics (e.g., snow cover area).*

**Please see our answer to your first main comment. Moreover, we think that a classification that uses snow-related characteristics is suitable since what drives the regime changes is the increasingly important role of snow.**

*L240: should it be "it is expressed in mm per unit area and time step"?*

**We will rephrase the sentence: "Considering the data set investigated by Staudinger et al. (2017) as a starting point, we compare the monthly streamflow (flow is relative to catchment area: i.e., it is expressed in mm per time step)..."**

*L251: "...more than weekly…" do you mean biweekly?*

**We will rephrase: "Temporally, this relatively recent satellite has increased the visitation frequency to a sub-weekly temporal resolution…"**

*Eq: (5) the denominator ($N_{tot} - N_{clouds}$): This could result in an overestimation of $f_{SCA}$. What is the maximum fraction of cloud cover in these images?*

**We follow the approach of Hoffmeister et al. (2022) and Di Marco et al. (2020) that define $f_{SCA}$ as $N_{snow}/(N_{tot\text{-}clouds})$. They did not comment on the effect (i.e., underestimation or overestimation) of the mathematical expression of $f_{SCA}$ on their results. If the two detection algorithms (snow detection and cloud detection) would work with a 100% accuracy, values greater than 1 cannot be encountered. In fact, the maximum cloud cover fraction can also be very close to 1 in some dates (e.g., > 90% as encountered in our data set), but if the snow detection algorithm works well $N_{snow}$ will be at most "complementary" to $N_{clouds}$ (i.e., $N_{snow} + N_{clouds} = N_{tot}$) and $f_{SCA}$ will be at most 1. Sometimes these algorithms can result in a misclassification and $f_{SCA}$ values > 1 can be encountered: i.e., $N_{snow} > (N_{tot}\text{-}N_{clouds})$. Our approach was to set $f_{SCA} = 1$ if $f_{SCA} > 1$. The authors will deepen if this approach can overestimate the $f_{SCA}$ and, if so, we will think about the application of an alternative approach.**

*L276-279: The error in $f_{SCA}$ is still there with the "moving window" approach, it is just smoothed. Anyways, at the end, you calculated the average $f_{SCA}$ over the whole period so applying "moving average on a window" does not have any effect?*

**Of course, the average of the series after the application of the moving average (that is what we call $F_{SCA}$) is not the same as the average over the original series.**

*L282-289: "Some authors have revealed … $F_{yw}$ in Alpine catchments". This is more suitable for Introduction than Methodology. In addition, what is "key possibility"? Does it mean "high possibility"*

**Thank you for this comment: it is a good idea; we will move this part to the Introduction. With "key possibility" we wanted to express "the importance" of quaternary deposits (moraines, alluvium, and talus) in storing groundwater. We will change this expression.**

*L292: … 23 Swiss catchments … Is $F_{qd}$ calculated only for 23 sub-catchments, while WFI and $F_{bf}$ for all 27? Why? How does it affect the interpretation of the results? Be clearer about which indices are available for each study site.*

**We have written: "Operatively, for the 23 Swiss catchments of our dataset, we calculate the portion of the catchment area occupied by quaternary deposits using the Geological Atlas of Switzerland…" The number "23" refers to catchments located in Switzerland because, for these catchments, we have used the GeoCover dataset for estimating the area covered by quaternary deposits; for the remaining four Italian catchments we have used regional geological dataset. $F_{qd}$ , WFI and $F_{bf}$ are calculated for all the 27 catchments.**

*L299-301: For the DOR and SOU ... provided by Dr. Giulia Zuecco. This part is not relevant here, should be moved to the data section.*

**Thank you for this recommendation. We will move this part to the data section.**

*L315-318: "For VdN, NBPV and BCC catchments we consider the time windows ... we consider discharge measurements in the period November 2017 - January 2022". I think you should indicate at the beginning the different study periods, because it is confusing to read a lot of data (e.g., stable isotopes of water, $F_{yw}$, streamflow...) and indices (e.g., $F_{qd}$, WFI, $F_{bf}$) for your methodology and find out that your study areas were analyzed in different periods. You should say this explicitly each time you mention a new data item or index or create a table in which you explain it.*

**Thank you for this recommendation. We will insert this info in the Table reported in the "Study sites" section.**

*Section 4.1. I think this can be moved to the data section or supporting information, as this is only for 2 catchments.*

**Thank you for this suggestion. We will move this part to the Supplementary Material.**

*L334-335: "these have the same names as the ones proposed by Staudinger et al. (2017 but the classification is not based on the same criteria" why? I think should be explained earlier in the methodology section.*

**Please see our answer to your first main comment. We will explain this point earlier in the methodology section.**

*L336-337: "In order to achieve a classification as consistent as possible with that of Staudinger et al. (2017), but based on these two variables, we propose the thresholds presented in Table 2:" I cannot understand why. If the authors want to have consistent results with Staudinger et al. (2017), why did not they use the method proposed by Staudinger et al. (2017)?*

**We propose a new criterion for the regime classification because our dataset comprises catchments outside the Swiss borders (i.e., the four Italian catchments) in which the Weingartner and Aschwanden (1992) and Staudinger et al. (2017) classification scheme cannot be strictly applied since they were designed for the Swiss territory. Therefore, we decided to design a new classification scheme based on $F_{SCA}$ and $Q_{June}/Q_{DJF}$. We "manually calibrate" some thresholds of $F_{SCA}$ and $Q_{June}/Q_{DJF}$ for classifying catchments in "rainfall-dominated", "hybrid" and "snow-dominated" as in Staudinger et al. (2017). In this way we "calibrated" the classification scheme on Staudinger et al. (2017) catchments and we can apply it also outside the Swiss borders.**

*L345-346: "Following our classification scheme, ... and 9 snow-dominated catchments". How do you compensate for the fact that the catchments data belong to different periods?*

**$F_{SCA}$ is calculated for all the catchments in the same period (2017-2021). $Q_{June}$ and $Q_{DJF}$ are the average values of June discharge and the sum of December,January and February discharge for periods that can be different from one site to another. We suppose that by calculating the average values we are capturing the typical $Q_{June}$ and $Q_{DJF}$ so that the classifiers are comparable for the different sites, also if they are referred to different periods.**

*L353: "snow-regime" should be explained here*

**We have written:" This result suggests that the easy-to-calculate $Q_{June}/Q_{DJF}$ ratio is a good predictor of the snow regime, here represented by $F_{SCA}$". The term snow-regime is here used to describe the snowpack persistence. In fact, we consider the snowpack persistence as represented by $F_{SCA}$.**

*L354: "for the first order estimate of the second classifier" what does it mean?*

**It means that, inverting the exponential relationship we have obtained (indicated in Fig.3 of the manuscript), it is possible to calculate $F_{SCA}$ from $Q_{June}/Q_{DJF}$.**

*"Section 4.3: New explanatory variables for the $F_{yw}$ elevation gradients" I would expect all subsections in section 4.3 will use variables that are related to elevation to explain the relation between $F_{yw}$ with elevation. However, I cannot see what is the relation between the variables in the section title (e.g., Section 4.3.1. Fractional Snow Cover Area ($f_{SCA}$) and $F_{yw}$) and elevation (Please also see my main comments)*

**We will follow your suggestions and we will insert, for each variable, a figure that relates the variable with elevation.**

*L361-368: part of this information was already described in the introduction, can be removed here or merged into the introduction.*

**Thank you for this suggestion. We will merge this part in the Introduction.**

*L389-391: Our results ... for hybrid catchments (median $F_{yw}$ of 0.32) ... Why are there these differences? I suggest arguing and explaining them.*

**We have structured the manuscript so that in the "Results" section we minimally discuss the results that are explained and argued in the "Discussion" section. In fact, the difference in median values of $F_{yw}$ for hybrid and snow-dominated catchments can be explained considering the perceptual model reported in Section 5.2. However, we will add an explanation referring to the differences about these median values.**

*Figure 4a can be moved to the data section, figure caption: "the horizontal bars correspond to +/- standard deviation" of slope or elevation?*

**Thank you for the suggestion. We will move the figure to the data section. The horizontal bars correspond to +/- standard deviation" of slope.**

*L367: "Despite this" why should an increase in slope with elevation result in a correlation between Fwy and slope?*

**Thank you for this comment. This is probably a typo. We could expect a negative correlation between $F_{yw}$ and slope because of the results of Jasechko et al. (2016): "...topographic gradient provides the strongest correlation with young streamflow fractions in our data set..."**

*L393: "lowering" could be changed to "decreasing of $F_{yw}$ with increasing $F_{SCA}$"*

**Thank you.**

*L408: Why were the two catchments with $F_{qd} = 0$ is excluded? why do the group need to have features as close as possible to those used by Arnoux et al. (2001)*

**We are assuming that groundwater is stored by unconsolidated sediments and we want to understand the role of unconsolidated sediments in modulating $F_{yw}$ elevation gradient. If we include catchments with $F_{qd} = 0$, we are biasing our analysis since, for these catchments, of course $F_{qd}$ cannot have a role.**

*Section 5.1. I would expect here a discussion about the advantages of the new classification compared to other approaches (e.g., Staudinger et al., 2017), especially when the focus of the study is to understand what drives $F_{yw}$ variations with elevations. The text written in this section does not seem to be relevant to this study.*

**Thank you for the suggestion. We will discuss the advantages of the new classification reported in the answer to your first main comment (i.e., it can be applied also outside the Swiss borders and the $F_{SCA}$ can be estimated from $Q_{June}/Q_{DJF}$ inverting the equation obtained in this study). We will consider moving this part to an Appendix or Supplementary Material.**

*L473: How does your work harmonize with previous results? I suggest expanding this point by making it clearer and highlighting the novelty of your work compared to the previous studies.*

**Thank you for this suggestion. We will explain better how our results harmonize with previous studies and how our work sheds light on the literature gap about the processes hidden behind low $F_{yw}$ values in high-elevation catchments. Section 5.2 is rich of references to past works that support our perceptual model and the reviews to the present manuscript gave us further suggestions about the linkages of our study with previous findings: they will be integrated in the manuscript.**

*L477: "increase of precipitation and slope with elevation (Fig.12a, Fig. 4a)" I cannot see this in these figures*

**Precipitation is indicated by the point color (with the relative colorbar). However, we will add a plot of annual precipitation against elevation.**

*L483: higher up sounds odd. Simply say upstream.*

**Thank you.**

*L484-486: Therefore, it is more likely that ... possibly ephemeral, snowpack. I do not see a connection between these two statements. If you are saying that lower-order (i.e., more downstream) channels release greater amounts of old water than higher-order (i.e., more upstream) channels, why do you say that water age decreases with elevation? Please clarify this point.*

**This sentence refers to the discussion about the rising limb of the $F_{yw}$ vs $F_{SCA}$ bell-shaped relationship. In other words, for rainfall-dominated and hybrid catchments with ephemeral snowpack, water age decreases with elevation:**
- **The increase of precipitation with elevation and the reduction of evaporation with elevation, due to reduced temperatures, promote wetness conditions that increase $F_{yw}$.**
- **Considering the Strahler's stream order, lower order channels, upstream, are more rarely activated (e.g., because of intense rainfall/snowmelt events) draining young water.**

> Vice versa, higher order channels, downstream, are more often active (e.g., because of low/medium rainfall/snowmelt events) and inclined to drain more old water.

- The limited number of snowfall days and the mid-winter melt (due to an ephemeral snowpack) reduce the snow accumulation. Such a snowpack does not protect the underlying soils from freezing thereby inhibiting infiltration and favoring rapid flow paths during mid-winter melt/rainfall events, with subsequent increase of $F_{yw}$.

*L493: "a persistent, deep snowpack can promote deep vertical infiltration by insulating the soil and thereby preventing freezing" do you mean this happens in winter? If in winter, there might be only snow, how can it be melted and promote deep vertical infiltration? Where is the source of water for vertical infiltration?*

**The persistent and deep snowpack prevents soil freezing during winter so that during snowmelt onset in spring, meltwater can infiltrate and recharge the groundwater storage. This will be made clearer.**

*L495: what's a temporal concentration? Make it clearer.*

**We have written: "The resulting effect on water partitioning between the surface and the subsurface has however to be analyzed in light of the temporal concentration of water input on the snowmelt period and remains largely unexplored to date (Rey et al., 2021)"**

**Temporal concentration: the time-interval in which the snowmelt enters the system as water input.**

*L499-501: This is for the karst area, how relevant is it for your area?*

**In our dataset we have two dolomitic catchments: BCC and OVA. We will specify this in the text.**

*L518: I suppose the fast flow paths are due to the fact that the glacier acts as an impermeable layer and thus promotes rapid overland flow? Please explain what you mean.*

**This comment refers to a possible explanation of the high $F_{yw}$ for the glacier-covered catchment. The current text reads as "Such (glacier-covered) catchments could show fast flow paths and small storages as e.g. discussed in the work of Jansson et al. (2003), reviewing glacier-dominated environments. Moreover, reduced baseflow during winter can be related to increasingly high temperatures causing the glaciers retreat, thus reducing and anticipating the glacier melt fluxes that possibly recharge groundwater (Hayashi, 2020)". We will rephrase to:**
**"The high $F_{yw}$ of the high elevation glacier-covered catchment can be explained considering that the glacier-melt produces high amounts of streamflow that transit the glacier-system very quickly during the summer and only limited water storage capacity in the glacier forefield (Muller et al. 2022). Accordingly, fast flow paths and small storages were described reviewing glacier-dominated systems (Jansson, 2003, Ceperley et al. 2020). Schmieder et al. (2019) also found a high $F_{yw}$ in an Austrian glacier-covered (35%) catchment leading to the conclusion that the basin behaves like a 'Teflon basin' with fast transmitted ice melt, also if this behavior is distributed in space (i.e., part of the catchment defined "sponge" behaves differently delaying the release of water).**

*Figure 13: Which subfigure is for lower altitudes (< 1500m) and which one is for higher altitudes? Figure caption: the word "panels" can be removed because I thought a panel always consists of two subfigures (e.g., the lower panel contains two subfigures c,d)*

**Subfigures titled with "Ephemeral snowpack" refer to lower elevations, while subfigures titled with "Seasonal snowpack" are for higher elevations. We will remove the word "panels" as suggested.**

L531: "unconsolidated sediments are not the only…" could be changed to "water storage in unconsolidated seidments are not the only …"

**Thank you.**

---

## Author Comment (AC2)

*This review was prepared as part of graduate program Earth & Environment (course Integrated Topics in Earth & Environment) at Wageningen University, and has been produced under supervision of dr Ryan Teuling. The review has been posted because of its potential usefulness to the authors and editor. Although it has the format of a regular review as was requested by the course, this review was not solicited by the journal, and should be seen as a regular comment. We leave it up to the authors and editor which points will be addressed.*

**Dear Peter Jansson and Ryan Teuling,**

**we would like to thank you for having selected our paper for this class. The input is highly appreciated and will help to improve our paper. For simplicity, we cut the first part, the overall impression section, of the comment in this response. But we thank the authors for the summary and the appreciation of our work.**

**We would like to point out here that we find the review extremely helpful but overly harsh at instances. The wording implies that we made mistakes. While this wording style is omnipresent (unfortunately!), we take the opportunity to call for a change in wording style in scientific reviews. A simple example is the following: "the authors should further stress the relevance of this study" could be reformulated to "the study would gain from a more concise presentation of its relevance".**

**Please find below a point-by-point response to both your general and specific comments.**

**With kind regards,**

**The Authors**

**Overall impression**
(..)

*Another useful finding of this paper is the introduction of a new, formal classification of alpine catchments into 3 hydro-climatic regimes. Though similar classification systems have existed, the system proposed in this paper used formal and objective criteria, making it suitable for cross-catchment datasets even beyond the alps. This classification system also includes a new proxy for estimating snow coverage using only discharge data – $Q_{june}/Q_{DJF}$. This can be a very useful tool for future research looking to estimate snow cover requiring the necessary satellite data to estimate $F_{SCA}$ directly.*

**Thank you for this comment. As detailed in our answer to reviewer 1 and reviewer 2 (J. von Freyberg), we might consider modifying the classification scheme to make it more straightforward to link to previous classification. But it will remain transferable to other regions.**

*At first glance, the results and conclusion appear impressive. Interpretations of the results were synthesised to address a relevant knowledge gap regarding the understanding of alpine catchment processes. However, the perceptual model proposed by the authors do not appear to be sufficiently backed up; the explanatory variables used may not accurately represent the catchment processes as the authors intended. In my view, the authors should adapt their methodology to limit the model shortcomings, while also investigating other explanatory variables which can give more credibility to their conclusions. Overall, I believe the paper requires significant revisions before it can be considered for publication.*

We would like to thank you for your critical comment. At this stage, this critical comment remains vague for us, and we do not provide an answer here, but to the detailed comments below.

**General Comments**

**We number the comments below for cross-referencing.**

*Use of Fbf to represent groundwater storage processes*

*[1] Arguably a larger weakness of the paper compared to the one above, $F_{bf}$ is by definition related to $F_{yw}$, so the correlation found may not be strong enough evidence to convincingly corroborate the perceptual model.*

**We agree that in snow-free systems, $F^*_{yw}$ ('*' indicates a flow-weighted variable) is by definition related to $F_{bf}$: baseflow is composed of groundwater and groundwater is the dominant source of old water in snow-free systems (in absence of large lakes). However, in snow-influenced systems, part of the old water is temporarily stored in the snowpack. This is at a first glance not measured by $F_{bf}$. We nevertheless show here that $F_{bf}$ obtained from streamflow alone (with the selected baseflow filter) leads to an approximately complementary relationship to $F^*_{yw}$ ($F^*_{yw} + F_{bf} \simeq 1$), which is an important result for catchments where we do not have isotope measurements. Why is this so? Simply because the snowmelt in snow-dominated systems largely transits through the groundwater store and leads to high baseflow. This will be made much clearer in the revised version.**

*[2] It could already be reasonably assumed that streamflow with higher mean residence times would indicate a slower flow path. Additionally, $F_{bf}$ does not directly infer the type of catchment storage; catchments with high snow storage and low groundwater storage may also yield higher $F_{bf}$.*

**Yes, we agree that if snow is being stored for a long time on the surface, it will also lead to high shares of old water. However, if potential groundwater storage is low, the snowmelt cannot transit through the groundwater storage, i.e., it flows off quickly and cannot lead to a high baseflow. As mentioned in our response to J. von Freyberg, high baseflow in summer (and thus high $F_{bf}$ in high elevation catchments) means that snowmelt is transiting via groundwater. We will make this very clear in the paper.**

*[3] Baseflow could also account for a myriad of other catchment characteristics, such as the mean slope and shape coefficient. Hence, the assumption made by the authors that $F_{bf}$ could be used as a proxy for groundwater storage processes is questionable.*

**We do not understand this comment. Baseflow is "the slowest responding and longest lasting component of streamflow" and "it has been described as flow from groundwater storage" (Duncan, 2019 cum bibl.) In absence of big lakes, it is simply the groundwater released to the stream.**

*[4] As a result, the perceptual model and conclusions may have been supported with the wrong evidence. In line 549 the authors state that the inverse correlation between $F_{bf}$ and $F_{yw}$ "clearly indicates that the observed patterns of $F_{yw}$ are related to water partitioning between the surface and*

*subsurface". However, $F_{bf}$ may be inferring to other processes, inferring the need for cross-validation.*

**Thanks for pointing this out, we agree that we took a shortcut here not explaining the link between $F_{bf}$ and groundwater storage explicitly. However, we do not think that $F_{bf}$ might be inferring any other processes besides groundwater (in absence of big lakes).**

*[5] It is hence highly recommended that the authors run additional correlations to identify how $F_{bf}$ links to groundwater-related processes. Figure 8 a) shows such an example, where $F_{qd}$ is shown to be positively correlated with WFI. This helps justify the inclusion of $F_{qd}$ and WFI as explanatory variables that act as proxies for groundwater flow. A correlation between WFI and $F_{bf}$ would hence illustrate the strength of $F_{qd}$ as a proxy for groundwater flow and storage. If WFI and $F_{bf}$ are indeed positively correlated, the use of $F_{qd}$ would be further justified. A correlation between $F_{SCA}$ and $F_{bf}$ would show the extent of snow-related processes being incorporated in $F_{bf}$.*

**Thanks for this comment. We will pay attention to better explain the link between all presented variables.**

*Trends with precipitation not analysed*

*[6] Precipitation can have a large effect on $F_{yw}$ variations: catchments with higher rainfall may have faster flow paths and hence higher $F_{yw}$. Figure 12a illustrates differences in precipitation levels, and seems to indicate an increasing trend between $F_{yw}$ and precipitation. That would indicate that precipitation should be included as an explanatory variable. Neglecting to include this variable further weakens the strength of the discussion and conclusion; the authors intended to investigate potential drivers of $F_{yw}$ variations with elevation, without including a variable that is known to affect catchment residence times and to vary with elevation. I therefore suggest that the authors include correlations between $F_{yw}$ and precipitation. Doing so would provide a more holistic view of the alpine catchment processes, and further enrich the perceptual model, conclusions, and the scope for further research in this new and exciting field.*

**Thanks for this comment, which is similar to a comment by J. von Freyberg. We copy here the answer from our response to J. von Freyberg:**

***"We decided to use variables that were not previously considered for explaining young water fractions variations; Jasechko et al. (2016) wrote: "Although topographic gradient provides the strongest correlation with young streamflow fractions in our data set, the fraction of unexplained variance is large, suggesting that other variables also play a significant role. We observe no significant correlations between the young streamflow fraction and catchment size, annual precipitation, bedrock porosity, population density, or the fraction of catchment area comprised of pasture land or open water".***
***We explain in the manuscript that, below roughly 1500 m, the increase of $F^*_{yw}$ with elevation also depends on the increase of precipitation with elevation. In fact, in such cases, annual precipitation can be considered as a proxy of catchment wetness since we mainly observe a liquid water input.***
***Above 1500 m, using mean annual precipitation as a proxy for catchment wetness is misleading because the seasonal snowpack leads to a very dry period of the year despite high (solid) water input. In other words, the total amount of precipitation is not the variable of interest, rather the temporal concentration of water input is the relevant variable. It is possible to observe the saturation of the system (i.e., high wetness conditions) also when annual precipitation is low if a large volume of water (stored in the snowpack) is released in a relatively concentrated time interval. After the long winter period, we expect high infiltration that recharges the groundwater storage.***

*This process can bring the system to saturation (high wetness) so that ultimately rain or snowmelt can more rapidly reach the stream as overland flow.*
*Additionally, Lutz et al. (2018), estimating the young water fraction for 24 catchments in Germany, have found exactly the opposite of what is generally thought: the young water fraction decreases with increasing mean annual precipitation. They stated that this result reflects "the impact of various factors relevant in the mountainous region, resulting in the decrease of young water fractions for higher-elevation catchments" (Lutz et al. 2018). Thus, what we did in our work is searching for the "various factors" that lead to low $F^*_{yw}$ in mountainous catchments not considering the mean annual precipitation as an explanatory variable."*

**Specific Comments**

*[7] From the methodology of the paper, a new hydro-climatic regime classification came about, including a new proxy variable that can be used to estimate $F_{SCA}$ with high accuracy ($R^2=0.99$). Though useful in strengthening the methodology, the authors only dedicate a small section of the discussion and a couple of sentences in the conclusion on this classification scheme. The development of the hydro-climatic regime classification is given a lot of attention to despite not being part of the research objectives. If the authors believe that this new method has potential in future studies, it is recommended that these methods could be mentioned more explicitly: either being incorporated in the abstract and treated as a research objective, or moving elaborations to the appendix or to a separate publication.*

**Thanks for this suggestion. We will decide on the final classification scheme during the work on the revised version and consider moving technical detail in the supplementary material.**

*[8] Isotope data timespan and resolution not stated. This uncertainty propagates to form some of the uncertainty in $F_{yw}$. However, by including the temporal data span/resolution, and/or the goodness of fit of the sine curves, readers could get some idea whether the uncertainty stems from the lack of data or from the dynamic nature of the catchment. This can be done in a separate table in the methodology, or in the supplementary material.*

**Thank you for this comment. We will add info about isotope timespan and resolution.**

*[9] Implications of problem statement/results could be elaborated. This paper provides a perceptual model that addresses a clear knowledge gap in existing literature. However, the paper fails to link the implications of the results in the wider context of this field. Sure, the knowledge of high-altitude catchment processes is improved, but what does this mean for the environment? Existing literature (i.e. Hayashi 2019) point out that if high-altitude catchments have a large capacity for groundwater storage, the flow dynamics would respond differently to climate change than previously thought, with consequences to downstream settlements. Reflecting their results in this larger scope may give more importance to the study by putting their results in context.*

**In answer also to J. von Freyberg comment, we will make the research objective and the relevance of the outcome much clearer in the revised version. We will also pick up the above suggestion to broaden the discussion, without becoming speculative. In fact, it is important to point out that the storage in the high alpine environments studied here is relatively large but this is a local effect. We cannot make statements about the role of this headwater storage at a more regional scale.**

*[10] Significance of results compared to existing knowledge. While the perceptual model manages to synthesise the results in an effective way, a lot of the results gained have already been known (or*

*could be inferred) when looking at existing literature. For instance, the strong relationship between $F_{bf}$ and $F_{yw}$ could be inferred by definition, (..)*

**As discussed in our response to comment [1] above, we respectfully disagree that this result was known before. Specifically, to the best of our knowledge, a complementarity between $F_{bf}$ (using a baseflow filter without parameters calibration) and $F^*_{yw}$ was never presented before. Moreover, in answer also to J. von Freyberg comment:** *"The potentially dominant water flow and storage processes driving young water fractions are of course discussed in previous work (Jasechko et al. 2016, von Freyberg et al. 2018, Lutz et al. 2018); to the best of our knowledge, this is the first attempt to present a perceptual model that harmonizes these known processes with the surprising low $F^*_{yw}$ values of high elevation catchments. This will be explicitly stated in the introduction."*

*[11] ... while the fact that a significant portion of precipitation is stored in the seasonal snowpack at high-elevated catchments is also previously known.*

**We do not think that we presented this as a new result.**

*[12] Hence, the authors should further stress the relevance of this study (e.g. using a new tracer-based empirical method to investigate catchment processes to explain "counterintuitive" evidence found in previous studies) in the aim and conclusions of the paper.*

**See also our response to comment [10], we will make this clearer.**

*[13] Limited number and variety of catchments in dataset. Though I appreciate the practical issues when it comes to gathering this much data, I feel that stronger, more significant conclusions could have been drawn with a larger dataset. As stated in activity 1, only 2 catchments have limestone-dominated bedrock, despite a large portion of the alps having such geology. Only 1 catchment had a significant coverage of a glacier (NBPV), despite them covering a large number of high-altitude alpine catchments. Indeed, NBPV was seen as an outlier in many of the correlations – would this have still been the case if more glacier-dominated catchments were included?*

**We would of course be extremely happy to include more case studies but this will only be possible once more data becomes available. Year-around isotope sampling in very high elevation environments is rare.**

*[14] Use of $F_{qd}$ to represent groundwater storage processes. The paper uses $F_{qd}$ and $F_{bf}$ to represent the groundwater storage- related processes that may explain the variation of $F_{yw}$ with elevation.*

**We would argue that we use them as proxies but not to "represent processes". We will carefully revise the manuscript to avoid misuse of terminology.**

*[15] However, I find that the use of these variables have inherent shortcomings that limit the strength of the conclusions and the validity of the perceptual model. $F_{qd}$ represents only a limited part of the catchment geology responsible for groundwater flow, as suggested by the weak correlation between $F_{qd}$ and $F_{yw}$. This results clashes with those obtained by Arnoux et al., (2021) who found that quaternary deposits played an important role in groundwater storage in alpine catchments, though that study used a modelling approach instead of an empirical one. Additionally, Arnoux et al., (2020) investigated 13 catchments, 4 of which are included in the dataset of Gentile et al., suggesting that not all alpine catchments have quaternary deposits as a major store of groundwater. Hayashi (2020) hypothesised that groundwater storage in alpine catchments could be controlled by the amount of quaternary deposits, the type of underlying bedrock or the presence of cracks and fissures in the*

*underlying bedrock. This leads me to believe that the quaternary deposit coverage can only act as a first-order measure of geological groundwater storage in alpine catchments, since there can be large differences in the geological structures between various catchments, and how they function to store groundwater.*

**Yes, you are exactly right about $F_{qd}$: it is a first-order measure, but the best that we can do for the catchments at hand. In answer also to reviewer 1: "_The missing information about the portion of fractured bedrocks, the thickness of quaternary deposits and the bedrock topography will demand future attention for a complete picture of the role of geology (potentially resulting in a statistically significant correlation with $F_{yw}$)._"**

[16] As a result, the perceptual model lacks information on groundwater storage processes.

**The proposed model is supposed to summarize what we can know at the moment. Especially, we want to link well-known hydrological processes with information related to the water age in order to harmonize empirical evidence of $F^*_{yw}$ values with the processes that lead to these values. In other words, many of the processes are known, but how these processes are linked with $F^*_{yw}$ is poorly addressed in the past literature: this last point is what we focus on in the manuscript. We will be pleased if future studies can add additional information to refine our perceptual model.**

*[17] Though the shortcomings of representing groundwater storage using $F_{qd}$ has been adequately explained in the discussion and conclusions, it is recommended that the authors explore other geological information as potential explanatory variables. For instance, bedrock type could have been added as an additional explanatory variable for a geological form of groundwater storage. Additionally, the depth of the deposits are not considered using this methodology, though that data may be difficult to obtain.*

**Yes, the above information cannot be easily obtained for all catchments. The bedrock type does not seem to add relevant information (Fig. 1 below) and part of this information is already included in $F_{qd}$.**

[Figure]

**Fig.1 $F^*_{yw}$ with varying bedrock type. Sed.rock: sedimentary rock, unc.sed.: unconsolidated sediments, dol: dolomite, met: metamorphic.**

**Minor issues: line-by-line**

**Thanks for these detailed suggestions, highly appreciated. We will implement them in the revised version.**

---

## Author Comment (AC3)

**Dear Jana von Freyberg,**

**thank you for your care and attention during your reading of the manuscript, your positive remarks and your comments that will help to improve the work. Please find below the responses to all your comments.**

**We will take into account all your constructive feedback in the revised version of the manuscript once we receive the editor's response.**

**With kind regards,**

**The Authors**

**General comments**

*Gentile et al. address the scientific questions of "... what drives $F_{yw}$ variations with elevation in Alpine catchments clarifying why $F_{yw}$ is low at high altitudes» (L20). For this, the authors combine existing and new $F_{yw}$ values from Switzerland and Italy and compare them with several other variables that describe snow cover, baseflow conditions, and geology. From these comparisons the authors develop a perceptual model, suggesting that a longer persistence of the seasonal snowpack results in deeper groundwater flow paths and thus smaller $F_{yw}$ values, in contrast to hybrid catchments with ephemeral snow packs. The authors also present a new classification scheme to identify a catchment's hydro-climatic regime. The analysis of the used data is thorough and most figures are clear and informative. The analysis of satellite images to explore the linkages between snow cover duration and $F_{yw}$ are certainly interesting.*

**Thanks for the positive overall assessment.**

*However, I would like to encourage the authors to highlight more the novelty of their findings and the scientific contribution of their work, considering that they cite several papers in which comparable analyses have been carried out and similar conclusions (with respect to flow and storage processes) have been reached. I think that the research objectives (or research questions) should be formulated more explicitly in the Introduction in order to guide the following analysis. It is not clear whether the authors attempt to explain the scatter in the $F_{yw}$-gradient relationship (L76), the low $F_{yw}$ values in steep and/or high-elevation catchments (L79), or both.*

**We will make the research objectives clearer in the introduction. The research question that motivated our work is: What are the hydrological processes hidden behind the low young water fractions in high elevation catchments? Our main hypothesis is that this relationship can at least partially be explained by snow cover persistence and quaternary deposits. These factors were not previously considered in the scientific literature for explaining variations in $F^*_{yw}$ ('*' indicates a flow-weighted variable) at different elevations:**

- **The snowpack persistence (quantified with $F_{SCA}$ in our work) is hereby seen as an essential factor that drives the duration of the low-flow period at high elevation sites, i.e., of the period where streamflow is only fed by groundwater (water that in terms of age is old water). In this context, we decided to also consider, as an explanatory variable, the groundwater contribution to streamflow. In this regard, we have applied a recent baseflow filter that emphasizes the physical relevance of the separated flow components (Duncan, 2019) (i.e., it describes the physics of baseflow better than other filters).**

- **Quaternary deposits are thought to be a factor that influences potential subsurface storage that contributes to the stream (Arnoux et al. 2021).**

**The potentially dominant water flow and storage processes driving young water fractions are of course discussed in previous work (e.g., Jasechko et al. 2016, von Freyberg et al. 2018, Lutz et al. 2018); to the best of our knowledge, this is the first attempt to present a perceptual model that harmonizes these known processes with the surprising low $F^*_{yw}$ values of high elevation catchments. This will be explicitly stated in the introduction.**

**In the analysis, we have also considered catchments at low elevations (rain-dominated) for having a complete vision about the dominant hydrological processes at different elevations, but the results presented in the pre-print do not convey new insights into the knowledge of $F^*_{yw}$ variations at low-elevation sites. We have in particular not explored the role of the low-flow period length (which cannot be related to the snow cover persistence); but could have a key role in driving $F^*_{yw}$ also at lower elevation sites. We will deepen this point in the revised version of the manuscript.**

**Below, we give some more detailed justification for our hypothesis about the link of low flow and $F^*_{yw}$. This will also be made clear in the paper:**

**$F^*_{yw}$ predicts the flow-weighted average over a certain time-interval (Kirchner 2016a, 2016b):**

$$F*_{yw} \simeq \frac{\sum_{i=1}^{n} Q(t_i) F_{yw}(t_i)}{\sum_{i=1}^{n} Q(t_i)} \qquad \text{Eq. (1)}$$

**where $n$ is the number of days in the considered time-interval, $Q$ is the daily discharge, $F_{yw}(t_i)$ is the daily young water fraction. As is clear from this equation, $F^*_{yw}$ becomes low if either $F_{yw}(t_i)$ is low for a high flow or if $F_{yw}(t_i)$ is very low for many time steps or both.**

**Low flow situations happen when the streamflow is baseflow-dominated (i.e. groundwater dominated, i.e. old water dominated), i.e. we can anticipate that low $F_{yw}(t_i)$ occur together with low $Q(t_i)$ and that $F_{yw}(t_i)$ is higher during high flow periods. The overall effect upon $F^*_{yw}$ remains thus a priori unclear. It is however tempting to think that i) the duration of low flow periods or ii) the share of baseflow could explain $F^*_{yw}$. Indeed, a plot of $F^*_{yw}$ against the duration of low flow periods (where a low flow period is defined as a period when 85% of the total flow is composed of baseflow) shows a strongly negative correlation (Fig. 1a below). Moreover, a plot of low-flow duration against elevation shows a decrease until 1500 m asl and an increase thereafter (Fig. 1b) (i.e., an opposite behavior with respect to $F^*_{yw}$ against elevation, see Fig. 12b of the pre-print). The NBPV does not fit the overall picture (Fig. 1b)**

[Figure]

[Figure]

*Figure 1. a) young water fraction against low-flow duration b) low-flow duration against mean catchment elevation*

Baseflow filters were applied in previous studies and results were correlated with $F^*_{yw}$. However, applying the Duncan (2019) baseflow filter (with the parameter suggested by Nathan and McMahon, 1990, without any type of calibration) to the discharge data of our catchments, the estimated $F_{bf}$ is roughly the complementary term of $F^*_{yw}$:

$$F^*_{yw} + F_{bf} \simeq 1 \qquad\qquad Eq.\ (2)$$

We better show this complementarity in Fig.2a and Fig.2b (below) and will include relevant aspects in the revised version. In these figures, we added the $F_{bf}$ uncertainty calculated through a Gaussian error propagation, considering the baseflow filter parameter as distributed according to a Gaussian distribution.

[Figure]

*Figure 2. a) young water fraction against fraction of baseflow b) complementarity between $F_{bf}$ and $F^*_{yw}$.*

This result indicates that $F^*_{yw}$ could potentially be estimated as: $F^*_{yw} \simeq 1 - F_{bf}$, without the application of the amplitude ratio approach using stable water isotopes. This could be a useful result for catchments in which stable water isotopes measurements are not available. Of course, future studies could compare the $F_{bf}$ with $F^*_{yw}$ for new catchments to validate or refute this result in different hydroclimatic conditions or geologies.

In summary: we will make the research questions and hypotheses clearer throughout the paper and emphasize on the novelties in the conclusion and the abstract.

*L156 "we classify the catchments in the three hydro-climatic regimes (snow-dominated, hybrid and rainfall-dominated) proposed by Staudinger et al. (2017), but we introduce a new formal criterion of classification": Why is a new definition of the catchments' hydro-climatic regimes needed? As far as I can tell, only two catchments, BIB and GUE, were newly classified. The new sites outside of Switzerland could have easily been categorized as hybrid or snow-dominated based on their streamflow and topographical data. Furthermore, the discussion of this new classification scheme (Sect. 4.2 and 5.1) somewhat distracts from the main topic of the paper, which is the investigation of small $F_{yw}$ in high-elevation catchments.*

**Thank you for this comment, which was already made by reviewer 1. We copy here the answer that we gave to reviewer 1:**

*"We propose a new criterion for the regime classification because our dataset includes catchments outside the Swiss borders (i.e., the four Italian catchments) for which the Weingartner and Aschwanden (1992) and Staudinger et al. (2017) classification scheme cannot be strictly applied since they were designed for the Swiss hydro-climatic regimes. We "manually calibrate" the thresholds of $F_{SCA}$ and $Q_{June}/Q_{DJF}$ for classifying catchments in "rainfall-dominated", "hybrid" and "snow-dominated" as in the work of Staudinger et al. (2017). In this way, the classification scheme is "calibrated" on the Staudinger et al. (2017) catchments and we can apply it also outside the Swiss borders. According to the referees' comments, we will consider the possibility of modifying the classification scheme to make it more straightforward to link to previous classification (e.g., using streamflow and topographical data), but it will remain transferable to other regions."*

*I was surprised to see that the authors did not include annual or seasonal precipitation in their analysis. This variable should be tightly related to $F_{bf}$ and $F_{SCA}$. Annual precipitation is also very low at some Swiss high-elevation sites, which would also explain why $F_{yw}$ is low there. What is the reason for not considering precipitation at all?*

**We decided to use variables that were not previously considered for explaining young water fractions variations; Jasechko et al. (2016) wrote: "Although topographic gradient provides the strongest correlation with young streamflow fractions in our data set, the fraction of unexplained variance is large, suggesting that other variables also play a significant role. _We observe no significant correlations between the young streamflow fraction and_ catchment size, _annual precipitation_, bedrock porosity, population density, or the fraction of catchment area comprised of pasture land or open water".**

**We explain in the manuscript that, below roughly 1500 m, the increase of $F^{*}_{yw}$ with elevation also depends on the increase of precipitation with elevation. In fact, in such cases, annual precipitation can be considered as a proxy of catchment wetness since we mainly observe a liquid water input.**

**Above 1500 m, using mean annual precipitation as a proxy for catchment wetness is misleading because the seasonal snowpack leads to a very dry period of the year despite high (solid) water input. In other words, the total amount of precipitation is not the variable of interest, rather the temporal concentration of water input is the relevant variable. It is possible to observe the saturation of the system (i.e., high wetness conditions) also when annual precipitation is low if a large volume of water (stored in the snowpack) is released in a relatively concentrated time interval. After the long winter period, we expect high infiltration that recharges the groundwater storage. This process can bring the system to saturation (high wetness) so that ultimately rain or snowmelt can more rapidly reach the stream as overland flow.**

**Additionally, Lutz et al. (2018), estimating the young water fraction for 24 catchments in Germany, have found exactly the opposite of what is generally thought: the young water fraction decreases with increasing mean annual precipitation. They stated that this result reflects "the impact of various factors relevant in the mountainous region, resulting in the decrease of young water fractions for higher-elevation catchments" (Lutz et al. 2018). Thus, what we did in our work is searching for the "various factors" that lead to low $F^{*}_{yw}$ in mountainous catchments not considering the mean annual precipitation as an explanatory variable.**

*The important aspect of snow pack storage in high-elevation, snow-dominated catchments, which the authors only touch on in the Conclusions section, should instead be brought up much earlier in the*

*manuscript. In fact, it has been discussed already in another paper: «Another analytical decision that affects the interpretation of $F^*_{yw}$ and $F_{yw}$ relates to whether snowpack storage is considered to be part of catchment storage, or not. If one measures precipitation to the snow surface as the catchment input, then snowpack accumulation and melt are implicitly included in catchment storage (e.g. Staudinger et al., 2017). In this case, comparisons of seasonal cycles in precipitation and streamflow should reflect the young water fraction resulting from the combination of snowpack and subsurface storage. Alternatively, if one uses precipitation and snowmelt arriving at the soil surface as the catchment input (for example, with melt pan lysimeters, or modelled snowpack out- flows), then snowpack accumulation and melt are implicitly excluded from catchment storage. In this case, comparisons of seasonal cycles in streamflow and sub-snowpack catchment input should reflect the young water fraction resulting from subsurface storage alone. Because the total catchment storage in the first case (including snowpack storage) is larger than the subsurface storage alone, the resulting young water fractions are expected to be smaller.» (von Freyberg et al., 2018). In addition, in high-elevation catchments with perennial snow packs, snowmelt in spring and summer is likely to be older than 2-3 months (because the snow fell more than 3 months before the melt occurs). As a result, although summer discharge might be high it will consist mainly of old snowmelt and groundwater rather than recent rainfall (i.e., $F_{yw}$ is small). In hybrid and rain-dominated catchments, streamflow receives relatively more young water from young snow packs and recent rainfall events, respectively.*

**Thank you for this comment. As reported in von Freyberg et al. (2018), the young water fractions are virtually identical between "direct" and "delayed" input, but, of course, there is a "conceptual" difference in using the "direct" or "delayed" input, which we omitted to discuss in the pre-print. We will take this in account for the revised version of the manuscript.**

**Considering our hypothesis (supported by scientific literature) that snowmelt mainly transits through the groundwater store, we will estimate, in the revised version, the $F^*_{yw}$ with the "direct" input (as suggested in the comment to L553): i.e., we will consider the snowpack as part of the catchment storage. Moreover, we will bring up the role of snowpack storage earlier in the text and we will cite that the role of snowpack storage in high-elevation, snow-dominated catchments has been discussed already in the work of von Freyberg et al. (2018), in which the authors state that, including the snowpack storage in the catchment storage, the resulting young water fractions are expected to be smaller. However, we will clarify in the revised version that the main aim of this work is not to focus on how the snowpack affects the $F^*_{yw}$ estimation in a single catchment, since it was treated by previous works (e.g., von Freyberg et al. 2018, Ceperley et al. 2020), but to describe what are the hydrological processes (also related to the snowpack storage) hidden behind the $F^*_{yw}$ variations with elevation gradient, focusing on low $F^*_{yw}$ at high elevations.**

**We agree that, for snow-dominated systems, snowmelt in spring and summer is likely to be older than 2-3 months (because the snow fell more than 3 months before the melt occurs). We also agree that summer discharge will consist mainly of old snowmelt (or groundwater)** rather than recent rainfall (i.e., $F^*_{yw}$ is small).**

**\*\*However, we believe it is not correct to distinguish between old snowmelt and groundwater. We know that recent snowmelt is likely to be older than 2-3 months and we also know that, according to several papers, recent snowmelt has a key role in recharging the groundwater storage during summer. Therefore, groundwater storage is assumed to be mainly composed of old snowmelt. If the groundwater storage contributes to the stream (mainly during the long winter low-flow period, but also during summer), this contribution will reduce the $F^*_{yw}$.**

*The authors seem to overlook this storage aspect of the snowpack and instead focus mainly on the groundwater contribution to streamflow (L82).*

**Thanks for this important comment. In addition to our above answers, it is of prime importance to point out here and in the paper that large parts of the snowmelt actually transit through the groundwater storage: i) the very high baseflow in high mountain catchments during summer is a direct sign of this fact. ii) groundwater in such catchments often has the isotopic signature of snowmelt (Michelon et al., HESSD paper, others). We will make this much clearer in the paper. Please see also our answer to the community comment by Jansson, the general comment number (1) on the relation between $F^*_{yw}$ and $F_{bf}$.**

*A main finding of the paper is a strong negative correlation between the baseflow fraction $F_{bf}$ and $F_{yw}$ (Sect. 4.3.3, Fig. 10) from which the authors derive several statements which I'd like to comment on (Sect. 5.4):*

*L553: "We find the highest $F_{bf}$ for snow-dominated catchments confirming the presence of high subsurface storage, contributing to streams, in high-elevation catchments». I would include the snowpack as part of the storage here because winter precipitation is stored in the snowpack until summer when it recharges aquifers or runs off into the stream.*

**Thank you for this comment. You are right that we have not pointed out the fact that, in our analysis, we are assuming that the snowpack is part of the catchment storage. We will make this coherent with the estimation of $F^*_{yw}$ through the "direct" input. We will include it and mention the relation between snowmelt and groundwater.**

*L554.: "Moreover, the annual baseflow is strongly positively correlated with the $F_{SCA}$ ($\rho_{Spearman}= 0.81$ p-value < 0.01) suggesting a major groundwater contribution with increasing snow cover persistency (Fig. S6)». This depends strongly on your baseflow estimation method.*

**Calculating the annual baseflow through another baseflow filter (e.g., the Lyne and Hollick, 1979) the positive correlation with $F_{SCA}$ does not change (Fig. 3a and Fig. 3b).**

[Figure]

***Figure 3. Annual baseflow against $F_{SCA}$ considering two different methods of baseflow estimation: a) Duncan (2019), b) Lyne and Hollick (1979).***

*Further, increasing baseflow and snow cover persistency are both results of increasing catchment elevation and/or annual precipitation. Thus, baseflow cannot simply be linked to snow cover persistency.*

**We are not sure if there is literature that shows that baseflow increases as a function of catchment elevation but that's indeed what we find in our work (see Figure 2b). In general, authors assume that low elevation catchments have more groundwater storage potential because of large alluvial aquifers.**

**Baseflow could increase with precipitation (given that there is enough storage potential in the subsurface), but in Switzerland, a general increase of precipitation with elevation is observable only up to 1600 m asl., at higher elevation the trend is unclear (Sevruk, 1997).**

**Finally, we would like to underline that we do not pretend that snow cover persistence alone explains baseflow, we simply show the statistical link. This will be made clear.**

L558: "The hydro-climatic regime is generally a good indicator of the proportion of young water that contributes to streamflow..." What does this mean exactly? If the authors refer to Tab. 3, there is quite some overlap between the rainfall- and snow-dominated regimes with respect to $F_{yw}$, and thus $F_{yw}$ cannot be estimated from the regime types alone.

**We are saying that through the regime type one could roughly say the "order of magnitude" of $F^*_{yw}$. However, since this sentence can create misunderstanding, we will reformulate or remove it.**

*L570 (&L37): "Therefore, we can conclude that the contribution of groundwater storage to streamflow, which is driven by snowpack duration, can be considered as the best explanatory variable of the $F_{yw}$ elevation gradients." Again, I would rather argue that not snowpack duration but rather storage capacity (both in the subsurface and the snowpack) together with the hydro-climatic conditions (P-ET) and catchment properties affect the contribution of old water (not necessarily only groundwater) to streamflow, and thus $F_{yw}$. In high-elevation catchments, the snowpack can function like a subsurface water storage that releases (>3 months) old water during the melting season. This old water is meltwater, not groundwater and I suspect that the baseflow separation method used in this paper is not able to differentiate between the two.*

**We will make the link between snow cover, groundwater and baseflow much clearer in the revised version. We agree that snow releases old water, but groundwater is largely composed of snowmelt in these systems. The sustained high flows in July in high elevation catchments (without glacier) are not the result of continuous overland flow composed by meltwater, but the result of groundwater release, i.e. of groundwater that was previously recharged by melt. We thus cannot distinguish between snowmelt and groundwater. But with the help of the baseflow ratio, $F_{bf}$, we can quantify the share of streamflow that is due to groundwater release; the share of snowmelt (with age > 3 months) that flows off quickly will not show up in $F_{bf}$ and, in many sites, this could explain the "residuals" of 1- ($F_{bf}$ + $F^*_{yw}$). We quote here for completeness also our answer to general comment [1] of Jansson:**

*"We agree that in snow-free systems, $F^*_{yw}$ is by definition related to $F_{bf}$, the ratio of baseflow in annual flow: baseflow is composed of groundwater and groundwater is the dominant source of old water in snow-free systems (in absence of large lakes). However, in snow-influenced systems, part of the old water is temporarily stored in the snowpack. This is at a first glance not measured by $F_{bf}$. We nevertheless show here that $F_{bf}$ obtained from streamflow alone (with the selected baseflow filter) leads to an approximately complementary relationship to $F^*_{yw}$ ($F^*_{yw}$ + $F_{bf} \simeq 1$), which is an important result for catchments where we do not have isotope measurements. Why is this so? Simply because the snowmelt in snow-dominated systems largely transits through*

*the groundwater store and leads to high baseflow. This will be made much clearer in the revised version."*

*Based on the analysis of slope data the authors conclude that (L370) "… that there is an increasing rate of infiltration when the hydro-climatic regime transitions from hybrid to snow-dominated.". I don't think that this statement is well supported by using slope data in Fig. 4 (no data on infiltration is provided). Instead, the only conclusion that can be drawn from the data presented in this manuscript is that the hybrid catchments receive more precipitation than the rain-dominated catchments (L478), resulting in more recent precipitation becoming streamflow, i.e. higher $F_{yw}$ values. This is analogous to earlier findings in von Freyberg et al. (2018): "… young water fractions tend to be highest in humid catchments where prompt runoff response is facilitated by fast flow paths and/or high-intensity precipitation events."*

**Thank you for this comment. Yes, we do not have infiltration data and probably this conclusion only using slope data cannot be well supported, also if Jasechko et al. (2016) concluded that in steeper terrain the low $F^*_{yw}$ could be caused by rapid percolation through fractures and deep flow paths (as also reported in Lutz et al. 2018). We will change this part in the manuscript incorporating the conclusion you suggest and citing the relative paper.**

*One outcome is a "perceptual model of how snow persistency explains $F_{yw}$ during winter and summer along topographic gradients". This model, presented in Fig. 13, tries to summarize the combined effects of catchment properties (steepness, elevation) with processes (ET, wetness, snowmelt). The resulting figure is very complex and difficult to understand. For instance, if a reader seeks to understand the figure without reading the entire paper, is not clear as to what "increases/decreases with elevation" means. Does this refer to increases/decreases of $F_{yw}$ within a single catchment or between different (high- to low elevation) catchments?*

**We realize that Fig. 13 can be misleading since only a single catchment is represented. Therefore, we will work to improve this figure to better reveal our "step forward" regarding the hydrological processes behind the $F^*_{yw}$ variations between different catchments.**

**Specific comments**

*The title of the manuscript does not well reflect the content of the paper. It rather gives the impression that $F_{yw}$ was studied along elevation gradients within (individual) catchments. In addition, the term "Alpine" suggests that solely mountainous catchments within the Alps mountain range were considered, however, catchments such as ERG, AAB and MEN are located in the Jura Mountains and Swiss Plateau, respectively. It would be nice to define early on what is meant here by Alpine, given that the Introduction starts with the general statement (L41) "Alpine catchments are assumed to generate a high share of surface runoff …"*

**Thank you for this comment. We will work on the title to avoid reference to Alpine when we talk about all studied catchments.**

*Ideally, the time periods that were used to calculate the various metrics should be the same as those of the isotope data used to calculate $F_{yw}$. As far as I can tell, this has been considered only for $F_{bf}$, whereas $F_{SCA}$ was determined based on satellite data from 2017-2021. For WFI and $Q_{June}/Q_{DJF}$, no information is provided. The $F_{yw}$ values in von Freyberg et al. (2018) only cover the time periods 2010-2015, which is not even overlapping with the satellite images used to determine $F_{SCA}$. I would*

*like to encourage the authors to compare data only from the same time periods, especially when these periods included extremely dry/wet climatic conditions.*

**We agree. In fact, $F_{bf}$ and *WFI* were calculated in the same time period of isotope sampling, this will be specified. $F_{SCA}$ is calculated in the period 2017-2021 simply because of the availability of the Sentinel-2 satellite images (there are no Sentinel-2 images in the period 2010-2015). We will make this clear.**
**For the $Q_{June}/Q_{DJF}$, we used a long-term average since this ratio was used for a classification purpose. We will make the retained time periods explicit in the revised version.**

*The terms elevation and steepness should not be used synonymously, as in L361:" Initial evidence of low $F_{yw}$ in high-elevation catchments is given in the work of Jasechko et al. (2016). Based on the analysis of 254 worldwide watersheds, their work reveals a reduction of $F_{yw}$ in steeper terrains." Also, low elevation (rainfall dominated) catchments can be very steep, and there surely exist high-elevation (snow dominated) catchments with flat topography.*

**We agree. We will carefully review the language.**

*When I look closer at the $f_{SCA}$ time series (Fig. 5), I wonder how it is possible that the AAC catchment at around 500m asl. was almost entirely snow covered in summers of 2018 and 2020 ($f_{SCA}$ around 1)? The same is true for the catchments BIB and ERL where the snow cover usually disappears by June each year. Can it be that $f_{SCA}$ tends to be over-estimated with your approach?*

**Our estimation algorithm indeed suffers from overestimation (see our answers to reviewer 1) and will be modified in the revised version.**

*I would also expect $F_{SCA}$ to be strongly correlated with (mean) catchment elevation so that elevation instead of $F_{SCA}$ could be used in your analysis. As can be seen in Fig. 12, a similar grouping of catchments emerges.*

**A priori, we could imagine a strong correlation between these two variables, but elevation is not an explicative "tool" for the processes we focus on in the manuscript. In fact, it could be approximative to describe the snow cover persistence only with the increasing elevation: the persistence of snow in a catchment also depends on catchment's aspect, topography, (Painter et al. 2023), snow-related and climatic characteristics. In fact, catchments with very different characteristics (e.g., different elevation ranges, different areas etc.) can reveal a similar mean elevation, but the snow persistence could considerably change. Thus, we need a "tool" that is directly linked to the processes we want to describe and the $F_{SCA}$ is a variable that, of course, is related to the mean catchment elevation, but it is "better physically related" to the snow-cover persistence.**

*L275 mentions that "The Noce Bianco Pian Venezia (NBPV) catchment is an exception since it generally has snow over the glacier also during summer.». As far as I remember, the catchments VdN, DIS and OVA are also partially glacierized. Should they be considered as exceptions as well?*

**We consider only NBPV as an exception because 42% of its area is covered by glaciers. DIS only 2 %, VdN 3 %. For OVA we see it is not covered by glaciers (van Tiel et al., 2019). Thus, for the other catchments, we consider negligible the effect of glaciers on $F^*_{yw}$. We will make clearer the high glacier-cover ratio of NBPV.**

*Fig.10: A very similar result is presented already in von Freyberg et al. (2018) where $F_{yw}$ and the quickflow index QFI, the inverse of the baseflow index, showed a significant positive correlation (note that the QFI and 1/Fbf will likely not be exactly the same, although both were calculated through digital filtering of discharge time series).*

**Thank you for this, we will cite, in the discussion about Fig. 10, that a similar result was found by von Freyberg et al. (2018). However, we want to underline that Duncan (2019) improved the baseflow filter of Lyne, V. D. and M. Hollick (1979) [BaseflowSeparation, EcoHydrology package in R], used by von Freyberg et al. (2018), to separate flow components with physical relevance (Duncan, 2019). von Freyberg et al. (2018) found a positive correlation between $F*_{yw}$ and QFI: average of $(Q-Q_{bf})/Q$ (where $Q_{bf}$ is obtained with Lyne, V. D. and M. Hollick, 1979 baseflow filter). However, $F_{bf}$, average of $Q_{bf}/Q$ = average of 1-QFI, is not complementary to $F*_{yw}$ (i.e., $F*_{yw}$ + average of (1-QFI) ≠ 1). A good result of our work is that, using the baseflow filter parameter commonly proposed in literature (Nathan and McMahon, 1990), the Duncan (2019) baseflow filter returns a $F_{bf}$ that is roughly complementary to $F*_{yw}$ (i.e., $F*_{yw} + F_{bf} \simeq 1$) without any type of calibration. This could be a very useful result for catchments in which isotopes measurements are not available. In these cases, $F*_{yw}$ could be estimated as $1-F_{bf}$.**

*L484: "In addition, higher order channels, higher up, are more rarely activated than lower order channels that are more often active" If the authors refer to Strahler stream orders here, higher elevation streams usually have low Strahler orders (starting with first-order streams). The Strahler stream orders increase downstream.*

**Thanks for pointing out our language mistakes.**

*L560-565: Why was BCC not classified as snow-dominated, based on the evidence from previous research?*

**This was because of the new classification scheme we propose in the paper. We will come up with a final conclusion about the classification scheme in the revised version.**

*L566-569: Is it possible that precipitation isotopes in the NBPV catchment were sampled differently compared to the other catchments in this study, e.g. with a heated precipitation collector? This could result in a larger $A_S$ value. Can the authors confirm that the precipitation isotope sampling in the snow-dominated catchments was comparable across all sites?*

**Thank you for suggesting the clarification of the approach used for sampling precipitation. In NBPV catchment, we did not use a heated precipitation collector. Bulk samples of rain water were collected monthly at the outlet of the catchment by 5-L bottles equipped with a funnel and a layer of mineral oil to prevent evaporation, whereas snow samples were collected using an aluminum cylinder, inserted vertically from the surface to a depth of 20 cm (Zuecco et al., 2019). We applied the same sampling approach of precipitation in NBPV and BCC (Penna et al., 2016). For VdN "bulk rain samples were collected for isotopic analyses using funnels flowing into insulated bags at three locations corresponding to the rain gauges (1,253, 1,500 and 2,100 m a.s.l.), and emptied weekly or biweekly between June 2016 and November 2018. Between February 2016 and April 2018, snow samples were collected from the entire snow profile at various locations in the catchment" (Ceperley et al. 2020). For DOR and SOU precipitation samples were collected at a monthly resolution using a double rain and snowfall isotope sampler installed on a pole 3.7 m high. Therefore, we consider the precipitation isotope sampling comparable across all the new sites, while precipitation isotopes in the 22 sites of von Freyberg et al. (2018) are modeled through an interpolation method.**

*L596: "...leads to high baseflow throughout the year...». This contradicts the data shown in Fig. 9. I would suggest to replace 'baseflow' with 'baseflow fractions $F_{bf}$'.*

**Thank you for this suggestion.**

**Technical comments**

*The language of the manuscript is often not precise and needs to be improved. Some sentences are difficult to understand, e.g.*

**Thank you for all technical comments and for having taken the time to report the language issues. We will improve and clarify the language over the entire manuscript.**

*- (L310) "Additionally, Duncan (2019) provides a specific technique that allows estimation of separate components with physical relevance in the case that baseflow separation techniques were not applied to describe physical processes." This sentence is redundant and not scientifically specific (e.g., what are "separate components with physical relevance"?).*

**The separate components with physical relevance are baseflow and quickflow (i.e., total flow minus baseflow).**

*- (L33) "Finally, our work highlights that $F_{bf}$, considered as a proxy for groundwater flow, is roughly the one's complement of $F_{yw}$". Isn't $F_{bf}$ rather a proxy for the groundwater contribution to streamflow? It does not provide any information about flow processes. What does "roughly the one's complement of $F_{yw}$" mean?*

**Thank you for this comment. We consider $F_{bf}$ as a proxy for the groundwater contribution to streamflow knowing that the used baseflow separation method is able to describe how much is relevant the baseflow (or groundwater flow) on the entire hydrograph.**

**The meaning of "roughly the one's complement of $F_{yw}$" is that, for each catchment, $F_{bf} + F^*_{yw} \simeq 1$.**

*- L34 «...we find high $F_{bf}$ during all low-flow periods, which underlines that streamflow is mainly sustained by groundwater in such flow conditions.» That high $F_{bf}$ represents a major contribution of groundwater to streamflow is implicit in the method of Duncan (2019). This is not a new finding.*

**Thank you for this comment. We simply wanted to underline what was the meaning of high $F_{bf}$ during low-flow periods for a general reader. However, we can omit the second part of this sentence.**

*- L496 "the temporal dynamic of snow accumulation and melt and its effect on deep infiltration supports the pivotal role of snowmelt in recharging groundwater during summer in high-elevation environments ..." This sentence is redundant. Snow melt affects deep infiltration is equivalent to it plays a role in recharge.*

**Thank you for this. We will modify this sentence as: "the temporal dynamic of snow accumulation and melt supports the pivotal role of snowmelt in recharging groundwater during summer in high-elevation environments ..."**

*Sect. 3.2: To indicate whether a variable was flow weighted, earlier papers have added a "\*", and thus I would suggest to write F\*$_{yw}$ and A$_S$\* here as well.*

**Thank you for this suggestion. We will use this notation in the revised manuscript.**

*L328: Please verify whether the flow-weighted young water fraction of SOU is indeed 0.01. If so, the following statement "while flow-weighted F$_{yw}$ remains unchanged for the very small lateral subcatchment" is false.*

**Thank you for noticing this. It is simply an error in the text: the flow-weighted young water fraction of SOU is 0.1, not 0.01 and the statement "while flow-weighted F$_{yw}$ remains unchanged for the very small lateral subcatchment" is true.**

---

## Author Response (AR1)

**Author's response: a list of all relevant changes made in the manuscript and a point-by-point response to the reviews.**

Alessio Gentile[1], Davide Canone[1], Natalie Ceperley[4], Davide Gisolo[1], Maurizio Previati[1], Giulia Zuecco[3], Bettina Schaefli[2,4], and Stefano Ferraris[1]

[1]Interuniversity Department of Regional and Urban Studies and Planning (DIST), Politecnico and Università degli Studi di Torino, Torino, Italy
[2]Institute of Earth Surface Dynamic (IDYST), Faculty of Geosciences and Environment (FGSE), University of Lausanne, Lausanne, Switzerland
[3]Department of Land, Environment, Agriculture and Forestry (TESAF), University of Padova, Legnaro, Italy
[4]Institute of Geography (GIUB) and Oeschger Centre for Climate Change Research (OCCR), University of Bern, Bern, Switzerland

*Correspondence to*: Bettina Schaefli (bettina.schaefli@giub.unibe.ch)

Dear Editor and Referees,

we would like to thank you for both the overall appreciation of our work and the appreciation of our plan to revise it. Considering the referees' comments, the Editor decided that major revisions are necessary before the review process can be continued. The referees' comments have been very constructive for the paper improvement and have been the main drivers of our changes. Accordingly, you will find major changes in the revised manuscript. We have addressed all the issues raised in the interactive discussion, including the language improvement. To meet all the referees' requests, many sections have been deeply reorganized and rewritten, as it is possible to see in the track-changes version of the manuscript.

The present document is subdivided in two Sections. In the first section we summarize all the major changes applied to the submitted document you have revised. In the second section we report a point-by-point response to the reviews.

In the hope of having met your scientific expectations in the revised manuscript, we kindly ask you to reconsider the publication of our work on the Hydrology and Earth System Sciences Journal.

With king regards,

The Authors

**1     List of all relevant changes made in the manuscript.**

35   **1.1     Title: relevant changes**

Jana von Freyberg commented: "The title of the manuscript does not well reflect the content of the paper. It rather gives the impression that $F_{yw}$ was studied along elevation gradients within (individual) catchments. In addition, the term "Alpine" suggests that solely mountainous catchments within the Alps Mountain range were considered…".

40   Starting from this comment we have decided to change the paper title:

From:     "*What drives $F_{yw}$ variations with elevation in Alpine catchments?*"

To:       "*Towards a conceptualization of the hydrological processes behind changes of young water fraction with elevation: a focus on mountainous alpine catchments.*".

45

We think that this title reflects much better the content of the paper, which is concluded with the presentation of a perceptual model, the first step before conceptualization (Beven, 2012), that integrates all the results of our analysis to describe a framework for how hydrological processes control the $F^*_{yw}$ according to elevation. In the title, we underline that the focus of the paper is on mountainous alpine catchments. We use the term "**a**lpine" instead of "**A**lpine" to refer to the typical hydro-

50   climatic conditions of a mountain climate. In other words, we are not specifically referring to the Alps Mountain range. Finally, we have changed "$F_{yw}$" to "young water fraction" in the title as suggested by the referee #1: "… "$F_{yw}$" could be changed to "young water fraction" for general readers".

**1.2     Abstract: relevant changes**

We highly reformulate the Abstract to better communicate the scientific contribution of our research paper and the novelty of

55   our findings, as suggested by Jana von Freyberg: "I would like to encourage the authors to highlight more the novelty of their findings and the scientific contribution of their work".

**1.2.1     Scientific contribution**

-     We express the research gap about the missing of a harmonious framework of hydrological processes explaining low young water fraction in mountainous catchments and, accordingly, that our aim is to give an overview of what

60           drives the young water fraction variations according to elevation, thus clarifying why it generally decreases at high elevation.

**1.2.2 Novelty**

- We communicate in the Abstract that we provide the above-mentioned missing framework through our perceptual model.
- Our novel findings reveal that the low-flow duration (LFD), quantified in the revised version of the manuscript, is the main driver of $F^*_{yw}$ variations with elevation and that it is very high in high-elevation catchments because of the persistence of the winter seasonal snowpack which promotes the groundwater (or old water) storage emptying which sustains the streams during winter.
- We highly stress in the Abstract the usefulness of the complementarity between $F^*_{yw}$ and $F_{bf}$ (we have found in this work) to potentially estimate $F^*_{yw}$ in catchments where stable water isotopes measurements are not available.

**1.3 Introduction: relevant changes**

We have applied major changes to the Introduction. The main info presented in the submitted version remain in the revised Introduction, but the integration of additional information required by the referees forced us to both a reorganization and a rewriting of this Section. The relevant changes are summarized here below:

- **We rewrite some parts of the Introduction to make the research objectives clearer,** as requested by Jana von Freyberg: "I think that the research objectives (or research questions) should be formulated more explicitly in the Introduction in order to guide the following analysis".
- **We explain the difference between "direct input" and "delayed input", previously addressed by von Freyberg et al. (2018), for estimating the $F^*_{yw}$.** In other words, we better underline the role of snowpack for estimating $F^*_{yw}$ as requested by Jana von Freyberg: "The important aspect of snowpack storage in high-elevation, snow-dominated catchments, which the authors only touch on in the Conclusions section, should instead be brought up much earlier in the manuscript…".
- **We introduce the importance of the low-flow duration (LFD) and why it should affect (reduce) the $F^*_{yw}$.** We have explained this aspect inserting the formulation of the flow-weighted average young water fraction (Eq. 1 of the revised manuscript), that is what we accurately predict using the amplitude ratio approach (Kirchner, 2016b).
- **We move a paragraph, explaining the importance of considering the role of Quaternary deposits for the reduction of $F^*_{yw}$ at high elevations, from the Methods Section to the Introduction.** This was requested by referee #1: "L282-289: "Some authors have revealed … $F_{yw}$ in Alpine catchments". This is more suitable for Introduction than Methodology".
- **We add a paragraph in which we clearly explain why we have not considered the mean annual precipitation as a possible explanatory variable for snow-dominated catchments.** This was a comment posted by Jana von Freyberg: "…What is the reason for not considering precipitation at all?"

**1.4 Study sites: relevant changes**

**The Study sites Section was completely reorganized and rewritten** following the comment of referee #1: "…The data description section (entire section 2) needs to be restructured and revised to make it more concise and clearer.". The relevant changes are summarized here below:

- **We describe the assembled data set and catchments attributes in a single Section (merging the "2.1 Existing data set", "2.2 Additional data set" and "2.3 Complete data set" sections),** as suggested by referee #1: "Both sections about the data (e.g., Section 2: existing data, additional dataset, complete data, and Section 3: discharge data and catchment boundary), why do the authors need two different sections?". **A lot of information reported in the "old" 2.1, 2.2 and 2.3 sections have been inserted in the revised Supplementary Material.**

- **We insert the relevant information (elevation, slope, geology, average precipitation, discharge) about the study sites in a Table (Table 1 of the revised manuscript)** as suggested by referee #1: "…The data description section (entire section 2) needs to be restructured and revised to make it more concise and clearer. I think this can be done using a table.", "The authors can here focus more on catchment attributes, such as climate (e.g., average annual precipitation and discharge), land use cover, geology, and discharge." **In Table 1 we added a new column with the indication of the Period of isotope sampling (PoS) and we have updated the calculations so that precipitation and discharge are referred to the same time-window of isotope sampling**, as commented by Jana von Freyberg: "… Ideally, the time periods that were used to calculate the various metrics should be the same as those of the isotope data used to calculate $F_{yw}$" and as requested by referee #1: "Line 190: I suggest mentioning the study period for the isotope data and $F_{yw}$ for the different study catchments since it is different."

- **We insert Fig. 2 a,b and Fig. 2 c,d representing "the mean catchments slope against mean catchments elevation" and "the average precipitation and discharge against elevation", respectively,** as requested by referee #1: "…I am curious to see the relation between average elevations and average slopes for the 27 catchments, is there a positive correlation? (Also, for average elevation with annual precipitation)". **Fig. 2a of the revised version corresponds to Fig. 4a of the submitted version. We move this figure to the Study sites Section** following the referee #1 suggestion: "Figure 4a can be moved to the data section…". **The figure is described and commented in the text.**

- **We insert Fig. 3 representing the boxplots of the mean monthly flow for the 27 study catchments according to their hydro-climatic regime.** In the submitted version all the boxplots were overlapped in a single figure (Fig.2 of the submitted version), while in the revised version we have separated the boxplots, according to the hydro-climatic regimes, as suggested by the referee #1: "…it is not easy to differentiate between the three boxplots, I would suggest having three separated boxplot figures with the same y axis limit…". **In addition, we better explain this figure in the text** following the referee #1 comment: "In summer there is a higher average monthly flow from snow-dominated catchments than from rain-dominated ones (due to increased snowmelt, I suppose), and in winter it

is the other way around. Please explain this better in the text because it is not clear… This figure should be described in the text (there is no description of this figure, it was only cited in line 243)". Describing the figure, **we add in the text information about discharge dynamics of our study catchments** as requested by the referee #1: "Here, I would expect more description of the discharge dynamics (e.g., giving an order of magnitude to these data by telling what the annual discharge is, whether the runoff is seasonal, etc)".

- **We modify Fig. 1 to improve the map visibility** according to referee #1 suggestion: "Figure 1: the background cannot be easily seen; I think you could replace with a DEM map. In addition, I cannot differentiate between Quaternary deposits and hybrid catchments visually."

**1.5 Material and Methods: relevant changes**

We have applied major changes to the Material and Methods Section. The main info presented in the submitted version remain in the revised Material and Methods Section but following the referees' comments/suggestions, we applied a reorganization of this Section. The relevant changes are summarized here below:

- **We remove the "3.1 Discharge data and catchments boundaries" sub-section,** as suggested by referee #1: "Much of the information provided in Study Sites, and Material and Methods is not relevant (e.g., shape file, detailed source of data, etc.). Instead, citing the sources of the various data (both from individuals and organizations) can be moved to either the Authors' Contributions or Acknowledgements, or in the supporting information Sections". **Accordingly, we move the content of this sub-section to the *Data Availability* section (at the end of the revised manuscript).**

- **We partially rewrite and integrate with additional info the old Section "3.2 Young water fraction estimation from seasonal cycles of stable water isotopes in precipitation and streamwater".** In the revised manuscript, the content of this Section is reported in Section "3.1 Young water fraction estimation from seasonal cycles of stable water isotopes in precipitation and streamwater: the "direct" input.". We give here more details regarding the "direct input" approach (mentioned earlier in the Introduction) for the estimation of $F^*_{yw}$. In this regard, **we insert the Fig.4 of the revised manuscript (missing in the submitted version) for a visual representation of the "direct input" approach.**

- **We show here in Fig. 5 (i.e., Fig. 12 b of the submitted version) the relation between $F^*_{yw}$ and mean catchments elevation.** Please note that the $F^*_{yw}$ values are different from those used in the submitted version. **In the revised version, all the $F^*_{yw}$ values are obtained with the "direct input" approach.**

- **We remove the Section "3.3 A new hydro-climatic regime classification: the classifiers".** We decide to not introduce a new classification system as suggested by both the referees: "Why did the authors need to propose a new criterion for catchment classification? The authors used two variables: (1) streamflow ratio between different months and (2) snow cover fraction for the proposed catchment classification, but later they adjusted the threshold of these two variables to have consistent results with Staudinger et al. (2017). Why didn't they just use the method

of Staudinger et al. (2017)?", "Why is a new definition of the catchments' hydro-climatic regimes needed? As far as I can tell, only two catchments, BIB and GUE, were newly classified. The new sites outside of Switzerland could have easily been categorized as hybrid or snow-dominated based on their streamflow and topographical data." **Following the referees' comments, we decide to remove the new classification scheme**, also because Jana von Freyberg added: "Furthermore, the discussion of this new classification scheme (Sect. 4.2 and 5.1) somewhat distracts from the main topic of the paper, which is the investigation of small $F_{yw}$ in high-elevation catchments.". Accordingly, we classify Swiss catchments using the Staudinger et al. (2017) classification scheme, while we classify Italian catchments using streamflow and topographical data, as suggested by Jana von Freyberg. Specifically, we have found that Stoelzle et al. (2020) used a classification scheme (based on catchments elevation, typical low-flow period, typical snow onset and typical begin of snowmelt) to classify catchments outside Switzerland (e.g., German catchments).

- **We partially rewrite the Section "3.4 Average fractional Snow Cover Area ($F_{SCA}$) computation". This is because we revised the methodology for estimating the $f_{SCA}$ since the old methodology was prone to an overestimation of $f_{SCA}$ as noticed by the referees: "Eq: (5) the denominator ($N_{tot} - N_{clouds}$): This could result in an overestimation of $f_{SCA}$.", "Can it be that $f_{SCA}$ tends to be over-estimated with your approach?". If $f_{SCA} > 1$ we calculate $f_{SCA}$ as $N_{snow}/N_{tot}$ since this is the only heuristic solution that guarantees no overestimation.** The effect of the new methodology can be seen comparing Fig.13a of the revised manuscript with Fig. 12a of the submitted version. **The revised Section title is "3.2 Snow cover persistence quantified through the average fractional snow cover area ($F_{SCA}$)"**

- **We partially rewrite the Section "3.5 Accounting for groundwater: fraction of quaternary deposits ($F_{qd}$), Winter Flow Index (*WFI*) and baseflow fraction ($F_{bf}$)". In the revised version the section title is "3.3 Fraction of Quaternary deposits, low-flow duration and the groundwater contribution to the stream".** We added more details about the baseflow separation of Duncan (2019) and **we add here a paragraph in which we define the low flow period ($T_{Low}$) and the low flow duration (*LFD*). We also add a paragraph in which we explain how we estimate the uncertainty of $F_{bf}$ (it was not computed in the submitted version).**

- **We insert Fig. 6 to show the coverage of Quaternary deposits for some representative study catchments.**

**1.6 Results and Discussion: relevant changes**

**Please note that we change paper structure from "Results/Discussion" to "Results and Discussion".** The main difference is that in the submitted version we first present the "objective" results and then we comment on such results in the Discussion section. In the revised version results and comments are coupled and merged in a unique Section. Some major and minor comments of referee #1 have led us to the use of the "Results and Discussion" structure: e.g., "The manuscript needs to be restructured and revised. There is a lack of clarification in the text.", "L389-391: Our results … for hybrid catchments (median $F_{yw}$ of 0.32) … Why are there these differences? I suggest arguing and explaining them.". In this second

comment referee #1 asked us to argue about the results, probably because she/he expected to read a Discussion coupled with the result. In fact, we have accounted for an explanation of the differences required by the referee #1 in the Discussion section. For these reasons, we realise that a "Results and Discussion" structure could improve the readability and make the text more fluent.

Having said that, the relevant changes are summarized here below:

- **We move the sub-section "4.1 Young water fraction estimation for DOR and SOU catchments" to the Supplementary Material**, as suggested by referee #1: "Section 4.1. I think this can be moved to the data section or supporting information, as this is only for 2 catchments."

- **We remove the subsection "4.2 The new hydro-climatic regime classification".** This is because we have decided to remove the new classification scheme from the paper and to use the method of Stoelzle et al. (2020) to classify catchments outside Switzerland.

- **We remove the sub-section "4.3 New explanatory variables for the $F_{yw}$ elevation gradients". Some information of this sub-section have been moved to the Study sites section,** as suggested by referee #1: "Figure 4a can be moved to the data section…" (we move the figure and its description). We remove Fig. 4b and its description since the statements expressed in the submitted version were not well supported according to Jana von Freyberg: "Based on the analysis of slope data the authors conclude that (L370) "… that there is an increasing rate of infiltration when the hydro-climatic regime transitions from hybrid to snow-dominated.". I don't think that this statement is well supported by using slope data in Fig. 4 (no data on infiltration is provided)".

- **We modify the order of the results presentation from "4.3.1 Fractional Snow Cover Area ($f_{SCA}$) and $F_{yw}$",** "4.3.2 The role of quaternary deposits", "4.3.3 Groundwater contribution to streamflow: *WFI* and $F_{bf}$ related with $F_{yw}$" to "4.1. The role of Quaternary deposits","4.2 Stored (old) water contribution to streamflow ($F_{bf}$) and $F^*_{yw}$", "4.3 Low-flow duration (LFD) and $F^*_{yw}$", "4.4 The role of snowpack persistence"

- **We remove Fig. 8a and move Fig. 8b to the Supplementary Material.** This is because the *WFI* somewhat deviates from the fil rouge of the paper.

- **We update the Fig. 7 (in the revised version Fig. 7a) with the $F^*_{yw}$ obtained through the "direct-input" approach and we underline the negative correlation between $F^*_{yw}$ and $F_{qd}$ found for the snow-dominated catchments.**

- **We add the Fig. 7b representing the "Fraction of quaternary deposits against mean catchments elevation"** following the suggestion of referee #1: "The objective is to investigate what drives $F_{yw}$ variation with elevation. The authors proposed using a new set of hydrological variables, but what are the relations between these variables with elevation? For example, what are the relations between $F_{SCA}$, $F_{qd}$, $F_{bf}$ with elevation?". Accordingly, **we comment about the Fig. 7b in the text**.

- **We add a new sub-section "4.2.1 The complementarity between the fraction of baseflow ($F_{bf}$) and the young water fraction ($F^*_{yw}$).** We do this because we want to stress more the novelty of this result as suggested by Jana

von Freyberg: "…I would like to encourage the authors to highlight more the novelty of their findings and the scientific contribution of their work".

- **We update the old Fig. 10 with Fig. 9a of the revised manuscript.** In fact, we plot the $F^*_{yw}$ obtained through the "direct-input" approach against the $F_{bf}$. We show that the linear fit of these data is really close to the complementary line.

- **We add Fig. 9b to better show the complementarity of $F^*_{yw}$ and $F_{bf}$ both varying with elevation,** as requested by referee #1: "The objective is to investigate what drives $F_{yw}$ variation with elevation. The authors proposed using a new set of hydrological variables, but what are the relations between these variables with elevation? For example, what are the relations between $F_{SCA}$, $F_{qd}$, $F_{bf}$ with elevation?"

- **We add the sub-section "4.3 Low-flow duration (LFD) and $F^*_{yw}$".** In this Section we discuss the relation between $F^*_{yw}$ and $LFD$ (Fig. 11a) and the $LFD$ variations with elevation (Fig. 11b). We also insert the Fig. 10 showing the relation between LFD and $F_{bf}$.

- **We rewrite section "4.3.1 Fractional Snow Cover Area ($f_{SCA}$) and $F_{yw}$" since we have obtained new results (please note that both $F^*_{yw}$ and $F_{SCA}$ have been calculated in a different way with respect to the submitted version). Accordingly, we update the old Fig. 5 (Fig. 12 of the revised version) and the old Fig. 12a (Fig. 13a of the revised version) with the new values.**

- **We add the Fig. 13b representing the "$F_{SCA}$ against mean elevation"** following the suggestion of referee #1: "The objective is to investigate what drives $F_{yw}$ variation with elevation. The authors proposed using a new set of hydrological variables, but what are the relations between these variables with elevation? For example, what are the relations between $F_{SCA}$, $F_{qd}$, $F_{bf}$ with elevation?" Accordingly, **we comment about the Fig. 13b in the text**.

- **We add the Fig. 14 representing the "LFD against $F_{SCA}$" and we comment about it in the text.**

- **We add a paragraph discussing thoroughly the results obtained for the NBPV (glacier-dominated) catchment** since it was requested by referee #1: "L518: I suppose the fast flow paths are due to the fact that the glacier acts as an impermeable layer and thus promotes rapid overland flow? Please explain what you mean."

- **We add Table 2 reporting $F^*_{yw}$, $F_{qd}$, LFD, $F_{bf}$, $F_{SCA}$ and WFI values for all the study catchments.**

- **We add a new paragraph "4.5 Process interplay along elevation: perceptual model"** explaining the perceptual model.

- **We improve the old Fig.13 (representing the perceptual model) with Fig.15 of the revised manuscript.** We modify the figure according to the Jana von Freyberg comment: This model, presented in Fig. 13, tries to summarize the combined effects of catchment properties (steepness, elevation) with processes (ET, wetness, snowmelt). The resulting figure is very complex and difficult to understand. For instance, if a reader seeks to understand the figure without reading the entire paper, is not clear as to what "increases/decreases with elevation" means. Does this refer to increases/decreases of $F_{yw}$ within a single catchment or between different (high- to low elevation) catchments?"

**1.7    Conclusion: relevant changes**

**We reformulate the Conclusion** to better communicate the novelty of our results in a synthetic and effective way. Consequently, **we rewrite and reorganize this Section**. This is a direct consequence of the major changes applied at the whole manuscript.

- **We remove from the Conclusion the paragraphs related to the new classification scheme (no more considered in the revised paper) and related to the Winter Flow Index**.
- **We clearly communicate the role of the low flow duration in reducing the $F^*_{yw}$ (this was missing in the submitted version)**
- **We add future challenges** such as the collection of detailed geological information, the collection of isotopic data from glacier-dominated systems (which hydrological processes are still poorly understood), the development of new automated techniques to improve the modelling of the groundwater contribution to the stream.

**2    Response to Referees**

**2.1    Response to referee #1**

*The work of Gentile et al. investigated the causes for young water fraction ($F_{yw}$) variations with elevation ($F_{yw}$ is low at high altitudes) in Alpine catchments. The study areas are 27 catchments in Switzerland and Italy. The authors proposed new criteria for catchment classification into different hydro-climatic regimes. To gain insight into the reason for $F_{yw}$ variations with elevation, this author used a new set of hydrological variables, namely the fractional snow cover area ($F_{SCA}$), the fraction of quaternary deposits ($F_{qd}$), and the fraction of baseflow ($F_{bf}$). In general, the idea of this paper about what drives $F_{yw}$ variations with elevations is novel and of interest for understanding the functioning of catchments in Alpine regions as well as for understanding flow and transport in this region and potentially in other areas. However, the methodology and results do not fully support this idea. The text was not well written. Please find my main comments and line-by-line comments below.*

**Dear referee #1,**

**We would like to thank you for the overall positive assessment and the numerous detailed comments, which contributed to our manuscript's improvement considerably.**

**Please find below a point-by-point response to both your main and minor comments. We have incorporated all your constructive feedback in the revised manuscript.**

**Sincerely,**
**The Authors**

**2.1.1     Main comments**

*1)   Why did the authors need to propose a new criterion for catchment classification? The authors used two variables: (1) streamflow ratio between different months and (2) snow cover fraction for the proposed catchment classification, but later they adjusted the threshold of these two variables to have consistent results with Staudinger et al. (2017). Why didn't they just use the method of Staudinger et al. (2017)?*

**We propose a new criterion for the regime classification because our dataset includes catchments outside the Swiss borders (i.e., the four Italian catchments) for which the Weingartner and Aschwanden (1992) and the Staudinger et al. (2017) classification scheme cannot be strictly applied since they were designed for the Swiss hydro-climatic regimes. We have "manually calibrated" the thresholds of $F_{SCA}$ and $Q_{June}/Q_{DJF}$ for classifying catchments in "rainfall-dominated", "hybrid" and "snow-dominated" as in the work of Staudinger et al. (2017). In this way, the classification scheme is "calibrated" on the catchments studied by Staudinger et al. (2017) and we can apply it also outside the Swiss borders.**
**However, according to the referees' comments, we have removed the new criterion for catchment classification from the revised version of the manuscript.** **We use the classification scheme of Stoelzle et al. (2020) which is based on catchments elevation, typical low-flow period, typical snow onset and typical begin of snowmelt. This scheme was used in the past by the authors to classify catchments outside Switzerland (i.e., German catchments).**

*2)   The objective is to investigate what drives $F_{yw}$ variation with elevation. The authors proposed using a new set of hydrological variables, but what are the relations between these variables with elevation? For example, what are the relations between $F_{SCA}$, $F_{qd}$, $F_{bf}$ with elevation? With $F_{SCA}$, I can infer from the text, but it was not explained in the text until the last sections (Section 5.2) of the manuscript. $F_{SCA}$ cannot be directly related to elevation, instead, it needs to be related to the catchment classification then from catchment classification to mean elevation. However, in other areas, can we still relate $F_{SCA}$ to elevation? With the other variables ($F_{qd}$ and $F_{bf}$), it is unclear to me what are their relations to elevations. In addition, $F_{qd}$ does not seems to be a good variable because there is no significant relation between $F_{yw}$ and $F_{qd}$.*

**We added in the revised manuscript, for each variable ($F_{SCA}$, $F_{qd}$ and $F_{bf}$) a figure (Fig. 13b, Fig. 7b, Fig. 9b, respectively) that shows the relation with mean catchments elevation.**

a)  The $F_{SCA}$ increases with the mean catchment elevation in our data set, revealing a positive, statistically significant correlation. This suggests the increasing snowpack persistence with elevation. See Section 4.4 of the revised manuscript.

b)  $F_{qd}$ decreases with the mean catchment elevation in our data set, revealing a negative, statistically significant correlation. This negative correlation reflects the fact that $F_{qd}$ decreases when the mean slope increases (Arnoux et al., 2021) (mean slope increases with mean elevation for the catchments analyzed in this study, as shown in Fig. 2a of the revised manuscript). We use $F_{qd}$ because Arnoux et al. (2021) demonstrated a strong positive correlation between $F_{qd}$ and Winter Flow Index (WFI) highlighting the role of unconsolidated deposits in storing groundwater since this low-flow indicator reflects the groundwater (in terms of age, old water) contribution to the stream (Arnoux et al., 2021; Cochand et al., 2019; Paznekas and Hayashi, 2016). The missing information about the portion of fractured bedrocks, the thickness of Quaternary deposits and the bedrock topography will demand future attention for a complete picture of the role of geology (potentially resulting in a statistically significant correlation with $F^*_{yw}$). See Section 4.1 of the revised manuscript.

c)  $F_{bf}$ against elevation reveals an opposite (and complementary) behavior with respect to $F^*_{yw}$: it decreases until 1500 m a.s.l. and it increases at higher elevations (Fig. 9b of the revised manuscript). This complementarity is an important result for catchments where isotope measurements are missing. In such catchments, the $F^*_{yw}$ could be potentially estimated without the application of the amplitude ratio approach as: $F^*_{yw} \simeq 1 - F_{bf}$.

3)  *The manuscript needs to be restructured and revised. There is a lack of clarification in the text. More description of the study area characteristics is needed. Much of the information provided in Study Sites, and Material and Methods is not relevant (e.g., shape file, detailed source of data, etc.). Instead, citing the sources of the various data (both from individuals and organizations) can be moved to either the Authors' Contributions or Acknowledgements, or in the supporting information Sections or to a table rather than describe them within the text of the article, making it very difficult to read such detailed information. If possible, I would also suggest the authors publish their data in an open repository.*

The whole manuscript has been restructured and revised (See Section 1 of this document to see all the details about the relevant changes). We revised both the fil rouge that allows the reader to follow our reasoning and the language, thus improving the text clarification. As reported in Section 1 of this document, the Study sites Section was completely reorganized and rewritten, and all the data sources have been moved to the *Data availability* Section (at the end of the manuscript), as you have suggested.

1) Title: "$F_{yw}$" could be changed to "young water fraction" for general readers.

   **We change the title from: "What drives $F_{yw}$ variations with elevation in Alpine catchments?" to "Towards a conceptualization of the hydrological processes behind changes of young water fraction with elevation: a focus on mountainous alpine catchments." We have changed "$F_{yw}$" to "young water fraction" for general readers.**

2) L14: "The young water fraction ($F_{yw}$),...., is increasingly used in hydrological studies, replacing the widely used Mean Transit Time, which is subject to aggregation error." This sentence provides misleading information. I think $F_{yw}$ cannot replace Mean Transit Times ($MTT$) since the two characterize different aspects of the transit times, e.g., $F_{yw}$ contains information about the younger part of the TT distribution (how much water in outflow is younger than 0.2 years) while MTT contains information about the whole TT domain. "Aggregation error" could be changed to "aggregation bias".

   **Thanks. Our statement was indeed not precise. We wanted to say that, before the work of Kirchner (2016a, b), the Mean Transit Time ($MTT$), obtained with convolution, was used in catchment intercomparison studies. After that key paper, it is generally replaced with $F^*_{yw}$ that, of course, does not provide the same information as $MTT$. To avoid misunderstanding, we change the sentence simply by writing: "The young water fraction ($F^*_{yw}$), defined as the fraction of catchment outflow with transit times of less than 2-3 months, is increasingly used in hydrological studies that exploit the potential of isotope tracers."**

3) *L33-34: The sentence "..$F_{bf}$, considered...complement of $F_{yw}$" does not clearly show the relation you found between $F_{yw}$ and the baseflow fraction. Please be clearer about what you mean by explicitly saying that $F_{bf}$ is a good proxy for $F_{yw}$ as the higher $F_{yw}$ is, the lower $F_{bf}$.*

   **Thank you. In the revised version we have dedicated a whole sub-section to the (complementary) relation between $F^*_{yw}$ and $F_{bf}$ in which we clearly express the negative correlation between $F^*_{yw}$ and $F_{bf}$, also indicated by the linear fit of the data. For more details, please see Section 4.2.1 of the revised version. In the abstract we write: "In our data set $F_{bf}$ reveals a strong complementarity with $F^*_{yw}$, suggesting that the latter could be estimated as $F^*_{yw} \simeq 1 - F_{bf}$ for catchments in which stable water isotopes measurements are not available."**

4) *L44: "the streamflow is older than the annual snowmelt" is not clear to me, what is the age of streamflow and the age of snowmelt water in this case?*

   **We have clarified this in the revised version. From Introduction: "An early work in the Swiss Alps shows that high celerity is caused by massive meltwater infiltration that pushes out groundwater reserves: streamflow following snowmelt is older than meltwater infiltrated in the current year (Martinec, 1975)"**

5) *L46: why "even"? I would expect exactly that during the absence of rainfall and snowmelt the streamflow is mainly sustained by groundwater.*

**We wanted to underline that the hydrograph separation results show that the hydrograph is generally mainly composed of old water at the peak flow. Of course, during no-rain and no-snowmelt periods we expect that streamflow is mainly sustained by groundwater and this is also confirmed by our results. However, we remove this sentence from Introduction.**

6) *L46-50: The two sentences here do not seem to be connected, one about residence time and the next one about transit times.*

**Of course, the transit time distribution and the residence time distribution are two separate concepts. Nevertheless, the streamwater age is influenced by the storage age depending on how much the storage contributes to the stream. To avoid confusion, we remove the second sentence.**

7) *L53: "Kirchner (2016a, b) proposed a new metric to quantify water age at the catchment scale". I think you are mentioning the $F_{yw}$, I don't think this is "the water age at the catchment scale" but the amount of water with age < 0.2 years. How can we know the "water age at the catchment scale "only based on the amount of water in outflow (discharge) that is < 0.2 years?  For example, if $F_{yw} = 0.2$, what is the "water age at the catchment scale"*

**We wanted to say that *MTTs*, obtained with the classic convolution approach, are no longer used since they are subject to the aggregation bias. However, we can say something reliable about water age using a new metric: the young water fraction, that is calculated at the catchment scale. Of course, $F^*_{yw}$ is not giving the same information of *MTT*. We have rephrased the sentence: "Kirchner (2016a, b) proposed a new metric to quantify the share of catchment outflow with transit times lower than roughly 0.2 years or 2-3 months: the young water fraction."**

8) *L55-58: please revised the sentence structure*

**We have revised the sentence: "$F^*_{yw}$ is increasingly used in hydrological studies because it has the advantage of being free from the aggregation errors inherent to Mean Transit Time (MTT) estimates obtained through the classical convolution approach (Kirchner, 2016a)".**

9) *L58-59: please see my comments on line 14*

**We have rewritten: "Even more so, $F^*_{yw}$ is an informative descriptor of catchment hydrological functions, of nutrients cycles and of pollutant transport (Stockinger et al., 2019; Benettin et al., 2017; Jasechko et al., 2016)"**

10) *L70: "In line with these findings" can be removed because Lutz et al. (2018) did not state that $F_{yw}$ above 1500 m decreases*

**Yes, you are right. Lutz et al. (2018) did not state that $F^*_{yw}$ decreases above 1500 m a.s.l., but they said "In agreement with the results from the global study of European catchments, there is a slight tendency toward smaller $F_{yw}$ values for the subcatchments in the mountainous region". Therefore, to be more precise, we**

write "**Interestingly, in their data set, a statistically significant positive correlation with elevation was obtained after removing from their analysis the five snow-dominated catchments, which revealed the smallest $F*_{yw}$ values (von Freyberg et al., 2018). Likewise, Lutz et al. (2018) estimated $F*_{yw}$ for 24 catchments in Germany and found the smallest values for higher-elevation sites.**"

11) L82-83: "…more efficient groundwater recharge, consequently reducing or increasing the young streamflow…" It is not clear to me, should it be "reducing" only instead of "reducing or increasing"?

**Ceperley et al. (2020) said: "our highest elevation study site (NBPV) deviates from this trend by yielding a higher $F_{yw}$, it is too early to draw the conclusion that low $F_{yw}$ could be due to seasonal versus intermittent snow cover dynamics alone." So, in the submitted version, we remain vague in the Introduction saying "reducing or increasing". From our results presented at the end of the manuscript, we can say that seasonal snow cover favors the groundwater storage emptying during winter and the groundwater storage recharge (because of meltwater infiltration) during summer, thus reducing the young streamflow reaching the stream. In the revised Introduction we write: "However, it is still unclear if seasonal or ephemeral snow cover dynamics can affect the $F*_{yw}$ (Ceperley et al., 2020)".**

12) *L88: "...remarkable fraction of groundwater…" it is a bit vague, could you please be more precise?*

**To be more precise, we have specified the average percentages of groundwater according to the cited works: "Several studies located in the Rocky Mountains and Andes show that, on average, about 47% of groundwater annually sustains the streamflow (Saberi et al., 2019; Somers et al., 2019; Carroll et al., 2018; Harrington et al., 2018; Cowie et al., 2017; Baraer et al., 2015; Gordon et al., 2015; Frisbee et al., 2011; Liu et al., 2004; Clow et al., 2003; Baraer et al., 2009). Similar results are also found in the Himalayas (49%) and the Alps (48%) (Chen et al., 2018; Engel et al., 2016; Käser and Hunkeler, 2016; Williams et al., 2016; Wilson et al., 2016; Andermann et al., 2012)".**

13) *L91-92: "...a dynamic storage contribution to streamflow…" Please clarify this term.*

**With "dynamic storage", we refer to the storage that controls the streamflow dynamics (Staudinger et al., 2017). However, we remove this sentence from the revised version.**

14) L99: Why don't the authors use a new set of hydrological variables ($F_{SCA}$, $F_{qd}$, $F_{bf}$, WFI) in combination with traditional variables to gain new insights into the $F_{yw}$ along elevation gradients?

**We do not know what is meant by "traditional" variables. However, we decided to use variables that were not previously considered for explaining $F*_{yw}$ elevation gradients. This is also because Jasechko et al. (2016) wrote: "Although topographic gradient provides the strongest correlation with young streamflow fractions in our data set, the fraction of unexplained variance is large, suggesting that other variables also play a significant role. We observe no significant correlation between the young streamflow fraction and catchment size, annual precipitation, bedrock porosity, population density, or the fraction of catchment area comprised of pasture land or open water". We specify this in the Introduction of the revised version.**

15) *L104: "…into three hydro-climatic regimes proposing a new criterion of classification…" Why? I think a brief explanation is needed.*

**Please see our answer to your first main comment.**

16) *Sections 2 and 3.1: Both sections about the data (e.g., Section 2: existing data, additional dataset, complete data, and Section 3: discharge data and catchment boundary), why do the authors need two different sections? The data description section (entire section 2) needs to be restructured and revised to make it more concise and clearer. I think this can be done using a table. In the text, the authors could summarize and report key information, so the reader does not have to search through the many sources you have cited. The authors can here focus more on catchment attributes, such as climate (e.g., average annual precipitation and discharge), land use cover, geology, and discharge.*

**Thank you. We apply the suggested structure that improved the paper readability. We have completely restructured Section 2 according to your comments. We have condensed "existing data, additional dataset and complete data" Sections in a single Section (Section 2 of the revised manuscript). Moreover, we have moved all the data sources, reported in Section 3 of submitted version, to the "*Data availability*" Section of the revised version. We have improved the study sites description summarizing the main topographic and hydro-climatic quantities in a Table (Table 1 of the revised version) and some figures (Fig.2 and Fig.3 of the revised version).**

17) *"Furthermore, 21 out of the 22 … (Staudinger et al., 2017)". This part is not relevant in my opinion.*

**You are right. We removed this part in the revised version.**

18) *Two high-elevation catchments …  Arnoux et al., 2021)". This part is not relevant in my opinion.*

**You are right. We removed this part in the revised version.**

19) *L147: In my opinion, the "Complete Dataset" subsection is not necessary. It is sufficient to illustrate the existing data in subsection 2.1 and conclude the section with 2.2.*

**Thank you. we removed the "Complete dataset" subsection as suggested.**

20) *L156-160: Like von Freyberg (2018) … are reported in Table 1. If subsection 2.3 is deleted, move it to 2.2 as the final sentence.*

**We have completely restructured the Section 2.  The paragraph about the hydro-climatic classification is no more at the end of the Section. We have written: "In order to be consistent with previous studies (von Freyberg et al., 2018; Staudinger et al., 2017), we classify the 23 Swiss catchments according to the hydro-climatic regimes proposed by Staudinger et al. (2017) which group the regimes defined by Weingartner and Aschwanden (1992) in three categories: rainfall-dominated (R), hybrid (H) and snow-dominated (S). For the four Italian catchments, where the aforementioned classification schemes cannot be rigorously applied, we use that proposed by Stoelzle et al. (2020).".**

*21)* *Figure 1: the background cannot be easily seen; I think you could replace with a DEM map. In addition, I cannot differentiate between Quaternary deposits and hybrid catchments visually.*

**We have changed Fig.1 according to your suggestions (see new Fig. 1 of the revised manuscript). We do not show the quaternary deposits cover in Fig. 1. This is shown for some representative study catchments in Fig. 6 of the revised manuscript.**

*22)* *Table 1: I am curious to see the relation between average elevations and average slopes for the 27 catchments, is there a positive correlation? (also for average elevation with annual precipitation).*

**We have inserted these two figures in the "Study sites" section. We have commented in the revised version: "The average slope ranges from 4° to 34°, and our study sites reveal an increase of steepness with elevation (Fig. 2a, Fig.2b). Precipitation increases with elevation until 1500 m a.s.l. and it decreases for higher elevations (Fig. 2c, Fig. 2d), highlighting a change of precipitation regime as described by previous studies (Santos et al., 2018)."**

*23)* *Section 3.1: Here, I would expect more description of the discharge dynamics (e.g., giving an order of magnitude to these data by telling what is the annual discharge, whether the runoff is seasonal, etc). I would suggest moving the description of how discharge was measured and derived to the appendix. The source of data could be combined into the same table suggested for section 2 (or move to the appendix or data availability section).*

**Thank you for these suggestions. In the revised version we have moved all the info about the data sources to the "*Data availability*" Section, while we have described discharge dynamics in the "Study sites" Section.**

*24)* *Line 190: I suggest mentioning the study period for the isotope data and $F_{yw}$ for the different study catchments since it is different.*

**Thank you for this suggestion. In the revised version we have reported this info in the last column of Table 1 ("Study sites" Section).**

*25)* *Figure 2: In summer there is a higher average monthly flow from snow-dominated catchments than from rain-dominated ones (due to increased snowmelt, I suppose), and in winter it is the other way around. Please explain this better in the text because it is not clear. In addition, it is not easy to differentiate between the three boxplots, I would suggest having three separated boxplot figures with the same y axis limit. This figure should be described in the text (there is no description of this figure, it was only cited in line 243)*

**We have subdivided the figure in three separated boxplot (see Fig. 3 of the revised version) and we have explained it better in the "Study sites" Section: "Across the three considered streamflow regimes, a shift of the monthly hydrograph peak (computed using discharge data in the PoS) from winter to summer months is observed (Fig. 3): this "flow peak-shifting" is a clear sign of the increasing predominance of snowmelt in the streamflow generation processes.".**

*26)* L197: no comma after "where"

**Thank you.**

27) L221: As I understood from the text (before and after this line), there is indeed a "formal" classification method

**Yes, there is a formal classification method proposed by Weingartner and Aschwanden (1992), but it was designed for Switzerland. The regimes defined by Weingartner and Aschwanden (1992) were grouped by Staudinger et al. (2017) in three categories: rainfall-dominated, hybrid and snow-dominated. Searching in scientific literature, after the interactive discussion phase, we have found that Stoelzle et al. (2020) proposed a classification method based on topographical and streamflow data and that this method was used in the past to classify catchments outside Switzerland. Thus, we decide to use this formal classification method to classify the Italian catchments of our dataset.**

28) *Section 3.3: After reading the entire manuscript up to section 3.3. I am not clear why the authors need to classify streamflow into three regimes and why the classifier should be based on snow-related characteristics (e.g., snow cover area).*

**Please see our answer to your first main comment.**

29) *L240: should it be "it is expressed in mm per unit area and time step"?*

**We have removed the Section "3.3 A new hydro-climatic regime classification: the classifiers" since we use literature classification schemes.**

30) *L251: "...more than weekly..." do you mean biweekly?*

**We have rewritten: "Temporally, this relatively recent satellite has increased the visitation frequency to a sub-weekly temporal resolution and increased the spatial resolution to 20 m for snow cover (Gascoin et al., 2019)."**

31) *Eq: (5) the denominator ($N_{tot} - N_{clouds}$): This could result in an overestimation of $f_{SCA}$. What is the maximum fraction of cloud cover in these images?*

**We follow the approach of Hofmeister et al. (2022) and Di Marco et al. (2020) that define $f_{SCA}$ as $N_{snow}/(N_{tot}-N_{clouds})$. They did not comment on the effect (i.e., underestimation or overestimation) of the mathematical expression of $f_{SCA}$ on their results. If the two detection algorithms (snow detection and cloud detection) would work with a 100% accuracy, values greater than 1 cannot be encountered. In fact, the maximum cloud cover fraction can also be very close to 1 in some dates (e.g., > 90% as encountered in our data set), but if the snow detection algorithm works well $N_{snow}$ will be at most "complementary" to $N_{clouds}$ (i.e., $N_{snow} + N_{clouds} = N_{tot}$) and $f_{SCA}$ will be at most 1. Sometimes these algorithms can result in a misclassification of pixels and $f_{SCA}$ values > 1 can be encountered: i.e., $N_{snow} > (N_{tot}-N_{clouds})$. Our approach was to set $f_{SCA} = 1$ if $f_{SCA} > 1$. We deepen that this approach can overestimate the $f_{SCA}$. Thus, we propose a new approach: if $f_{SCA} > 1$ we calculate $f_{SCA}$ as $N_{snow}/N_{tot}$ since this is the only heuristic solution that guarantees no overestimation. Please see Section 3.2 of the revised version for more details.**

32) L276-279: The error in $f_{SCA}$ is still there with the "moving window" approach, it is just smoothed. Anyways, at the end, you calculated the average $f_{SCA}$ over the whole period so applying "moving average on a window" does not have any effect?

**Of course, the average of the series after the application of the moving average (that is what we call $F_{SCA}$ in the submitted version) is not the same as the average over the original series. However, in the revised manuscript we change the Methodology and we do not apply the moving average. Please see Section 3.2 of the revised version for more details.**

33) *L282-289: "Some authors have revealed ... $F_{yw}$ in Alpine catchments". This is more suitable for Introduction than Methodology. In addition, what is "key possibility"? Does it mean "high possibility"*

**Thank you. We have moved this part to the Introduction. With "key possibility" we wanted to express "the importance" of Quaternary deposits (moraines, alluvium, and talus) in storing groundwater. We have simply written in the Introduction: "Some authors have revealed the possibility of quaternary deposits (e.g., talus, moraine, alluvium) to store groundwater in high-elevation alpine catchments (Arnoux et al., 2021; Hayashi, 2020; Christensen et al., 2020)."**

34) L292: … 23 Swiss catchments … Is $F_{qd}$ calculated only for 23 sub-catchments, while *WFI* and $F_{bf}$ for all 27? Why? How does it affect the interpretation of the results? Be clearer about which indices are available for each study site.

**We have written: "Operatively, for the 23 Swiss catchments of our dataset, we calculate the portion of the catchment area occupied by Quaternary deposits using the Geological Atlas of Switzerland…" The number "23" refers to catchments located in Switzerland because, for these catchments, we have used the GeoCover dataset for estimating the area covered by Quaternary deposits; for the remaining four Italian catchments we have used regional geological dataset. $F_{qd}$, *WFI* and $F_{bf}$ are calculated for all the 27 catchments: see also Table 2 of the revised manuscript.**

35) *L299-301: For the DOR and SOU ... provided by Dr. Giulia Zuecco. This part is not relevant here, should be moved to the data section.*

**We have moved this part to the *Data availability* section as you have suggested.**

36) L315-318: "For VdN, NBPV and BCC catchments we consider the time windows … we consider discharge measurements in the period November 2017 - January 2022". I think you should indicate at the beginning the different study periods, because it is confusing to read a lot of data (e.g., stable isotopes of water, $F_{yw}$, streamflow...) and indices (e.g., $F_{qd}$, *WFI*, $F_{bf}$) for your methodology and find out that your study areas were analyzed in different periods. You should say this explicitly each time you mention a new data item or index or create a table in which you explain it.

**Thank you for this recommendation. We have inserted this info in Table 1 reported in the "Study sites" Section of the revised version.**

590     *37) Section 4.1. I think this can be moved to the data section or supporting information, as this is only for 2 catchments.*

**Thank you for this suggestion. We have moved this subsection to the Supplementary Material.**

*38) L334-335: "these have the same names as the ones proposed by Staudinger et al. (2017 but the classification is not based on the same criteria" why? I think should be explained earlier in the methodology section.*

595     *39) L336-337: "In order to achieve a classification as consistent as possible with that of Staudinger et al. (2017), but based on these two variables, we propose the thresholds presented in Table 2:" I cannot understand why. If the authors want to have consistent results with Staudinger et al. (2017), why did not they use the method proposed by Staudinger et al. (2017)?*

**Regarding comments 38 and 39, please see our answer to your first main comment.**

600     *40) L345-346: "Following our classification scheme, ... and 9 snow-dominated catchments". How do you compensate for the fact that the catchments data belong to different periods?*

*41) L353: "snow-regime" should be explained here*

*42) L354: "for the first order estimate of the second classifier" what does it mean?*

**Regarding comments 40,41 and 42, we have removed Section "4.2 The new hydro-climatic regime**
605     **classification" since we use literature classification schemes.**

*43) "Section 4.3: New explanatory variables for the $F_{yw}$ elevation gradients" I would expect all subsections in section 4.3 will use variables that are related to elevation to explain the relation between $F_{yw}$ with elevation. However, I cannot see what is the relation between the variables in the section title (e.g., Section 4.3.1. Fractional Snow Cover Area ($f_{SCA}$) and $F_{yw}$) and elevation (Please also see my main comments)*

610     **Thank you for this comment. Following this comment, we have related all the variables with elevation and all the new figures are explained in the text. Please see in the "4 Results and Discussion – Towards a harmonious and exhaustive framework of the hydrological processes that drive the young water fraction variations with elevation." Section of the revised manuscript Fig. 7b, Fig. 9b, Fig. 11b, Fig. 13b.**

44) L361-368: part of this information was already described in the introduction, can be removed here or merged into
615     the introduction.

**Thank you for this suggestion. We have merged this part in the Introduction: "These results are partially consistent with those of Jasechko et al. (2016): based on the analysis of 254 watersheds worldwide, their work revealed a reduction of $F*_{yw}$ in mountainous, steeper terrains. This could be related to deep vertical infiltration caused by fractures generated by high rock stress in complex terrain morphologies or by freely**
620     **draining soils (i.e., cambisols and luvisols), both associated to high-elevation environments (Lutz et al., 2018; Jasechko et al., 2016; Gleeson et al., 2014). In addition, the higher the topographic roughness is, the longer are the flow paths, with a consequent rise of transit times (Gleeson and Manning, 2008; Frisbee et al., 2011; Jasechko et al., 2016)."**

*45) L389-391: Our results ... for hybrid catchments (median $F_{yw}$ of 0.32) ... Why are there these differences? I suggest arguing and explaining them.*

**Thank you. We have explained the differences in $F^*_{yw}$ values with varying hydro-climatic regimes in Sections 4.2, 4.3 and 4.4 of the revised manuscript.**

46) Figure 4a can be moved to the data section, figure caption: "the horizontal bars correspond to +/- standard deviation" of slope or elevation?

**Thank you for the suggestion. We have moved the figure to the "Study sites" section (see Fig. 2a of the revised manuscript), but we have inverted the axes. The horizontal bars of the "old" Fig. 4a correspond to +/- standard deviation of slope. In Fig. 2a of the revised manuscript the standard deviation of slope is represented by the vertical bars.**

*47) L367: "Despite this" why should an increase in slope with elevation result in a correlation between $F_{yw}$ and slope?*

**Thank you for this comment. This is probably a typo. We could expect a negative correlation between $F^*_{yw}$ and slope because of the results of Jasechko et al. (2016): "...topographic gradient provides the strongest (negative) correlation with young streamflow fractions in our data set…"**

48) L393: "lowering" could be changed to "decreasing of $F_{yw}$ with increasing $F_{SCA}$"

**Thank you.**

49) L484-486: Therefore, it is more likely that … possibly ephemeral, snowpack. I do not see a connection between these two statements. If you are saying that lower-order (i.e., more downstream) channels release greater amounts of old water than higher-order (i.e., more upstream) channels, why do you say that water age decreases with elevation? Please clarify this point.

**We have clarified this point in Section 4.4 of the revised manuscript. L484-486 of the preprint refer to the discussion about the rising limb (when $F_{SCA}$ remains below roughly 0.3) of the $F^*_{yw}$ vs $F_{SCA}$ bell-shaped relationship (see Fig. 13a of the revised manuscript). In other words, for rainfall-dominated and hybrid catchments with ephemeral snowpack, $F^*_{yw}$ increases with elevation because:**

- **The increase of precipitation with elevation and the reduction of evaporation with elevation, due to reduced temperatures, promote wetness conditions that increase $F^*_{yw}$.**

- **The limited number of snowfall days and the mid-winter melt (due to an ephemeral snowpack) reduce the snow accumulation. Such a snowpack does not protect the underlying soils from freezing thereby inhibiting infiltration and favoring rapid flow paths during mid-winter melt/rainfall events, with subsequent increase of $F^*_{yw}$.**

- **These short-lived snowpacks melt during the winter season resulting in little delay between precipitation input and melt (i.e., no water aging in the snowpack).**

- **Considering the Strahler's stream order, lower order channels, upstream, are more rarely activated (e.g., because of intense rainfall/snowmelt events) draining young water. Vice versa, higher order channels,**

**downstream, are more often active (e.g., because of low/medium rainfall/snowmelt events) and inclined to drain more old water. Please note that we have removed this last point from the paper because we do not want to generate confusion: we are studying $F*_{yw}$ with elevation among different catchments and not within a single catchment. Probably, considerations about the age of water flowing in higher/lower order channels is more suitable if looking for water age variations within a single catchment.**

-

*50) L493: "a persistent, deep snowpack can promote deep vertical infiltration by insulating the soil and thereby preventing freezing" do you mean this happens in winter? If in winter, there might be only snow, how can it be melted and promote deep vertical infiltration? Where is the source of water for vertical infiltration?*

**The persistent and deep snowpack prevents soil freezing during winter so that during snowmelt onset in spring, meltwater can infiltrate and recharge the groundwater storage. We clarify this in Section 4.4 of the revised manuscript.**

*51) L495: what's a temporal concentration? Make it clearer.*

**We have written: "The resulting effect on water partitioning between the surface and the subsurface should be analyzed considering the temporal concentration of water input on the snowmelt period, but this remains largely unexplored to date (Rey et al., 2021)"**

**Temporal concentration: the time-interval in which the snowmelt enters the system as water input.**

*52) L499-501: This is for the karst area, how relevant is it for your area?*

**In our dataset we have two dolomitic catchments: BCC and OVA. We specify this in Table 1 and in Section 4.4 of the revised manuscript.**

*53) L518: I suppose the fast flow paths are due to the fact that the glacier acts as an impermeable layer and thus promotes rapid overland flow? Please explain what you mean.*

**This comment refers to a possible explanation of the high $F_{yw}$ for the glacier-covered catchment. The "old" text reads as "Such (glacier-covered) catchments could show fast flow paths and small storages as e.g. discussed in the work of Jansson et al. (2003), reviewing glacier-dominated environments. Moreover, reduced baseflow during winter can be related to increasingly high temperatures causing the glaciers retreat, thus reducing, and anticipating the glacier melt fluxes that possibly recharge groundwater (Hayashi, 2020)".**

**We have rewritten (see Section 4.4): "The high $F*_{yw}$ of the high elevation glacier-covered (42%) catchment can be explained considering that the glacier-melt produces high amounts of streamflow that transit the glacier-system very quickly during the summer, given generally fast englacial and subglacial flow paths and the often limited water storage capacity in the glacier forefield (Müller et al., 2022; Saberi et al., 2019; Jansson et al., 2003). Schmieder et al. (2019) also found a high young water fraction in an Austrian glacier-covered (35%) catchment leading them to the conclusion that the basin behaves like a 'Teflon basin' with fast transmitted ice melt, also if this behavior is differentiated in space."**

*54) Figure 13: Which subfigure is for lower altitudes (< 1500m) and which one is for higher altitudes? Figure caption: the word "panels" can be removed because I thought a panel always consists of two subfigures (e.g., the lower panel contains two subfigures c,d)*

**Subfigures titled with "Ephemeral snowpack" refer to lower elevations, while subfigures titled with "Seasonal snowpack" are for higher elevations. Please note that we have deeply modified Fig. 13 in the revised version according to Jana von Freyberg comments. See Fig. 15 of the revised manuscript.**

*55) L531: "unconsolidated sediments are not the only…" could be changed to "water storage in unconsolidated seidments are not the only …"*

**Thank you.**

**2.2 Response to Jana von Freyberg**

**Dear Jana von Freyberg,**

**thank you for your care and attention during your reading of the manuscript, your positive remarks and your suggestions that helped to improve the work considerably. We have incorporated all your constructive feedback in the revised version of the manuscript.**

 **Please find below a point-by-point response to all your comments.**

**With kind regards,**
**The Authors**

**2.2.1 General comments**

*Gentile et al. address the scientific questions of "… what drives $F_{yw}$ variations with elevation in Alpine catchments clarifying why $F_{yw}$ is low at high altitudes» (L20). For this, the authors combine existing and new $F_{yw}$ values from Switzerland and Italy and compare them with several other variables that describe snow cover, baseflow conditions, and geology. From these comparisons the authors develop a perceptual model, suggesting that a longer persistence of the seasonal snowpack results in deeper groundwater flow paths and thus smaller $F_{yw}$ values, in contrast to hybrid catchments with ephemeral snow packs. The authors also present a new classification scheme to identify a catchment's hydro-climatic regime. The analysis of the used data is thorough and most figures are clear and informative. The analysis of satellite images to explore the linkages between snow cover duration and $F_{yw}$ are certainly interesting.*

**Thanks for the positive overall assessment.**

1) *However, I would like to encourage the authors to highlight more the novelty of their findings and the scientific contribution of their work, considering that they cite several papers in which comparable analyses have been carried out and similar conclusions (with respect to flow and storage processes) have been reached. I think that the research objectives (or research questions) should be formulated more explicitly in the Introduction in order to guide the following analysis. It is not clear whether the authors attempt to explain the scatter in the $F_{yw}$-gradient relationship (L76), the low $F_{yw}$ values in steep and/or high-elevation catchments (L79), or both.*

**We make the research objectives and the novelty of our work clearer in the Abstract and in the Introduction.**

**For example, in the Abstract: "Past works have shown surprising evidence that mountainous catchments worldwide yield low $F^*_{yw}$. These low values have been partially explained by isolated hydrological processes, including deep vertical infiltration and long groundwater flow paths. However, a harmonious framework illustrating the relevant mechanisms leading to a low $F^*_{yw}$ in mountainous catchments is missing."**

**"The main aim of this paper is to give an overview of what drives the $F^*_{yw}$ variations according to elevation, thus clarifying why it generally decreases at high elevation."…**

**"As a conclusion, we develop a perceptual model that integrates all the results of our analysis to describe a framework for how hydrological processes control $F^*_{yw}$ according to elevation, laying the foundations for an improvement of the theory-driven models."**

**Please see the "1 Introduction" Section of the revised manuscript to observe all the relevant changes about the new formulation of the research objectives (linked to a specific research gap) and how we intend to fill this gap using some new (previously unconsidered) variables ($F_{qd}$, $F_{bf}$, LFD, $F_{SCA}$) reflecting some specific hydrological processes.**

**In the Introduction, we guide the reader to the importance of understanding such processes for explaining the changes of $F^*_{yw}$ with elevation, mainly focusing on the reasons why such processes can be hidden behind the low $F^*_{yw}$ at high elevations.**

**The fraction of Quaternary deposits, the low-flow Duration and the snowpack ephemerality have never been used to explain the low $F^*_{yw}$ at high elevations. Thus, the relations between $F^*_{yw}$ and such variables are novel. $F_{bf}$ was indirectly used in past studies, but it was estimated with different methodologies. With the methodology presented in this paper, i.e., baseflow filter of Duncan (2019), we achieve a complementary relationship between $F_{bf}$ and $F^*_{yw}$ suggesting that $F^*_{yw} \simeq 1 - F_{bf}$. We can say that this is a by-product of our work (i.e., our main aim was not to find an alternative method to estimated $F^*_{yw}$). However, this is a novel**

result since it suggests that $F^*_{yw}$ could potentially be estimated for catchments in which stable water isotopes measurements are not available.

Moreover, the research objectives are also clearly expressed in the new title: "Towards a conceptualization of the hydrological processes behind changes of young water fraction with elevation: a focus on mountainous alpine catchments"

2) *L156 "we classify the catchments in the three hydro-climatic regimes (snow-dominated, hybrid and rainfall-dominated) proposed by Staudinger et al. (2017), but we introduce a new formal criterion of classification": Why is a new definition of the catchments' hydro-climatic regimes needed? As far as I can tell, only two catchments, BIB and GUE, were newly classified. The new sites outside of Switzerland could have easily been categorized as hybrid or snow-dominated based on their streamflow and topographical data. Furthermore, the discussion of this new classification scheme (Sect. 4.2 and 5.1) somewhat distracts from the main topic of the paper, which is the investigation of small $F_{yw}$ in high-elevation catchments.*

Thank you for this comment, which was already made by reviewer 1. We copy here the answer that we gave to reviewer 1:

"We propose a new criterion for the regime classification because our dataset includes catchments outside the Swiss borders (i.e., the four Italian catchments) for which the Weingartner and Aschwanden (1992) and the Staudinger et al. (2017) classification scheme cannot be strictly applied since they were designed for the Swiss hydro-climatic regimes. We have "manually calibrated" the thresholds of $F_{SCA}$ and $Q_{June}/Q_{DJF}$ for classifying catchments in "rainfall-dominated", "hybrid" and "snow-dominated" as in the work of Staudinger et al. (2017). In this way, the classification scheme is "calibrated" on the catchments studied by Staudinger et al. (2017) and we can apply it also outside the Swiss borders. However, according to the referees' comments, we have removed the new criterion for catchment classification from the revised version of the manuscript. We use the classification scheme of Stoelzle et al. (2020) which is based on catchments elevation, typical low-flow period, typical snow onset and typical begin of snowmelt. This scheme was used in the past by the authors to classify catchments outside Switzerland (i.e., German catchments)."

3) *I was surprised to see that the authors did not include annual or seasonal precipitation in their analysis. This variable should be tightly related to $F_{bf}$ and $F_{SCA}$. Annual precipitation is also very low at some Swiss high-elevation sites, which would also explain why $F_{yw}$ is low there. What is the reason for not considering precipitation at all?*

Thank you for this comment. We have clearly explained why we have not considered precipitation in the Introduction: "A special focus of our work is on variables that were not previously considered for explaining

elevation gradients of young water fractions. We namely exclude catchment size, annual precipitation, bedrock porosity, pasture cover, open water cover that have been discussed and shown to have little correlation in the work of Jasechko et al. (2016).

A special case in terms of explanatory variables is mean annual precipitation: Jasechko et al. (2016) in their worldwide study did not observe any significant correlation between the $F^*_{yw}$ and annual precipitation. Lutz et al. (2018) found, based on 24 catchments in Germany, that $F^*_{yw}$ decreases with increasing mean annual precipitation. In contrast, in the relatively wet rainfall-dominated and hybrid catchments studied by von Freyberg et al. (2018), $F^*_{yw}$ was shown to increase with precipitation, which in turn both increase with elevation. In their study, discharge (unsurprisingly correlated with precipitation) was considered as a proxy of catchment wetness, which favours rapid flow paths and thereby increases $F^*_{yw}$ (von Freyberg et al., 2018). In snow-dominated systems, the use of mean annual precipitation as a proxy for catchment wetness could be misleading because the seasonal snowpack leads to a very dry period of the year despite the high *solid* water input. In other words, the temporal concentration of the liquid water input is the relevant variable. Indeed, the saturation of the system (i.e., high wetness conditions) can be observed also when the annual precipitation is low if a large volume of water (stored in the snowpack) is released in a relatively concentrated time interval. Indeed, despite precipitations, and correspondingly discharges, are relatively higher in snow-dominated than in rainfall-dominated catchments, $F^*_{yw}$ is generally lower in snow-dominated systems that are potentially wetter than rainfall-dominated ones. This suggests that the only precipitation can only partially explain the variations of $F^*_{yw}$ and that other variables should be put under observation."

To be precise, in the Discussion we use precipitation to explain why $F^*_{yw}$ increases with elevation until 1500 m a.s.l. (i.e., for rainfall-dominated and hybrid catchments), but we do not use precipitation for explaining why $F^*_{yw}$ is low at higher elevations (i.e., for snow-dominated catchments).

4)  *The important aspect of snow pack storage in high-elevation, snow-dominated catchments, which the authors only touch on in the Conclusions section, should instead be brought up much earlier in the manuscript. In fact, it has been discussed already in another paper: «Another analytical decision that affects the interpretation of $F^*_{yw}$ and $F_{yw}$ relates to whether snowpack storage is considered to be part of catchment storage, or not. If one measures precipitation to the snow surface as the catchment input, then snowpack accumulation and melt are implicitly included in catchment storage (e.g. Staudinger et al., 2017). In this case, comparisons of seasonal cycles in precipitation and streamflow should reflect the young water fraction resulting from the combination of snowpack and subsurface storage. Alternatively, if one uses precipitation and snowmelt arriving at the soil surface as the catchment input (for example, with melt pan lysimeters, or modelled snowpack out- flows), then snowpack accumulation and melt are implicitly excluded from catchment storage. In this case, comparisons of seasonal cycles in streamflow and sub-snowpack catchment input should reflect the young water fraction resulting from subsurface storage alone. Because the total catchment storage in the first case (including snowpack storage) is larger than the*

*subsurface storage alone, the resulting young water fractions are expected to be smaller.» (von Freyberg et al., 2018). In addition, in high-elevation catchments with perennial snow packs, snowmelt in spring and summer is likely to be older than 2-3 months (because the snow fell more than 3 months before the melt occurs). As a result, although summer discharge might be high it will consist mainly of old snowmelt and groundwater rather than recent rainfall (i.e., $F_{yw}$ is small). In hybrid and rain-dominated catchments, streamflow receives relatively more young water from young snow packs and recent rainfall events, respectively.*

**Thank you very much for this comment. Following your suggestion, we have inserted a paragraph in the Introduction in which we clearly address the age of the snowmelt (we totally agree with you), the importance of snowpack storage and its role about the estimation of $F^*_{yw}$, specifying that it was previously addressed by von Freyberg et al. (2018). We have also introduced in this Section the difference between "direct" and "delayed" input and that, from a water storage perspective, and water age perspective, the snowpack and the groundwater storage can be considered as a single entity, thus they both constitute the catchment storage. This aspect was also deepened in Section "3.1 Young water fraction estimation from seasonal cycles of stable water isotopes in precipitation and streamwater: the "direct" input of the revised version.**

**We would like to specify that in snow-dominated systems there is a thin line between old snowmelt and groundwater. We know that recent snowmelt is likely to be older than 2-3 months and we also know that, according to several papers, recent snowmelt has a key role in recharging the groundwater storage during summer. Therefore, groundwater storage is assumed to be mainly composed of old snowmelt.**

5) *The authors seem to overlook this storage aspect of the snowpack and instead focus mainly on the groundwater contribution to streamflow (L82).*

**Thanks for this important comment. In addition to our above answers, it is of prime importance to point out here and in the revised paper (see Section 4.2) that large parts of the snowmelt actually transit through the groundwater storage: i) the very high baseflow in high mountain catchments during summer is a direct sign of this fact. ii) groundwater in such catchments often has the isotopic signature of snowmelt (Michelon et al., 2022; Pavlovskii et al., 2018).**

6) *A main finding of the paper is a strong negative correlation between the baseflow fraction $F_{bf}$ and $F_{yw}$ (Sect. 4.3.3, Fig. 10) from which the authors derive several statements which I'd like to comment on (Sect. 5.4):*

7) *L553: "We find the highest $F_{bf}$ for snow-dominated catchments confirming the presence of high subsurface storage, contributing to streams, in high-elevation catchments». I would include the snowpack as part of the storage here because winter precipitation is stored in the snowpack until summer when it recharges aquifers or runs off into the stream.*

**Thank you for this comment. You are right that in the preprint we have not pointed out the fact that, in our analysis, we are assuming that the snowpack is part of the catchment storage. In the revised version this is**

 **addressed firstly in the Introduction and, secondly, in Section "3.1 Young water fraction estimation from seasonal cycles of stable water isotopes in precipitation and streamwater: the "direct" input". Accordingly, the $F^*_{yw}$ of the revised version has been obtained through the "direct input" approach.**

8) *L554.: "Moreover, the annual baseflow is strongly positively correlated with the $F_{SCA}$ ($\rho_{Spearman}$= 0.81 p-value < 0.01) suggesting a major groundwater contribution with increasing snow cover persistency (Fig. S6)». This depends strongly on your baseflow estimation method.*

   **Calculating the annual baseflow through another baseflow filter, e.g., the Lyne and Hollick (1979), the positive correlation between Annual baseflow and $F_{SCA}$ does not change (please see Fig. 3a, Fig. 3b of the public 'Reply on RC2').**

9) *Further, increasing baseflow and snow cover persistency are both results of increasing catchment elevation and/or annual precipitation. Thus, baseflow cannot simply be linked to snow cover persistency.*

   **We would like to underline that we do not pretend that snow cover persistence alone explains baseflow, we have simply shown the statistical link. However, we have decided to remove this sentence in the revised version.**

10) *L570 (&L37): "Therefore, we can conclude that the contribution of groundwater storage to streamflow, which is driven by snowpack duration, can be considered as the best explanatory variable of the $F_{yw}$ elevation gradients." Again, I would rather argue that not snowpack duration but rather storage capacity (both in the subsurface and the snowpack) together with the hydro-climatic conditions (P-ET) and catchment properties affect the contribution of old water (not necessarily only groundwater) to streamflow, and thus $F_{yw}$. In high-elevation catchments, the snowpack can function like a subsurface water storage that releases (>3 months) old water during the melting season. This old water is meltwater, not groundwater and I suspect that the baseflow separation method used in this paper is not able to differentiate between the two.*

   **We made the link among snowpack storage, groundwater storage and catchment storage clearer in the revised version.**

   **Snowpack releases old water during the melting season. Such meltwater preferentially infiltrates so that the groundwater storage is mainly composed by old snowmelt. This is why we can consider the snowpack and the groundwater storage as a single entity: the catchment storage. We quantify the contribution of the catchment storage (i.e., snowpack storage + groundwater storage) to the stream through the baseflow (which cannot distinguish between snowmelt and groundwater). Specifically, with the help of the baseflow, we can quantify the share of streamflow that is due to the catchment storage release. Of course, the share of snowmelt (with age > 3 months) that flows off quickly will not show up in the baseflow.**

   **In snow-dominated catchments, during the winter season, the baseflow measures the groundwater storage contribution to streamflow (but not the new snowpack contribution since such snowpack does not melt during winter). Such groundwater storage is mainly composed by the old snowmelt (previously infiltrated).**

**The duration of the groundwater-sustained (or old water sustained) winter low-flow depends on the snowpack persistence in snow-dominated systems. The longer the persistence the longer is the contribution from the groundwater (or old water) storage to the stream. This is the reason why the low-flow Duration (LFD) explains the reduced $F^*_{yw}$ in high-elevation catchments.**

**These concepts and reasoning have been thoroughly addressed along the entire revised manuscript, especially in the "Results and Discussion" Section.**

11) *Based on the analysis of slope data the authors conclude that (L370) "… that there is an increasing rate of infiltration when the hydro-climatic regime transitions from hybrid to snow-dominated.". I don't think that this statement is well supported by using slope data in Fig. 4 (no data on infiltration is provided). Instead, the only conclusion that can be drawn from the data presented in this manuscript is that the hybrid catchments receive more precipitation than the rain-dominated catchments (L478), resulting in more recent precipitation becoming streamflow, i.e. higher $F_{yw}$ values. This is analogous to earlier findings in von Freyberg et al. (2018): "… young water fractions tend to be highest in humid catchments where prompt runoff response is facilitated by fast flow paths and/or high-intensity precipitation events."*

**Thank you for this comment. Yes, we do not have infiltration data and probably this conclusion only using slope data cannot be well supported, also if Jasechko et al. (2016) concluded that in steeper terrain the low $F^*_{yw}$ could be caused by rapid percolation through fractures and deep flow paths (as also reported in Lutz et al. 2018). We have removed this part in the revised manuscript.**

12) *One outcome is a "perceptual model of how snow persistency explains $F_{yw}$ during winter and summer along topographic gradients". This model, presented in Fig. 13, tries to summarize the combined effects of catchment properties (steepness, elevation) with processes (ET, wetness, snowmelt). The resulting figure is very complex and difficult to understand. For instance, if a reader seeks to understand the figure without reading the entire paper, is not clear as to what "increases/decreases with elevation" means. Does this refer to increases/decreases of Fyw within a single catchment or between different (high- to low elevation) catchments?*

**We realized that Fig. 13 can be misleading since only a single catchment is represented. Therefore, we improve Fig. 13 to better reveal our "step forward" regarding the hydrological processes behind the $F^*_{yw}$ variations between different catchments. Please see Fig.15 of the revised manuscript.**

**2.2.2   Specific comments**

1) *The title of the manuscript does not well reflect the content of the paper. It rather gives the impression that $F_{yw}$ was studied along elevation gradients within (individual) catchments. In addition, the term "Alpine" suggests that solely mountainous catchments within the Alps mountain range were considered, however, catchments such as*

*ERG, AAB and MEN are located in the Jura Mountains and Swiss Plateau, respectively. It would be nice to define early on what is meant here by Alpine, given that the Introduction starts with the general statement (L41) "Alpine catchments are assumed to generate a high share of surface runoff ..."*

**Thank you for this comment. We report here the relevant changes made in the title described in Section 1.1 of this document:**

**"Starting from this comment we have decided to change the paper title:**

**From:** **"***What drives $\underline{F_{yw}}$ variations with elevation in $\underline{Alpine}$ catchments?***"**

**To:** **"***$\underline{Towards\ a\ conceptualization}$ of the hydrological processes behind changes of $\underline{young\ water\ fraction}$ with elevation: $\underline{a\ focus}$ on $\underline{mountainous\ alpine}$ catchments.***".**

**We think that this title reflects much better the content of the paper, which is concluded with the presentation of a perceptual model (the first step before conceptualization) that integrates all the results of our analysis to describe a framework for how hydrological processes control the $F^*_{yw}$ according to elevation. In the title, we underline that the focus of the paper is on mountainous alpine catchments. We use the term "alpine" instead of "Alpine" to refer to the typical hydro-climatic conditions of a mountain climate. In other words, we are not specifically referring to the Alps Mountain range. Finally, we have changed "$F_{yw}$" to "young water fraction" in the title as suggested by the referee #1: "… "$F_{yw}$" could be changed to "young water fraction" for general readers".**

2) *Ideally, the time periods that were used to calculate the various metrics should be the same as those of the isotope data used to calculate $F_{yw}$. As far as I can tell, this has been considered only for $F_{bf}$, whereas $F_{SCA}$ was determined based on satellite data from 2017-2021. For WFI and $Q_{June}/Q_{DJF}$, no information is provided. The $F_{yw}$ values in von Freyberg et al. (2018) only cover the time periods 2010-2015, which is not even overlapping with the satellite images used to determine $F_{SCA}$. I would like to encourage the authors to compare data only from the same time periods, especially when these periods included extremely dry/wet climatic conditions.*

**We agree. In fact, $F_{bf}$ and WFI have been calculated in the same time period of isotope sampling (called PoS in the revised manuscript): we specify this in the revised manuscript (Table 1 and Section "3.3 Fraction of Quaternary deposits, Low Flow duration and the groundwater contribution to the stream"). $F_{SCA}$ is calculated in the period 2017-2021 simply because of the availability of the Sentinel-2 satellite images (there are no Sentinel-2 images in the period 2010-2015): we specify this in the revised version (Section "3.2 Snow cover persistence quantified through the average fractional Snow Cover Area ($F_{SCA}$)" of the revised version).**

For the $Q_{June}/Q_{DJF}$, we used a long-term average since this ratio was used for a classification purpose. However, we do not have more $Q_{June}/Q_{DJF}$ in the revised manuscript since we remove our "new classification scheme" as suggested by the referees.

3) *The terms elevation and steepness should not be used synonymously, as in L361:" Initial evidence of low $F_{yw}$ in high-elevation catchments is given in the work of Jasechko et al. (2016). Based on the analysis of 254 worldwide watersheds, their work reveals a reduction of $F_{yw}$ in steeper terrains." Also, low elevation (rainfall dominated) catchments can be very steep, and there surely exist high-elevation (snow dominated) catchments with flat topography.*

We agree. We have carefully reviewed the language of the revised version.

4) *When I look closer at the $f_{SCA}$ time series (Fig. 5), I wonder how it is possible that the AAC catchment at around 500m asl. was almost entirely snow covered in summers of 2018 and 2020 ($f_{SCA}$ around 1)? The same is true for the catchments BIB and ERL where the snow cover usually disappears by June each year. Can it be that $f_{SCA}$ tends to be over-estimated with your approach?*

We report here the answer we give to the minor comment n°31 of referee #1:

"We follow the approach of Hofmeister et al. (2022) and Di Marco et al. (2020) that define $f_{SCA}$ as $N_{snow}/(N_{tot}-N_{clouds})$. They did not comment on the effect (i.e., underestimation or overestimation) of the mathematical expression of $f_{SCA}$ on their results. If the two detection algorithms (snow detection and cloud detection) would work with a 100% accuracy, values greater than 1 cannot be encountered. In fact, the maximum cloud cover fraction can also be very close to 1 in some dates (e.g., > 90% as encountered in our data set), but if the snow detection algorithm works well $N_{snow}$ will be at most "complementary" to $N_{clouds}$ (i.e., $N_{snow} + N_{clouds} = N_{tot}$) and $f_{SCA}$ will be at most 1. Sometimes these algorithms can result in a misclassification of pixels and $f_{SCA}$ values > 1 can be encountered: i.e., $N_{snow} > (N_{tot}-N_{clouds})$. Our approach was to set $f_{SCA} = 1$ if $f_{SCA} > 1$. We deepen that this approach can overestimate the $f_{SCA}$. Thus, we propose a new approach: if $f_{SCA} > 1$ we calculate $f_{SCA}$ as $N_{snow}/N_{tot}$ since this is the only heuristic solution that guarantees no overestimation. Please see Section 3.2 of the revised version for more details."

5) *I would also expect $F_{SCA}$ to be strongly correlated with (mean) catchment elevation so that elevation instead of $F_{SCA}$ could be used in your analysis. As can be seen in Fig. 12, a similar grouping of catchments emerges.*

We have explained the reason why we have used $F_{SCA}$ in Section 4.4 of the revised version: "$F_{SCA}$ is strongly correlated with the mean catchment elevation in our data set ($\rho_{Spearman} = 0.97$, p-value < 0.01, Fig. 13b). A posteriori, we could have considered mean elevation instead of $F_{SCA}$ as a proxy for snowpack persistence. However, a priori, it could be approximative to describe the snow cover persistence only with the increasing elevation: the persistence of snow in a catchment also depends on catchment aspect, topography (Painter et al., 2023) snow-related and climatic characteristics. In fact, catchments with very different characteristics (e.g., different elevation ranges, different areas etc.) can reveal a similar mean elevation, but the snowpack

persistence could considerably change. This is the reason why we decided to focus on $F_{SCA}$ that integrates these physical factors."

6) *L275 mentions that "The Noce Bianco Pian Venezia (NBPV) catchment is an exception since it generally has snow over the glacier also during summer.». As far as I remember, the catchments VdN, DIS and OVA are also partially glacierized. Should they be considered as exceptions as well?*

**We consider only NBPV as an exception because 42% of its area is covered by glaciers. DIS only 2 %, VdN 3 %. For OVA we see it is not covered by glaciers (van Tiel et al., 2020). Thus, for the other catchments, we consider negligible the effect of glaciers on $F^*_{yw}$ (we explicitly say this in the "Study sites" Section of the revised manuscript).**

7) *Fig.10: A very similar result is presented already in von Freyberg et al. (2018) where $F_{yw}$ and the quickflow index QFI, the inverse of the baseflow index, showed a significant positive correlation (note that the QFI and 1/Fbf will likely not be exactly the same, although both were calculated through digital filtering of discharge time series).*

**Thank you for this, we have cited in the discussion that a similar result was found by von Freyberg et al. (2018) (see Section 4.2 of the revised version). However, we want to underline that Duncan (2019) improved the baseflow filter of Lyne, V. D. and M. Hollick (1979)[BaseflowSeparation, EcoHydrology package in R], used by von Freyberg et al. (2018), to separate flow components with physical relevance (Duncan, 2019). von Freyberg et al. (2018) found a positive correlation between $F^*_{yw}$ and $QFI$: average of $(Q-Q_{bf})/Q$ (where $Q_{bf}$ is obtained with Lyne, V. D. and M. Hollick, 1979 baseflow filter). However, $F_{bf}$, average of $Q_{bf}/Q$ = average of $1-QFI$, is not complementary to $F^*_{yw}$ (i.e., $F^*_{yw}$ + average of ($1-QFI$) ≠ 1). A good result of our work is that, using the recession constant commonly proposed in literature (Nathan and McMahon, 1990), the Duncan (2019) baseflow filter returns a $F_{bf}$ that is roughly complementary to $F^*_{yw}$ (i.e., $F^*_{yw}$ + $F_{bf}$ ≃ 1) without any type of calibration. This could be a very useful result for catchments in which isotopes measurements are not available. In these cases, $F^*_{yw}$ could be estimated as $1-F_{bf}$. We clearly explain the relation between $F_{bf}$ and $F^*_{yw}$ in Section 4.2 and Section 4.2.1 of the revised version.**

8) *L484: "In addition, higher order channels, higher up, are more rarely activated than lower order channels that are more often active" If the authors refer to Strahler stream orders here, higher elevation streams usually have low Strahler orders (starting with first-order streams). The Strahler stream orders increase downstream.*

**Thank you for pointing out or language mistakes. Please note that we have removed the sentences about channels order because we do not want to generate confusion: we are studying $F^*_{yw}$ changes with elevation among different catchments and not within a single catchment. Probably, considerations about the age of water flowing in higher/lower order channels is more suitable if looking for water age variations within a single catchment.**

9) *L560-565: Why was BCC not classified as snow-dominated, based on the evidence from previous research?*

This was because of the new classification scheme we proposed in the preprint. However, using the classification scheme proposed by Stoelzle et al. (2020): "According to this classification scheme, the four Italian catchments (DOR, SOU, BCC and NBPV) are all categorized as snow-dominated (S). The classification of BCC is also consistent with the one given in a previous study without considering the application of a formal classification scheme (Penna et al., 2016)." We have written this in "Study sites" Section of the revised version.

10) L566-569: Is it possible that precipitation isotopes in the NBPV catchment were sampled differently compared to the other catchments in this study, e.g. with a heated precipitation collector? This could result in a larger $A_S$ value. Can the authors confirm that the precipitation isotope sampling in the snow-dominated catchments was comparable across all sites?

Thank you for suggesting the clarification of the approach used for sampling precipitation. In NBPV catchment, we did not use a heated precipitation collector. Bulk samples of rain water were collected monthly at the outlet of the catchment by 5-L bottles equipped with a funnel and a layer of mineral oil to prevent evaporation, whereas snow samples were collected using an aluminum cylinder, inserted vertically from the surface to a depth of 20 cm (Zuecco et al., 2019). We applied the same sampling approach of precipitation in NBPV and BCC (Penna et al., 2016). For VdN "bulk rain samples were collected for isotopic analyses using funnels flowing into insulated bags at three locations corresponding to the rain gauges (1,253, 1,500 and 2,100 m a.s.l.), and emptied weekly or biweekly between June 2016 and November 2018. Between February 2016 and April 2018, snow samples were collected from the entire snow profile at various locations in the catchment" (Ceperley et al., 2020). For DOR and SOU precipitation samples were collected at a monthly resolution using a double rain and snowfall isotope sampler installed on a pole 3.7 m high. Therefore, we consider the precipitation isotope sampling comparable across all the new sites, while precipitation isotopes in the 22 sites of von Freyberg et al. (2018) are modeled through an interpolation method. All these information are reported in the revised Supplementary material.

11) L596: "...leads to high baseflow throughout the year...». This contradicts the data shown in Fig. 9. I would suggest to replace 'baseflow' with 'baseflow fractions $F_{bf}$'.

Thank you for this suggestion.

**2.2.3 Technical comments**

1) The language of the manuscript is often not precise and needs to be improved. Some sentences are difficult to understand, e.g.

Thank you for all technical comments and for having taken the time to report the language issues. We improve and clarify the language over the entire manuscript.

2) *(L310) "Additionally, Duncan (2019) provides a specific technique that allows estimation of separate components with physical relevance in the case that baseflow separation techniques were not applied to describe physical processes." This sentence is redundant and not scientifically specific (e.g., what are "separate components with physical relevance"?).*

**The separate components with physical relevance are baseflow and quickflow (i.e., total flow minus baseflow). Specifically, during a rain or snowmelt event, groundwater storage is generally recharged. This recharge is visible in the hydrographs reported in Fig. 8 of the revised manuscript through the "smoothed" baseflow, proposed by Duncan (2019). This "smoothing" simulates a delayed storage contribution to the stream following the recharge phase during an event.**

3) *(L33) "Finally, our work highlights that $F_{bf}$, considered as a proxy for groundwater flow, is roughly the one's complement of $F_{yw}$". Isn't $F_{bf}$ rather a proxy for the groundwater contribution to streamflow? It does not provide any information about flow processes. What does "roughly the one's complement of $F_{yw}$" mean?*

**Probably the term proxy is not correct. $F_{bf}$ is the average fraction of baseflow and it is an estimate of the proportion of the catchment storage (snowpack storage + groundwater storage) contribution to the stream. The meaning of "roughly the one's complement of $F_{yw}$" is that, for each catchment, $F_{bf} + F^*_{yw} \simeq 1$.**

4) *L34 «...we find high $F_{bf}$ during all low-flow periods, which underlines that streamflow is mainly sustained by groundwater in such flow conditions.» That high $F_{bf}$ represents a major contribution of groundwater to streamflow is implicit in the method of Duncan (2019). This is not a new finding.*

**Thank you for this comment. We simply wanted to underline what was the meaning of high $F_{bf}$ during low-flow periods for a general reader. However, we have rewritten this part of the abstract in the revised version.**

5) *L496 "the temporal dynamic of snow accumulation and melt and its effect on deep infiltration supports the pivotal role of snowmelt in recharging groundwater during summer in high-elevation environments ..." This sentence is redundant. Snow melt affects deep infiltration is equivalent to it plays a role in recharge.*

**Thank you. We have modified the sentence in "The temporal dynamic of snow accumulation and melt supports the pivotal role of snowmelt in recharging groundwater during summer in high-elevation environments (Cochand et al., 2019; Du et al., 2019; Flerchinger et al., 1992)." See Section 4.4 of the revised manuscript.**

6) *Sect. 3.2: To indicate whether a variable was flow weighted, earlier papers have added a "*", and thus I would suggest to write $F^*_{yw}$ and $A_S^*$ here as well.*

**Thank you for this suggestion. We have used this notation in the revised manuscript.**

7) *L328: Please verify whether the flow-weighted young water fraction of SOU is indeed 0.01. If so, the following statement "while flow-weighted $F_{yw}$ remains unchanged for the very small lateral subcatchment" is false.*

1090       **Thank you for noticing this. It is simply an error in the text: the flow-weighted young water fraction of SOU is 0.1, not 0.01 and the statement "
[revised manuscript text omitted]

1230 van Tiel, M., Kohn, I., Van Loon, A. F., and Stahl, K.: The compensating effect of glaciers: Characterizing the relation between interannual streamflow variability and glacier cover, Hydrological Processes, 34, 553–568, https://doi.org/10.1002/hyp.13603, 2020.

Weingartner, R. and Aschwanden, H.: Abflussregimes als Grundlage zur Abschätzung von Mittelwerten des Abflusses, Hydrologischer atlas der Schweiz, Tafel 5.2, 1992.

1235    Williams, M. W., Wilson, A., Tshering, D., Thapa, P., and Kayastha, R. B.: Using geochemical and isotopic chemistry to evaluate glacier melt contributions to the Chamkar Chhu (river), Bhutan, Annals of Glaciology, 57, 339–348, https://doi.org/10.3189/2016AoG71A068, 2016.

Wilson, A. M., Williams, M. W., Kayastha, R. B., and Racoviteanu, A.: Use of a hydrologic mixing model to examine the roles of meltwater, precipitation and groundwater in the Langtang River basin, Nepal, Annals of Glaciology, 57, 155–168,
1240    https://doi.org/10.3189/2016AoG71A067, 2016.

Zuecco, G., Carturan, L., De Blasi, F., Seppi, R., Zanoner, T., Penna, D., Borga, M., Carton, A., and Dalla Fontana, G.: Understanding hydrological processes in glacierized catchments: Evidence and implications of highly variable isotopic and electrical conductivity data, Hydrological Processes, 33, 816–832, https://doi.org/10.1002/hyp.13366, 2019.

---

## Author Response (AR2)

**Author's response: a point-by-point response to the II round of reviews.**

Alessio Gentile[1], Davide Canone[1], Natalie Ceperley[2,3], Davide Gisolo[1], Maurizio Previati[1], Giulia Zuecco[4,5], Bettina Schaefli[2,3], and Stefano Ferraris[1]

[1]Interuniversity Department of Regional and Urban Studies and Planning (DIST), Politecnico and Università degli Studi di Torino, Torino, Italy
[2]Institute of Earth Surface Dynamic (IDYST), Faculty of Geosciences and Environment (FGSE), University of Lausanne, Lausanne, Switzerland
[3]Institute of Geography (GIUB) and Oeschger Centre for Climate Change Research (OCCR), University of Bern, Bern, Switzerland
[4]Department of Land, Environment, Agriculture and Forestry (TESAF), University of Padova, Legnaro, Italy
[5]Department of Chemical Sciences (DiSC), University of Padova, Padua, Italy

*Correspondence to*: Bettina Schaefli (bettina.schaefli@giub.unibe.ch)

Dear Editor and Referees,

we would like to thank you for the appreciation of our efforts in revising the work. On one side, we are happy to see that Jana von Freyberg considered as excellent the scientific significance, scientific quality, and presentation quality of our work and that she accepted as is our manuscript for final publication. On the other side, considering the anonymous referee #1 comments, the Editor decided that publish is subject to revisions. The anonymous referee #1 comments have been very constructive for the paper improvement. The anonymous referee #1 helped the authors providing a code to satisfy her/his requests and we thank her/him very much for this. We have addressed all the issues raised in the Report #1 and all the applied changes are visible in the track-changes version of the manuscript.

Since the number of comments of the anonymous referee #1 is limited, we directly provide a point-by-point response to her/his comments also including info about the changes made in the manuscript.

In the hope of having met your scientific expectations in the revised manuscript, we kindly ask you to reconsider the publication of our work on the Hydrology and Earth System Sciences Journal.

With king regards,

The Authors

**1 Response referee #1**

*The authors have made significant changes to the manuscript. I am satisfied with the authors' responses as well as the changes. I think the quality of the manuscript was significantly improved. In general, I only have one major comment regarding the $F^*_{yw}$ and the age threshold of $F^*_{yw}$ (please see below).*

**Dear referee #1,**

**We would like to thank you for the positive assessment and the detailed comments (including the source code for estimating $F^*_{yw}$ and α), which contributed to our manuscript's improvement considerably.**

**Please find below a point-by-point response to both your main and minor comments. We have incorporated all your constructive feedback in the revised manuscript.**

**Sincerely,**

**The Authors**

**1.1 Major comments**

*As I understood from the manuscript. The authors use $F^*_{yw}$ from 22 Swiss catchments from Freyberg et al. (2018). I am not sure if the authors and Freyberg et al. (2018) used the same code (method) for the calculation $F^*_{yw}$ or not. In the attached data along with this review, I demonstrated that using different source codes (methods) could result in different values of $F_{yw}$ (or $F^*_{yw}$) and the age thresholds of $F^*_{yw}$. If so, the differences in $F^*_{yw}$ among catchments could be driven by using different methods (source code) rather than the physical factors (e.g., precipitation, elevation, LFD, Fbf,...). Therefore, I would suggest the authors to use the same source code (method) for calculating $F^*_{yw}$ for all catchments and provide the source code for review/or make it publicly accessible.*

**The source code for calculating $F^*_{yw}$ was different from that of von Freyberg et al. (2018). Accordingly, based on the source code provided by the anonymous referee #1, we developed a © Matlab code for estimating $F^*_{yw}$. This code was directly applied to the isotope data of all the 27 study catchments (consequently, we did not consider published $F^*_{yw}$ values: the $F^*_{yw}$ you will find in the revised version are a result of our work). The © Matlab code has been made available in the Supplementary material of the paper. The used methodology has been thoroughly described in Section 3.1 of the revised manuscript.**

70 *When applying the sine wave fitting, the authors will get $F^*_{yw}$ corresponding to a specific age threshold ($\tau_{yw}$), depending on the shape factor (alpha) of the gamma distribution (Equations 13-14 from Kirchner 2016). The age threshold could be 1 to 3 months (Figure 10, Kirchner 2016) and even beyond this range. Therefore, the variation in $F^*_{yw}$ might be because of different age thresholds rather than physical factors. In this sense, the $F^*_{yw}$ or $F_{yw}$ might not be a useful matrix for catchment intercomparison studies. Therefore, I suggest showing the alpha and age thresholds for each catchment in the results (these*

75 *values were often not shown in previous studies). Discuss the consequences of the differences found with the age threshold in the limitation section of the paper (e.g., will the relations between $F^*_{yw}$ and other factors (e.g., precipitation, elevation, LFD, Fbf,...) change if somehow we can calculate $F^*_{yw}$ that corresponds to the same age threshold).*

**Based on the source code provided by the anonymous referee #1, we developed a © Matlab code for estimating α and**

80 **$\tau_{yw}$. The © Matlab code has been made available in the supplementary material of the paper. As anticipated by the anonymous referee #1, we find different $\tau_{yw}$ for each study catchment, also if they vary modestly between about 1.5 to 3 months, in agreement with Kirchner (2016a). We explicitly said what should be the effect of choosing a constant threshold age and that using the amplitude ratio approach (as in our work) $F^*_{yw}$ estimates refer to the proportion of runoff younger than a threshold age that is different among the studied catchments, and that this is the main**

85 **limitation of our work. In this regard, please see completely new Section 4.1 of the revised manuscript. In addition, in all the figures in which the $F^*_{yw}$ values are reported, we set the points size as proportional to the threshold age for having a more complete and exhaustive visualization of the results.**

**1.2    Minor comments**

90

*Figure 2c: Might be using a triangle (for P) and a circle (for Q) in combination with different colors (for rain, hybrid, and snow) for easier to differentiate between them.*

**Thank you for this comment. We use triangle for P and circle for Q in combination with different colours as you have**

95 **suggested. We remove the catchments labels to improve the visualization. Please note that we have changed all the figures importing them as \*.svg (scalable vector graphics) file.**

*L237-238: Should be noted that this demonstration uses thought experiments (not from real catchments)*

100 **Thank you for this. In Section 3.1, line 245 of the revised manuscript, we write: "_By using thought experiments_, Kirchner (2016a) has demonstrated that for a given shape factor α and across a wide range of scale factors β, the**

**theoretical young water fraction can be accurately predicted by the amplitude ratios of seasonal sine curves fitted to stream water and precipitation isotope values."**

105    *Figure 6: Why did the authors only show quaternary deposits for 7 catchments, I think either showing for all catchments or not showing the figure or not showing (Table 2 is sufficient)*

**Thank you for this suggestion. We have removed the figure from the revised manuscript.**

110    L207-211: "Our data set includes NBPV, whose area is 42% glacier covered and consequently shows a characteristic glacier-dominated streamflow regime with a monthly peak in late summer. Thus, NBPV may belong to a fourth category of glacier-dominated catchments, for which the effect of glacier-melt on $F^*_{yw}$ cannot be neglected, and this was partially discussed by Ceperley et al. (2020)": Why did the author still classify NBPV as "snow-dominated" catchment? Does glacier-melt account for when calculating $F^*_{yw}$?

115

**We do not classify NBPV as glacier-dominated since the classification provided by Stoelzle et al. (2020) does not consider this regime and the definition of the classifiers for a new category is outside the scope of this work (as also suggested by Jana von Freyberg in her previous review). However, we say that our results suggest that NBPV should belong to a fourth category of glacier-dominated catchments. In this regard, please see lines 210-215 of the revised**

120    **manuscript. Glacier-melt accounts for when calculating $F^*_{yw}$ since we are using the direct-input approach where the snowpack and/or the glacier are considered as part of the catchment storage. This is clarified in Section 3.1 (lines 274-275) of the revised manuscript.**

*Section 4.1. Why did the authors discuss the Winter Flow Index (WFI) - $F^*_{yw}$ relation in this section "The role of Quaternary*

125    *deposits"?*

**This is because Arnoux et al. (2021) found a strong positive correlation between $F_{qd}$ and Winter Flow Index (WFI). So, we discuss here the relation Winter Flow Index (WFI) - $F^*_{yw}$ to compare our results with previous findings of Arnox et al (2021). We have partially rewritten Section 3.3 to better underline our choice of discussing WFI- $F^*_{yw}$**

130    **relation in this section.**

Section 4.2. To be consistent, I would suggest using another title, e.g., "the role of groundwater flow (baseflow) in F*yw"

**Thank you, we use the title you have suggested in the revised version.**

135

*L414: I think the author mentions the fraction of based flow, not the amount of stored water.*

**Yes, thank you for noticing this. We have replaced "stored water" with "base flow".**